# Bead-jet printing enabled sparse mesenchymal stem cell patterning augments skeletal muscle and hair follicle regeneration

Yuanxiong Cao [1,2,5], Jiayi Tan [1,2,5], Haoran Zhao[1,2,5], Ting Deng[1,2,5], Yunxia Hu [1,2], Junhong Zeng [1,2], Jiawei Li[1,2], Yifan Cheng[1,2], Jiyuan Tang[1,2], Zhiwei Hu[1,2], Keer Hu[1,2], Bing Xu[1,2,3], Zitian Wang[1,2], Yaojiong Wu[1,2], Peter E. Lobie [1,2,3] & Shaohua Ma [1,2,3,4] ✉

Transplantation of mesenchymal stem cells (MSCs) holds promise to repair severe traumatic injuries. However, current transplantation practices limit the potential of this technique, either by losing the viable MSCs or reducing the performance of resident MSCs. Herein, we design a "bead-jet" printer, specialized for high-throughput intra-operative formulation and printing of MSCs-laden Matrigel beads. We show that high-density encapsulation of MSCs in Matrigel beads is able to augment MSC function, increasing MSC proliferation, migration, and extracellular vesicle production, compared with low-density bead or high-density bulk encapsulation of the equivalent number of MSCs. We find that the high-density MSCs-laden beads in sparse patterns demonstrate significantly improved therapeutic performance, by regenerating skeletal muscles approaching native-like cell density with reduced fibrosis, and regenerating skin with hair follicle growth and increased dermis thickness. MSC proliferation within 1-week post-transplantation and differentiation at 3 – 4 weeks post-transplantation are suggested to contribute therapy augmentation. We expect this "bead-jet" printing system to strengthen the potential of MSC transplantation.

Volumetric muscle loss (VML) injury is a common occurrence in severe extremity trauma. Regeneration of VML poses a significant challenge, which usually leads to functional impairment with limited regeneration capacity[1,2]. Other open trauma, such as integumentary damage, requires prevention of infection and regeneration of exocrine function and hair follicles to recover normal physiology and appearance[3]. Stem cell therapy, as a potential technology in regenerative medicine[1,4–7], promotes the repair of diseased, or injured tissues using stem cells or their derivatives. For example, human mesenchymal stem cells (MSCs) have been utilized in multiple exemplars of tissue regeneration, such as skin[8], musculoskeletal tissue[9], and endometrium[10].

However, direct injection of stem cells into a recipient injury results in suboptimal therapeutic outcomes, due to the hostile environment within damaged and diseased tissues, producing limited retention and survival of injected cells[11]. For example, most clinically relevant treatments require up to 200 million MSCs per dose[12,13], and yet it is reported that <5% of administered MSCs are present in the tissue a few hours after transplantation[13]. Furthermore, stem cell transplantation is expensive and limited to scalable cell availability, especially for autologous transplantation. Therefore, the maximization of stem cell usage with limited availability poses a significant clinical challenge.

[1]Tsinghua Shenzhen International Graduate School (SIGS), Tsinghua University, 518055 Shenzhen, China. [2]Tsinghua-Berkeley Shenzhen Institute (TBSI), 518055 Shenzhen, China. [3]Shenzhen Bay Laboratory, 518055 Shenzhen, China. [4]Institute for Brain and Cognitive Sciences, Tsinghua University, 100084 Beijing, China. [5]These authors contributed equally: Yuanxiong Cao, Jiayi Tan, Haoran Zhao, Ting Deng. ✉e-mail: ma.shaohua@sz.tsinghua.edu.cn

Scaffolding hydrogel encapsulation does improve stem cell retention and increases stem cell survival by moderating the extracellular environment[14–16]. Scaffold design is critically important[17,18], including material selection and spatial distribution of the scaffolding matrix. Moreover, it was reported that the spatial distribution of stem cells at both micro- and macro-scales affects their activity and therapeutic potential[19,20]. Traditionally, tissue damage is repaired by cell-laden or acellular bulk hydrogel that has nanoporous mesh. Dense gel usage produces limitations in recruiting surrounding cells to fill the implanted volume or accepting signaling cues and nutrient factors to feed the encapsulated cells. Bulk gel with engineered micro-porosity facilitates the transport of cells and signaling and nutrient molecules, but it still results in limited translocation and inhomogeneous regeneration for a large volume[21]. Granular microgel assembly produced using microfluidics offers a competitive capacity for promoting tissue regeneration and microtissues assembly, by providing both nano-porosity from the materials and microporosity from the structural assembly[22,23]. They are injectable, scalable, and adaptive to irregular substrates, and meanwhile, provide proper physical and mechanical cues for encapsulated or interspaced cells. The granular microgel assembly demonstrates improved efficacy in both cell-laden or acellular injections[3,10,21,24–27].

Both stem cell transplantation and granular microgel injection have proven strengthening effects in regenerative therapy. Stem cell-laden microgels, with the combination of both advantages, could be expected to further facilitate wound repairs. Matrigel, similar to extracellular matrix (ECM) for its basement membrane-like nature, was selected as the biomaterial in this research. Matrigel has advantages over commercial synthetic biomaterials in directing stem-cell rate[28]. The interaction of skin-derived stem cells with Matrigel gives rise to a series of spatially and temporally coordinated events that regulate the stem-cells fate[29].

However, further studies to elaborate on the underlying cues and the implementation techniques for improved translational and clinical application potential are lacking. For example, the distribution manner of cell-laden microgels has not been explored, nor is there any associated technique that allows the precision transplantation of individual building blocks. Cell density is another critical determinant in tissue engineering. Generally, high cell population density, before approaching confluence, benefits cell growth, and synthesis and remodeling of the extracellular matrices[19,30,31].

3D bioprinting is the additive manufacturing of bioactive substances by precision patterning of building blocks comprising cells, biomaterials, and bioactive factors to mimic native tissues[31–34]. Though the current reported 3D bioprinting techniques can tackle natural defects with irregular injury, they are limited in processing Matrigel or similar bioactive ultra-soft materials (elastic modulus <500 Pascal[17]) without any rigid supports. Ink-jet printing splashes the ink phase, losing the spatial precision and $z$-axis accumulation[35]; microextrusion is not compatible with slow-gelling and flowable materials[36]; and laser-assisted printing is limited to photocrosslinkable materials[37]. Using gelated Matrigel beads as the printing building blocks can access highly reproducible and definable large-scale stem cell patterning, but the capable technique development is not reported. Aspiration-assisted bioprinting[38] can position soft microgels with high precision and retention of bead integrals but is not capable of scale-up fabrication and is less efficient.

Herein, a protocol and associated technical platform for translational stem cell therapy have been established. Taking MSCs as an exemplar, the effects of distribution profile on MSCs functions were first explored, and it was observed that sparsely distributed high cell density MSCs-laden Matrigel beads exhibited substantially augmented MSCs proliferation, migration, and EV production. To enable automated sparse-patterning of regionally dense MSCs, a bead-jet printing system was developed, with the superior performance of sparsely

dense MSCs in promoting VML and skin repair. It must be noted that using Matrigel as the MSCs encapsulation scaffold provided high performance not attained by the gelatin counterpart, suggesting that, in regenerative medicine, gelatin is not a substitute for Matrigel or equivalent materials. Secondly, sparse patterning of high-density MSCs-laden beads among acellular beads also enlarged the transplantation coverage, thus enabling therapy of large-scale traumatic injury with a limited number of MSCs.

## Results

### MSCs-laden Matrigel beads and bead-jet printing

MSCs encapsulated in Matrigel beads gain advantages in microenvironment recapitulation, including efficient volumetric mass transport and versatile structural definition. Intra-operative printing of building blocks, e.g., MSCs-laden beads, facilitates the translation of stem cell therapy because it allows on-demand patterning and scaling of MSCs coverage. It is also speculated that cell distribution in the printed volume would affect the regeneration outcome. To elaborate on the different conditions, bead-jet printing was established. The regenerative performance of printed Matrigel beads loaded with high-density and low-density MSCs, and cast Matrigel bulk gel loaded with high-density MSCs was compared. The empty space was printed with acellular beads or cast with acellular bulk gel.

To proceed with bead printing, a "bead-jet printer" was first developed. This printing setup consists of a module for cell encapsulation and Matrigel bead formulation, and a second module for bead printing (jetting) (Fig. 1a and S1). The two modules operate in synchronization. In the formulation module, the cell suspension solution, e.g., Matrigel solution, is injected into the cascade tubing microfluidics (CTM) and pinched off periodically under the oil shearing effect, the capillary effect and the tubing-boundary condition, forming monodispersed cell-laden Matrigel droplets (Fig. S2). The droplets are conducted along the long, incubation tubing under a constant oil flow, and gradually solidify before approaching the tubing outlet. The outlet is connected to a 3-port connector, with an orthogonal channel open to a compressed air inlet. Compressed air is instantaneously generated once a sensor ~5 cm above the connector is activated by a passing bead. The air jet shoots a bead onto the substrate along a straight path. This process is repeated to print MSCs-laden beads following programmed commands. During printing, the subject animal is placed on the support platform, with the traumatic injury facing upwards to receive the ejected beads. This contactless bead-jet printer is compatible with a wide range of large open injuries.

Next, the bead-printing performance in repairing a muscle volumetric loss defect and skin injury was studied. Technically, there are three ways to deliver stem cell-laden Matrigel composites onto injury sites: high cell density beads (HDB) printed in sparse patterns, low cell density beads (LDB) printed in dense patterns, and high cell density bulk gel (HDG) cast onto the injury site (Fig. 1b). In all conditions, the initial cell number remains constant. For the bead-printing groups, the high-density MSCs-laden beads are printed in sparse patterns, interspaced by acellular beads, to generate equal transplantation coverage and transplanted cell number in accord with the low-density group (Fig. 1c).

### Capabilities of bead-jet printing platform and patterning of biologics

We first used gelatin beads to examine the bead-jet printing efficacy. A 5% (w/v) gelatin solution was formulated into uniform microbeads in the CTM module, crosslinked by a saturated Transglutaminase (TG) solution. To increase the cell encapsulation capacity, large beads (diameter, 650 μm) that approached the mass diffusion limit throughout the study were used. The bead-jet printing system is capable of printing different structures, in either homogeneous or heterogeneous arrangements of microbeads. Gelatin microbeads

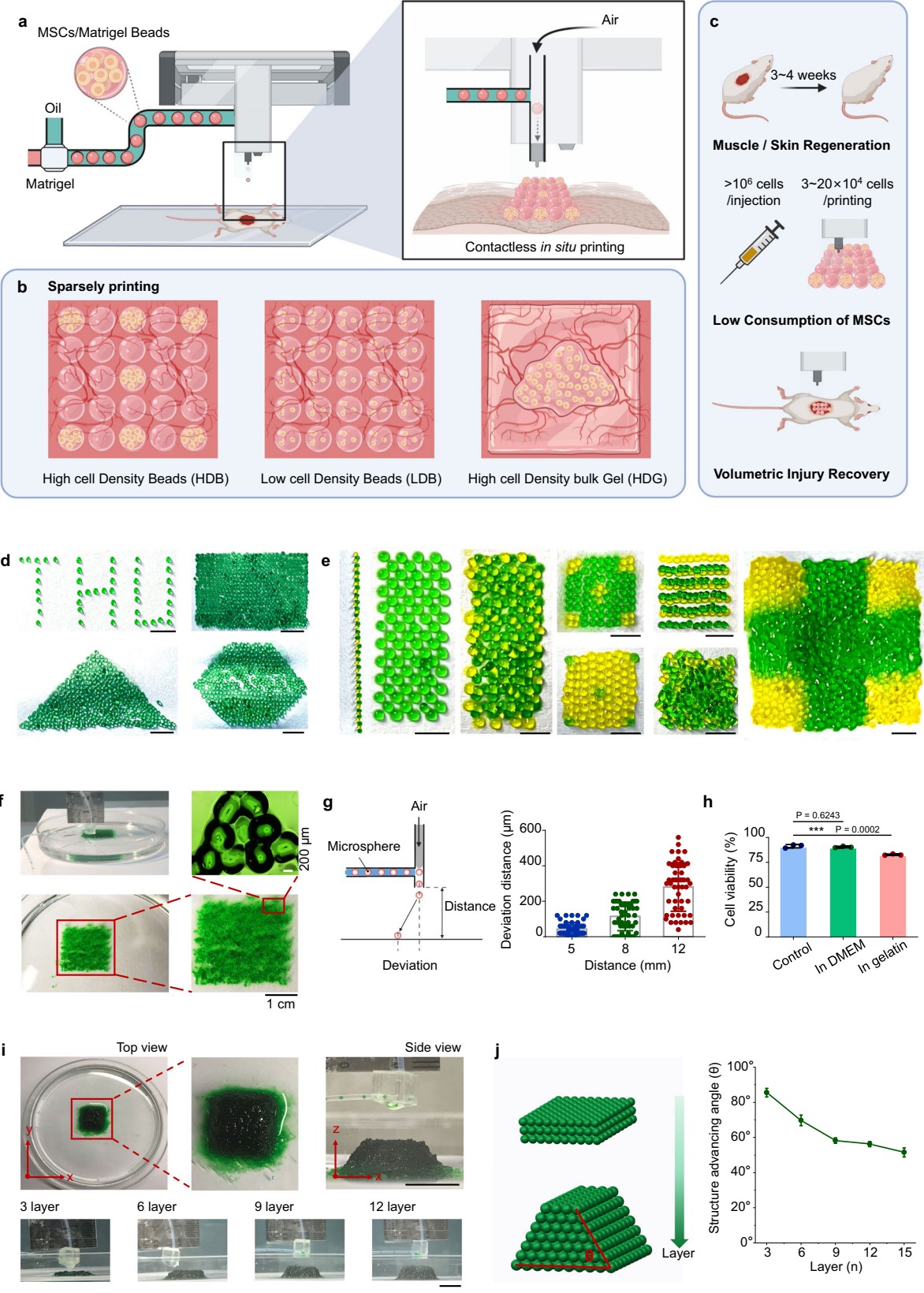

were printed onto fabric paper, forming a THU alphabetical pattern in a sparse array, and rectangular, triangular, and hexagonal patterns in dense connection (Fig. 1d). Two types of gelatin microbeads, yellow and green, with heterogeneous organization were printed (Fig. 1e). It may offer a solution to improve mimicking of tissue microenvironment, by patterning multiple types of cells and tissue

elements within one engineered architecture. Sacrificial support has positive role in bead-jet printing. It anchors the printed building blocks toward larger architecture on demanded positions. The printing bead bonds with the support of the neighboring beads by forming a robust gel (glue) layer by solidifying the fused interface upon contact. The mono-layer assembly exhibited precise bead-

**Fig. 1 | Schematic illustration and characterization of bead-jet printing. a** High-throughput intra-operative formulation and printing of MSCs-laden Matrigel beads. **b** Parallel comparison of printed high cell density beads (HDB) and low cell density beads (LDB), or cast high cell density bulk gel (HDG). The empty space was printed with acellular beads or cast with acellular bulk gel. **c** Bead-jet printing augments stem cell retention, reduces stem cell consumption, and increases volumetric injury recovery in muscle and skin regeneration. **d** Homogenous structures of printed gelatin beads, including a THU alphabetical sparse pattern, and rectangular, triangular, and hexagonal dense patterns. Scale bar, 5 mm. **e** Heterogeneous structures of printed gelatin beads of two different colors. Scale bar, 5 mm. **f** Mono-layer printed gelatin beads preserved the bead-to-bead interface and formed bead bonds on a sacrificial gelatin support. **g** Printing precision evaluation by measuring the

deposition deviation at ejection distances of 5, 8, 12 mm. $n = 50$ for each measurement. Data are represented as mean ± SD. **h** Viability ratios of MSCs in gelatin beads, measured before printing (positive control), after printing straight into the DMEM medium, and after printing first on sacrificial gelatin support followed by transferring to the DMEM medium. The viability rates were measured after 3 days in culture. $n = 3$ for each measurement. Data are represented as mean ± SD (standard deviation), $**P < 0.01$. The significant difference is determined by one-way ANOVA, followed by Tukey's test. **i** 3D Printing of multi-layer gelatin beads on sacrificial gelatin support. The printed assembly had 15 layers. Scale bar, 1 cm. **j** Advancing angle measurement of 3D printed bead assemblies with increasing layers. $n = 3$ for each measurement. Data are represented as mean ± SD.

positioning, integral maintenance, and bead-bonding (Fig. 1f, Fig. S3 and Video S1).

The printing resolution was evaluated by calculating the standard deviation of the printed bead position with respect to the targeted position. The resolution reached 70 µm, ~1/9 of a bead diameter when the ejection distance was 5 cm. It reached 300 µm when the distance was 12 mm (Fig. 1g). The increased deviation resulted from beads colliding with the vertical channel wall under the airflow. The beads may also be printed directly into the culture medium, and onto sacrificial and irregular supports, thus enabling in-situ or intra-operative printing.

Next, the effect of printing on cell viability was evaluated. The viability of MSCs encapsulated in gelatin beads was determined before printing, after printing into the culture media, and upon first printing onto sacrificial gelatin support followed by transfer to the culture medium. The printed groups were examined after 3 days in culture. The cell viability ratios before and after printing directly into the culture medium were nearly identical ($91.3 \pm 1.8\%$ and $90.4 \pm 0.9\%$, respectively), suggesting that the shear force in both bead formulation and jet printing exhibited minimal detrimental effects on cells (Fig. 1h). However, when the MSCs-laden beads were first printed onto the sacrificial support and then transferred to the medium, the viability ratio decreased to $82.7 \pm 0.8\%$. This loss of viability might be attributed to the dehydration occurring during printing, and could be solved by increasing the environmental humidity or by pre-hydration of the dish.

The bead-jet printing system can distribute beads into large-scale and defined patterns. For planar printing, the printed pattern covered nearly 100 cm$^2$ in this current in-house machine (Fig. S4). The printing scale could be further increased by improving the locomotion ability, i.e., the $x$–$y$ movement range, of the printing nozzle. For three-dimensional (3D) patterning, a 15-layer assembly ($2.5 \times 2.5 \times 1$ cm) (Fig. 1i) and a 50-layer 3D heterogeneous structures ($2 \times 2 \times 3$ cm) (Fig. S5) were printed. In this multi-layer printing, the bead positioning mismatch accumulated and appeared with an advancing angle that decreased from ~90° to ~50° as the layer counts increased from 3 to 15 (Fig. 1j). In this bulky assembly, the bead integrity and resolution were maintained (Fig. S6) and the increase in $z$ dimension was not significantly compromised, thus overcoming the limitations of ink-jet printing[35].

In addition, the bead-jet printer uses oil to pinch cell-laden liquid into monodispersed droplets that later were gelled into cell-laden beads. Thus, the bead formulation and printing process are accompanied by oil occupation. For bead-jet printing, a highly volatile and thin fluorinated oil, HFE7000 (3M) was used; which was cytocompatible and quickly evaporates in the printing process. The dense cytoarchitecture developed in a bead demonstrated the feasibility of HFE7000 as the sacrificial fluid (Fig. S7).

## Rheological properties and porosity characterization of microbead assembly and bulk gels

Microbead assembly is inherently porous and modulable, which could facilitate the infiltration of blood and cell transportation, thus possessing obvious advantages over the bulk gel. To investigate the

therapeutic potentials of microbead assemblies, different granular materials generated using a bead-jet printing system were tested for their rheological properties and porosity and compared with their bulk gel counterparts. First, rheological tests were performed to evaluate the granular and bulk gel properties of gelatin and Matrigel (Figure S8). Frequency sweep showed that the storage modulus of gelatin and Matrigel microbead assembly and their bulk counterparts at low frequency exhibited linear trends (Fig. 2a). Shear-thinning behaviors were also observed across the four groups by reduction in viscosity with increasing shear rates (Fig. 2b). Strain sweep test showed that all samples yielded ($G'' > G'$) at high strains (Fig. 2c). The yield strain of bulk gelatin was significantly higher than the granular gelatin (Fig. 2d, i). However, no significant difference in yield strains of Matrigel between its bulk and granular configurations was observed (Fig. 2d, ii). There was no significant difference in storage moduli of gelatin and Matrigel in their granular and bulk configurations (Fig. 2e). Further, granular Matrigel and bulk gel also exhibited similar properties on material degradation. Both acellular materials maintained structural integrity after being immersed in the culture medium at 37 °C for 1 week (Fig. S9). It suggested that the microbead configuration did not change the inherent properties of Matrigel.

Next, the interstitial porosity of microbead assemblies of gelatin, Matrigel, HAMA, and GelMa was investigated by adopting different bead sizes ($D = 650$ and $450$ µm) and degree of packing (low packing and high packing by different centrifugation rates). The microbead assemblies were infiltrated with FITC-dextran (Sigma-Aldrich, 2 M Da, 10 mg/mL in PBS) solution. Assembly of small gelatin beads ($D = 450$ µm) at low packing density had a void fraction of ~14.1%, significantly higher than the void fraction at high packing density (~1.3%). Assembly of large gelatin beads ($D = 650$ µm) at low packing density had a void fraction of ~11.5%, significantly higher than the void fraction at high packing density (~1.5%) (Fig. 2f–h, Video S2).

Likewise, the porosity of granular Matrigel was investigated (Fig. 2i–k, Video S3). Assembly of small Matrigel beads ($D = 450$ µm) at low and high packing density reached the void fractions of ~3.9% and ~2.4%, respectively. Assembly of large Matrigel beads ($D = 650$ µm) at low and high packing density reached void fractions of ~8.3% and ~1.7%, respectively. Both the void fractions at low-packing and high-packing, as well as the difference resulting from small bead packing and large bead packing, were decreased in granular Matrigel and granular gelatin. It might be attributed to the high deformability of Matrigel than gelatin under high centrifugation at $20,879 \times g$ for 5 min. Similar results were also obtained by using GelMA and HAMA microgels (Fig. S10, and Videos S4 and S5). The results proved that the granular gel, i.e., the microbead assemblies, possessed macroporosity, spanning tens to hundreds of microns, that may benefit cell invasion, and the porosity is tailorable by both the composing beads and their packing densities.

## Characterization of MSCs cultured in Matrigel HDG, LDB, and HDB groups in vitro

Matrigel was used as the MSCs in vitro culture and transplant scaffolding material. To test their behavior under different conditions,

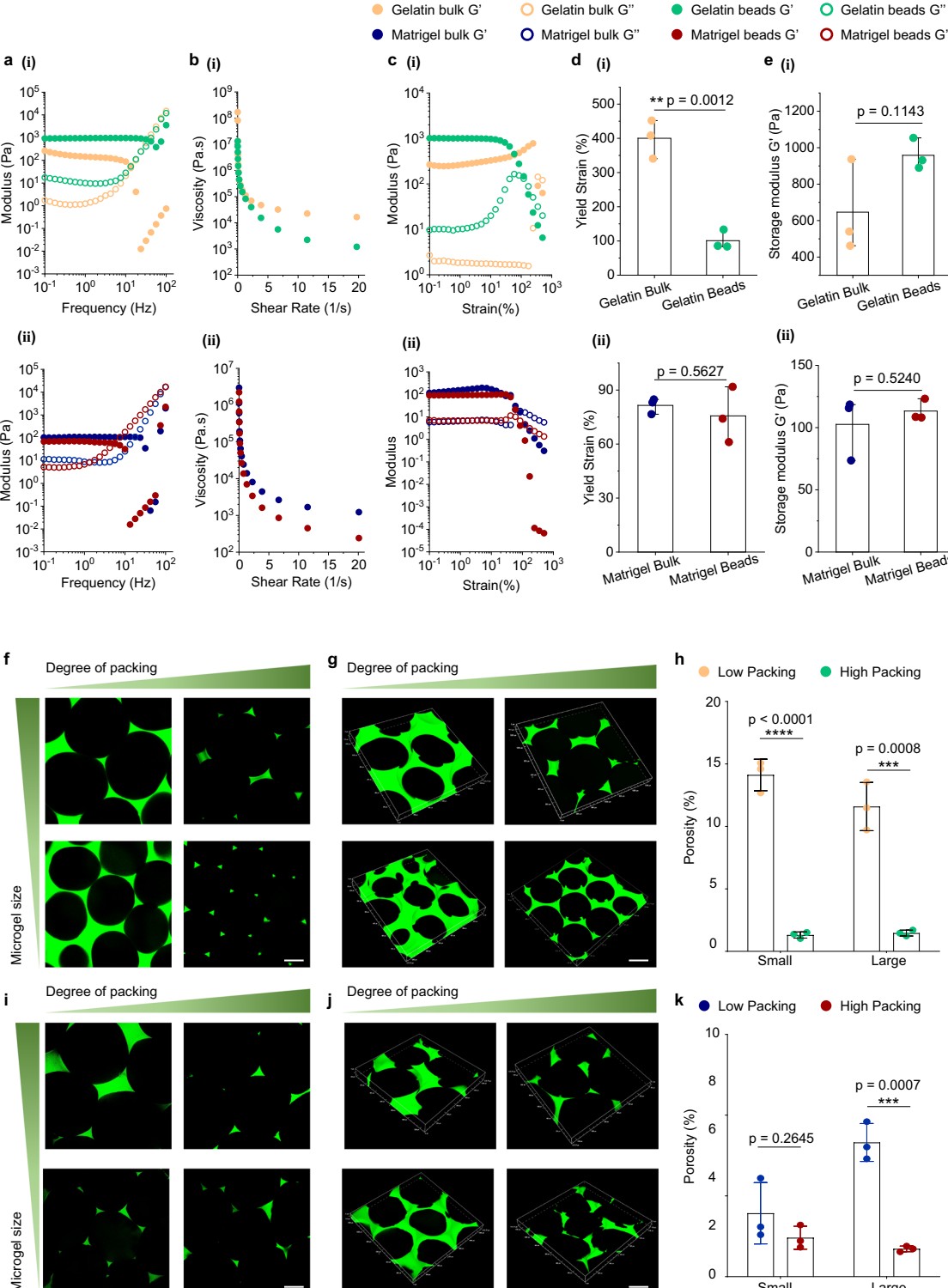

**Fig. 2 | Rheological properties and porosity characterization of gelatin and Matrigel microbead assemblies, i.e., granular microgels, and bulk gels.**
**a** Frequency sweep performed from 0.1 to 100 Hz of gelatin (i) and Matrigel (ii) microgels and bulk gels. **b** Shear-thinning properties of gelatin (i) and Matrigel (ii) microgels and bulk gels. **c** Strain sweep performed from 0.1% to 500% of gelatin (i) and Matrigel (ii) microgels and bulk gels. **d** Yield strains of gelatin (i) and Matrigel (ii) microgels and bulk gels. $n = 3$ for each group. **e**, Storage modulus $G'$ of gelatin (i) and Matrigel (ii) microgels and bulk gels. $n = 3$ for each group. **f–h** Representative confocal 2D (**f**) and 3D (**g**) representative images of granular gelatin microgels assembled from different microbeads ($D = 450$ and 650 μm) and packing densities. The bead samples were infiltrated with FITC-dextran (2 M Da, 10 mg/mL) solution.

**h** Quantification of granular gelatin porosity. $n = 3$ for each group. **i–k** Representative confocal 2D (**i**) and 3D (**j**) representative images of granular Matrigel microgels assembled from different microbeads ($D = 450$ and 650 μm) and packing densities. The bead samples were infiltrated with FITC-dextran solution.
**k** Quantification of granular Matrigel porosity. $n = 3$ for each group. Here, the high-packing and low-packing conditions were reached by centrifugation of the microbead assembly for 5 min at 93×$g$ or at 20,879×$g$. Scale bar, 200 μm. The data are represented as mean ± SD. The significant difference is determined by unpaired two-failed Student's $t$-test. **$P < 0.01$, ***$P < 0.001$; ****$P < 0.0001$.

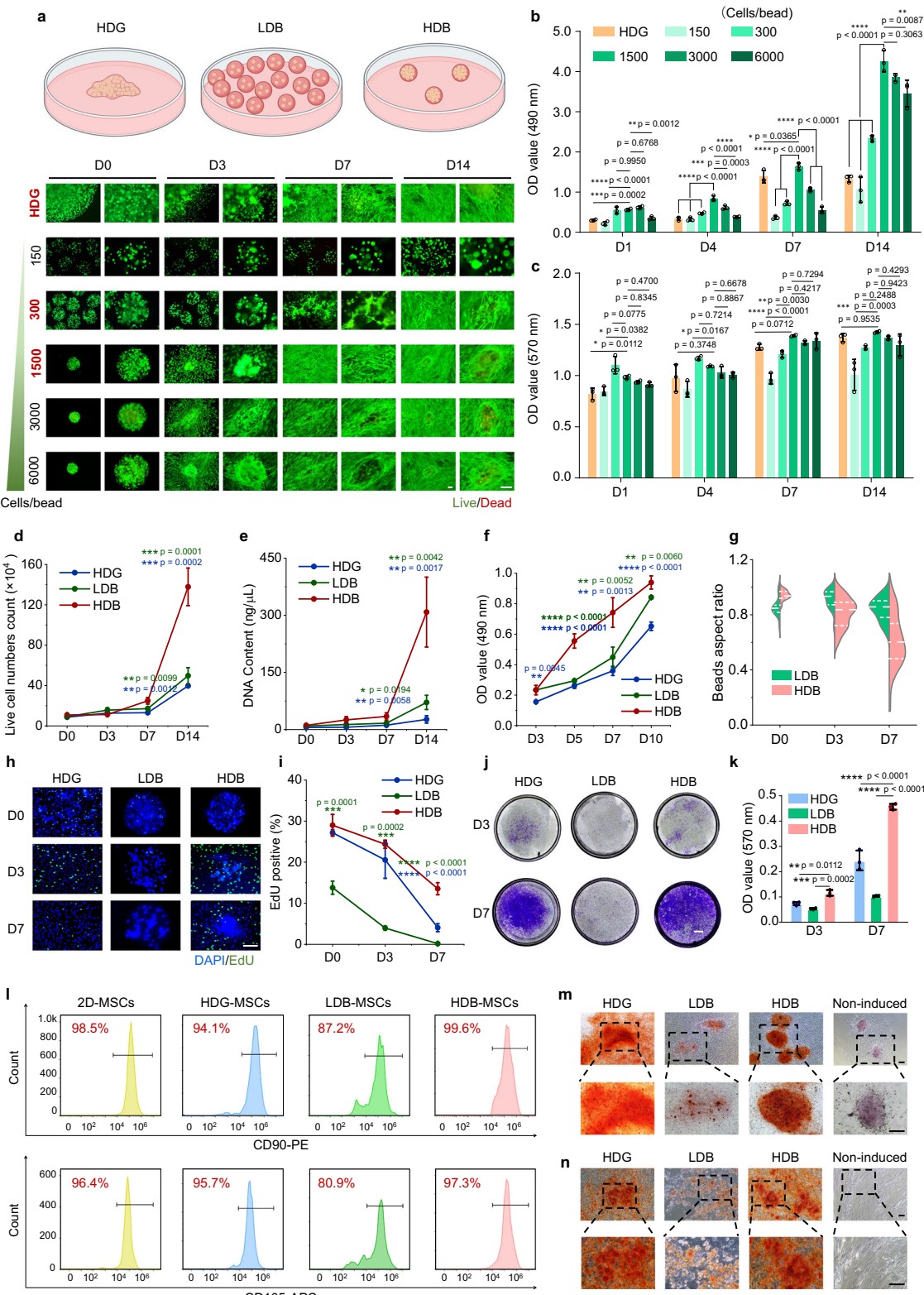

MSCs growth was tracked under different dosages using LDH-Cy Quant assay, cell metabolic test, and live/dead staining, by encapsulating 150, 300, 1500, 3000, and 6000 cells per microbead (Fig. 3a, Table S1). The total cell number was kept consistent among these groups, including the bulk gel condition. The LDH-Cy Quant assay showed that the high cell density of MSCs gained a significantly higher proliferation rate compared with the low cell density of MSCs and bulk gel of MSCs (Fig. 3b). After 4, 7, and 14 days in culture, MSCs growth

rates of 1500-cells-per-microbead were significantly higher than the other groups, including both the lower dosage and higher dosage conditions.

Alamar Blue assay was employed to test cell metabolic activity. After being cultured for 7 days, the cell metabolic activity in high-cell-density groups, including 1500, 3000, and 6000-cells-per-bead, were significantly higher than in the low-cell-density group, including 150 and 300-cells-per-bead (Fig. 3c). Cell viability across Matrigel HDG, and

**Fig. 3 | Characterization of MSCs proliferation, migration, and stemness possession in Matrigel HDG, LDB, and HDB groups. a** The schematic illustration of MSCs growth in the HDG, LDB, and HDB groups. Cell viability of MSCs in HDG and microbeads of different cell densities, including 150, 300, 1500, 3000, and 6000 cells per microbead, were characterized at day 0 (D0), D3, and D7, and D14 using live/dead staining. The initial cell numbers were kept consistent among all groups. Scale bar, 100 μm. **b** OD values of MSCs growth at 490 nm at D1, D3, D7, and D14, determined using the LDH-CyQuant assay. $n = 3$. **c** Metabolic activity of MSCs at D1, D3, D7, and D14, determined using the Alamar Blue assay. $n = 3$. **d** and **e** Quantification of cell number (**d**) and DNA content (**e**) of MSCs at D0, D3, D7, and D14. **d** Live cell counts using a hemocytometer. $n = 3$. **e** DNA content acquisition. $n = 3$. **f** OD values of MSCs growth at 490 nm at D3, D5, D7, and D10, determined using the MTT assay. $n = 3$. **g** Aspect ratios of Matrigel LDB and HDB beads at D0, D3, and D7. $n = 40$. **h** and **i** Representative images of EdU labeling (**h**) and quantification (**i**) of EdU-positive MSCs at D0, D3, and D7. $n = 3$. Green: EdU, blue: DAPI. Scale bar, 200 μm. $n = 3$. **j** and **k** Crystal violet staining of cell migration assay (**j**) and OD values at 570 nm of crystals dissolved in ethylic acid (**k**). $n = 3$. Scale bar, 1 mm. **l** Flow cytometry quantification of MSC markers CD90, CD105. MSCs were cultured in Petri dishes (2D condition), and in Matrigel HDG, LDB, and HDB groups for 4 days. **m** Osteogenic differentiation of microbead-MSCs from D0. **n** Osteogenic differentiation of MSCs retrieved from microbeads. Scale bar, 100 μm. **b–g, i, k** The data are represented as mean ± SD. The significant difference is determined by one-way ANOVA, followed by Tukey's test. The significant differences between parallel groups are compared with the HDB group.

different doses of Matrigel microbeads was further quantified using the live/dead staining. The result indicated that extremely high cell density, e.g., 3000–6000 cells-per-bead, may decrease cell viability (Fig. S11). This may resulte from the limited access of encapsulated MSCs to the cell culture medium and higher metabolic stress in the microbeads.

MSCs cell number and DNA content in the Matrigel HDG, LDB, and HDB groups were also examined, by counting the cells after dissociating them from Matrigel with collagenase I (2 μg/mL, 2 h, 37 °C) treatment. At the initial stage, cell numbers and DNA content were nearly identical across these groups. However, at the later stages (D7 and D14), the cell numbers and DNA content were significantly varied, presenting higher values in the Matrigel HDB-MSCs group compared with the other two groups. A 3-fold increase in live cell number was achieved in the Matrigel HDB group after 1 week, whereas only an 87.2% increase and 48.6% increase were achieved in the Matrigel LDB group and the Matrigel HDG group, respectively. A 12-fold increase of live cell number was achieved in the Matrigel HDB group after 2 weeks, significantly higher than the Matrigel LDB groups (440% increase) and the Matrigel HDG (360% increase) (Fig. 3d). A over 2-fold increase of DNA content was achieved in the Matrigel HDB group after 1 week, whereas only 84.6% increase and 109% increase were achieved in the Matrigel LDB group and the Matrigel HDG group, respectively. A 27-fold increase of DNA content was achieved in the Matrigel HDB group after 2 week, significantly higher than the Matrigel LDB group (670% increase) and the Matrigel HDG group (385% increase) (Fig. 3e).

MSCs metabolic activity was evaluated by 3-(4,5-dimethyl-2-thiazolyl)-2,5-diphenyl-2H-tetrazolium bromide (MTT) assay. The optical density (OD) values kept increasing in the 3 groups from day 3 to day 10, with significantly higher values in the Matrigel HDB group than in the parallel conditions (Figs. 3f and S12). The proliferating cells altered the bead morphology, by breaking down the bead scaffolds. We analyzed the aspect ratios of Matrigel LDB and HDB beads. The results showed that the bead ratio became different on day 7. The aspect ratios of HDB beads decreased from 0.91 (nearly spherical) on day 0 to 0.60 (highly deformed) on day 7, whereas the ratios of LDB beads remained above 0.85 on day 7 (Fig. 3g). The difference may be attributed to higher metabolic activity in HDB-MSCs (Fig. 3c).

MSCs proliferation was further explored by using 5-ethynyl-2′-deoxyuridine (EdU) labeling (Fig. 3h). From day 0 to day 7 in culture, the cell proliferation rates decreased among the three groups, but at both day 3 and day 7, the HDB group preserved more proliferating cells than the other groups (Fig. 3i). It must be noted that at day 0, the LDB-MSCs exhibited significantly fewer proliferating cells than the high-density groups, suggesting that cell density plays an important role in maintaining the activity of cell colonies. After 1 week, the HDB-MSCs preserved ~13% of cells as proliferative, whereas the level of proliferating HDG-MSCs and LDB-MSCs dropped to ~4% and ~0.2%, respectively (Fig. 3i). Next, MSCs migration in the Matrigel HDG, LDB and HDB groups using transwell plates (8 μm pore size) was examined. Both the MSCs colony morphology and the OD values at 570 nm demonstrated that more MSCs migrated through the transwell pores

in the Matrigel HDB group than in the HDG group. The LDB-MSCs exhibited the lowest migration ability (Fig. 3j, k). Next, two MSCs markers, CD90 and CD105, were used to investigate the retention of MSCs stemness under different conditions. Compared with 2D MSCs (source cells, positive control group, 98.5% CD90 and 96.4% CD105), both HDB-MSCs and HDG-MSCs maintained high expression of stemness markers (HDB, 99.6% CD90 and 97.3% CD105, HDG, 94.1% CD90 and 95.7% CD105). However, the LDB-MSCs exhibited a slight loss in stemness retention (87.2% CD90 and 80.9% CD105) (Fig. 3l).

Osteogenic-adipogenic differentiation was further performed to explore the stemness differences across these groups. Results showed that significantly higher rates of differentiation toward either the osteogenic lineage or the adipogenic lineage upon induction were possessed in the high-density conditions, including both the HDB-MSCs and the HDG-MSCs induced at as long as 3 weeks after normal culture (Figs. 3m, n and S13–16). Overall, the in vitro characterization implies that a low-density cell population may compromise cell proliferation, migration, and retention of stemness.

The extracellular scaffolding material also affects MSCs behaviors. We compared the MSCs growth activity in Matrigel beads, gelatin (5%, w/v) beads crosslinked by TG, and HAMA (5%, w/v). Gelatin is much stiffer than Matrigel[39,40]. The MSCs viability ratio in the gelatin HDB group (86.9 ± 1.4%) was significantly higher than the gelatin HDG (78.1 ± 1.7%) and the gelatin LDB (45.5 ± 10.6%) groups after 7 days in culture (Figs. S17a and S17b). The difference was also validated by the OD values (Fig. S17c). Furthermore, the proliferation rates of MSCs in the gelatin HDG, LDB, and HDB groups were examined. Similar to the Matrigel groups, the proliferating MSCs was significantly more in the gelatin HDB group than the other groups, characterized at day 0, day 3, and day 7 (Figs. S17d and S17e). These results indicate that HDB-MSCs outperform HDG-MSCs and LDB-MSCs, in both Matrigel and gelatin scaffolds. It further validated the finding that low cell density was detrimental to cell growth. However, when comparing the Matrigel HDB-MSCs with the gelatin HDB-MSCs, it was observed that Matrigel is substantially better than gelatin as the extracellular scaffold, evaluated by both cell viability and proliferation. 5% HAMA was crosslinked using 0.25% lithium phenyl-2,4,6-trimethylbenzoylphosphinate (LAP). Live/dead staining showed that encapsulated MSCs did not spread or migrate out from the microbeads (Fig. S18a). Viability rates of MSCs in HAMA microbeads and bulk gel decreased after 1-week culture, especially in the HAMA LDB group (Fig. S18b). A significantly lower proportion of EdU-positive cells appeared in the HAMA LDB group compared with other groups (Fig. S18c, d). Both the live/dead staining and EdU test suggested that low cell density was detrimental to cell growth, and HAMA had a low capacity to support cell growth compared with Matrigel. The metabolic activity of MSCs was also investigated. Cell metabolic activity in the HAMA LDB group was significantly lower on day 5 and day 7 when compared with the HDG and HDB groups (Fig. S18e). Again, the metabolic activity of MSCs in HAMA was dramatically reduced than in Matrigel, regardless of the gel configuration (Figs. S18e and 3c).

Apart from MSCs, we also studied other cell types to further verify the findings, including NIH3T3, a mouse fibroblast cell line, and SW620, a human colon cancer cell line. NIH3T3 in the Matrigel HDB group spread extensively after 1 week in culture. Cell number increased by nearly 3-fold, significantly higher than the proliferation rates observed in the Matrigel HDG and LDB groups. The migration ability of NIH3T3 cells was also increased in the HDB group than in the others (Fig. S19). For SW620, which is of epithelial origin, the cells exhibited limited migration ability and remained confined in the bead volumes. Less difference was observed across the three groups, except that HDB cells proliferated faster in the initial 3 days. The reduced proliferation from day 3 to day 7 might be attributed to the spatial constraint (Fig. S20).

## Matrigel HDB-MSCs have upregulated cell proliferation and cell migration through PI3K-Akt signaling and Wnt/β-CATENIN signaling

To explore the mechanism of MSCs augmentation under different encapsulation conditions, RNA-seq of MSCs cultured in the Matrigel HDG, LDB, and HDB groups for 7 days was performed. A total of 3414 differentially expressed genes (DEGs) were identified between the Matrigel HDG and Matrigel HDB groups (Fig. 4a). 1761 genes were up-regulated and 1653 genes down-regulated (Fig. 4b). Between the Matrigel LDB and Matrigel HDB group, 3303 DEGs were detected with 2875 up-regulated and 428 down-regulated genes (Fig. 4c). 1328 DEGs were overlapping among the HDB, HDG, and LDB groups. Notably, the 1328 DEGs classification heatmap differed HDB from HDG and LDB; ~77.5% of DEGs gained higher expression in HDB than in HDG and LDB (Fig. 4d). Kyoto Encyclopedia of Genes and Genomes (KEGG) analysis of the overlapped 1328 DEGs revealed that the PI3K-AKT signaling pathway substantially differed the Matrigel HDB group from the others (Fig. 4e, f). Within the PI3K-AKT signaling pathway, many secretory factors, including HGF, VEGFA, FGF7, and PDGFB promoting cell survival and proliferation, and receptors, including EPOR, OSMR, and TEK, were significantly enriched in the Matrigel HDB group (Fig. 4f). Expression of secreted genes was quantified using qPCR, including HGF, TBK, VRGFA, and EPOR. The expression of HGF, TBK, VRGFA, and EPOR was higher in the HDB group compared with the other two groups (Figure S21).

Among the 1975 non-overlapped DEGs, 89.6% of genes exhibited higher expression in HDB than in HDG and LDB (Fig. S22a). KEGG analysis revealed that metabolic pathways might be responsible for the differential MSCs behavior under different densities, as HDB-MSCs were distinguished from LDB-MSCs in the expression of 110 DEGs involved in metabolic pathways (Fig. S22b and S22c). Further, KEGG analysis of the 2086 non-overlapped DEGs revealed that the Wnt signaling pathway might be responsible for augmented MSCs abilities in Matrigel beads (HDB) and bulk gel (HDG) (Fig. S22d–f).

The expression and phosphorylation levels of AKT in the PI3K-AKT signaling pathway and GSK3β and β-CATENIN in the Wnt signaling pathway were examined. The Western blot results revealed that HDB increased the expression of AKT, along with the elevated levels of AKT phosphorylation at S473. Moreover, HDB exhibited increased levels of GSK3β phosphorylation at S9, implying that inactive GSK3β led to the increase of stable β-CATENIN (Figs. 4g and S23a). Next, the immunostaining for pAKT, AKT, and β-CATENIN was performed to further validate this finding. The intensity of pAKT, AKT, and β-CATENIN was significantly higher in the HDB group than in the HDG and LDB groups (Figs. 4h and S23b). Therefore, it is concluded that the HDB condition promotes MSCs migration and proliferation via the PI3K−AKT signaling and Wnt/β-CATENIN signaling pathway (Fig. 4i). Differential gene expression profiles related to cancer pathways were also analyzed among the Matrigel-based experimental groups. More genes were enriched in the HDB-MSCs group, which might be attributed to the augmented cell growth and regeneration (Fig. S24).

## Matrigel HDB-MSCs group has increased production of functional extracellular vesicles (EVs)

EVs are speculated to be equally important in MSCs-based therapy[41]. The capacity of MSCs production of EVs in the Matrigel HDG, LDB, and HDB groups was therefore examined. In all groups, MSCs with consistent initial cell numbers were first expanded in a complete medium for 5 days to reach a confluency of 70–80%. Then the medium was replaced by a complete medium containing EV-depleted fetal bovine serum (FBS) and collected after another 96 h. Transmission electron microscopy (TEM) and nanoparticle tracking analysis (NTA) on EVs isolated by ultracentrifugation were performed. HDG-EVs, LDB-EVs, and HDB-EVs exhibited similar morphology in TEM, appearing as round, lipid bi-layered vesicular structures (Figs. 5a–c and S25). Size distribution of the collected EVs was analyzed by NTA (Fig. 5a–c) and the mean sizes were calculated (Fig. 5d). More than 80% of the collected EVs were populated between 50 and 250 nm, which was consistent with the previous findings[41]. NTA also revealed that Matrigel HDB-MSCs had the highest production of EVs, reaching $2.6 \times 10^{10}$/mL, whereas the HDG-EVs and the LDB-EVs only reached $1.1 \times 10^{10}$/mL and $1.0 \times 10^{10}$/mL, respectively (Fig. 5e). The initial cell numbers across different conditions at EVs collection were identical ($9.0 \times 10^{5}$ cells/group).

Next, a mouse myoblast cell line, C2C12 cells, was used to examine the functionality of MSCs-EVs. HDG-EVs, LDB-EVs, and HDB-EVs. The cells were labeled by the membrane-binding fluorescent dye PKH67, and then supplemented into the C2C12 culture medium. After 48 h, the EVs were observed in strong colocalization with the C2C12 cells, labeled with the Dil cell membrane tracker (Figs. 5f, g and S26a–d). Moreover, EV treatment promoted C2C12 cell growth in an increasing dose-dependent manner (0, 2.5, and 5 µg/mL of HDB-EVs) (Fig. 5h, i). EdU labeling demonstrated more proliferating cells in the EVs (10 µg/mL) treated groups, compared with the non-treated group. However, no difference was observed among the EV treatment groups, because the EV doses were identical. It also implies that EV quality may not be as different as the quantity across different production groups (Figs. 5j, k and S26e). The Western blot results showed that the MSCs cultured in the Matrigel HDG, LDB, and HDB groups were all positive for EV-specific markers, including CD9, CD81, Hsp70, TSG101, Calnex, and CD63. The proteins were most abundant in the Matrigel HDB group (Fig. S26f). The therapeutic anti-inflammatory potential of EVs in the Matrigel HDG, LDB, and HDB groups was also analyzed, by comparing the gene expression of 3D groups with the 2D EVs that were proven to possess therapeutic anti-inflammatory effects[42–44]. No significant difference in gene expression was found across these groups. It suggested that Matrigel HDG, Matrigel LDB, and Matrigel HDB-EVs possessed the equivalent potential of therapeutic anti-inflammatory abilities as the standard EV products (Fig. S27).

## Function restoration in mouse volumetric muscle loss injury

Volumetric muscle loss injury was created by removing 60% of the gluteus maximus (GM) muscle mass in mice (Fig. S28). The Matrigel HDB-MSCs were sparsely printed into the defect for functional muscle regeneration. The empty space was filled with acellular beads (Fig. 6a). The LDB-MSCs were densely printed into the defect. The Matrigel HDG-MSCs were cast into the injury area (Fig. S29). Non-treated (defect only), acellular beads only, and MSCs suspension injection were used as the control groups. To mimic the printing process toward irregular tissue defects, we first created an irregular plastic mold, where the beads, dyed green, were printed and assembled. The recovered assembly was robust and replicated the mold concave (Fig. 6b). A bio-inert oil, HFE 7000 with an extremely low boiling point (b.p. 34 °C) was used in this study. It had minimal influence on cell growth[45] (Fig. S7). To further accelerate the evaporation of oil during surgical transplantation, the mice were placed under a heater for 2 min

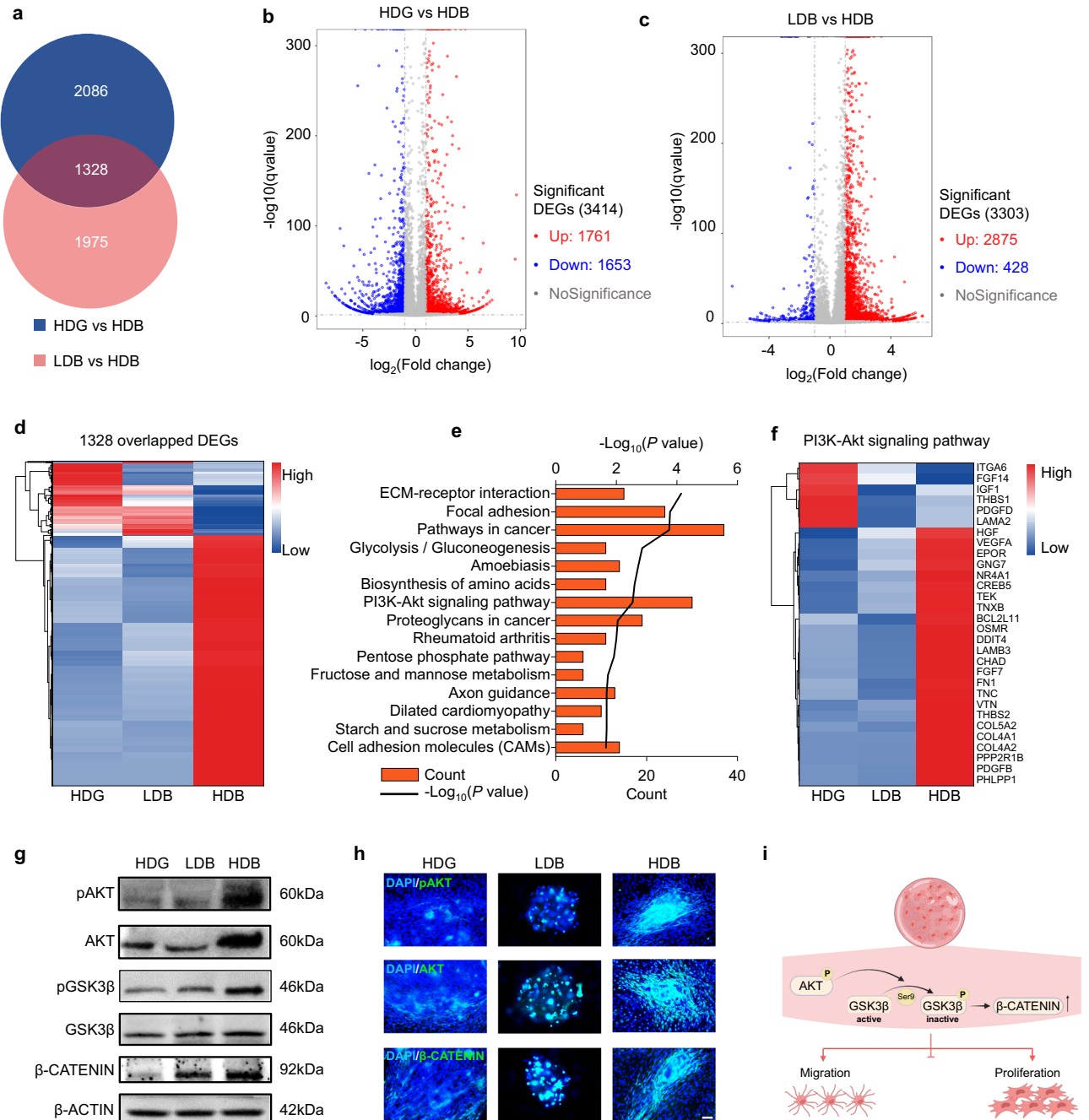

**Fig. 4 | Matrigel HDB group promotes MSCs proliferation and migration through the PI3K-Akt signaling pathway and Wnt/β-CATENIN signaling pathway. a** Venn diagram showing the differentially expressed genes (DEGs) in the Matrigel HDG group and the Matrigel LDB group, both compared to the Matrigel HDB group. **b** and **c** Volcano diagram of DEGs in the Matrigel HDG group (**b**) and the Matrigel LDB group (**c**), both compared to the Matrigel HDB group. **d** Heatmap view of the overlapped 1328 DEGs in the Matrigel HDG group and the Matrigel LDB group, both compared to the Matrigel HDB group. **e** KEGG pathway analysis of the 1328 DEGs. Significant differential genes were determined using a one-sided wald test in the DESeq2 package. **f** Heatmap view of 30 DEGs belonging to the PI3K-Akt pathway. **g** Western blotting of PI3K–Akt signaling pathway proteins, pAKT and AKT, and their downstream proteins, including pGSK3β, GSK3β, β-CATENIN. Samples were derived from the same experiment and the gels were processed in parallel. **h** Immunofluorescence for pAKT, AKT, and β-CATENIN. Scale bar, 200 μm. **i** Schematic elaboration of the PI3K–AKT and Wnt/β-CATENIN signaling pathways promoting MSCs migration and proliferation.

after surgery (Fig. S30). After 4 weeks, the non-treated (defect only), acellular beads only, and MSCs suspension injection groups gained limited recovery as the injury was still visible (Figs. 6c and S31a). Transplantation of the Matrigel HDG-MSCs, LDB-MSCs, and HDB-MSCs all achieved significant recovery, as the damaged area became indistinguishable from the surrounding native muscles (Fig. 6c). However, they were significantly different in the microscale. Hematoxylin and eosin (H&E) and Masson's trichrome staining (MTS) were employed to

observe the microscopic structures of retrieved muscles. H&E staining indicates myofibers and MTS stains collagen and indicates fibrosis. The Matrigel HDB-MSCs group regenerated the most abundant skeletal muscle featured with high-density aligned muscle fibers at week 4 (Figs. 6d and S32). Moreover, only a small fraction of collagen, indicative of fibrosis, was distributed amongst the regenerated muscle fibers in the Matrigel HDB-MSC group. All other conditions generated fewer muscle fibers but heavier fibrosis (Fig. 6e).

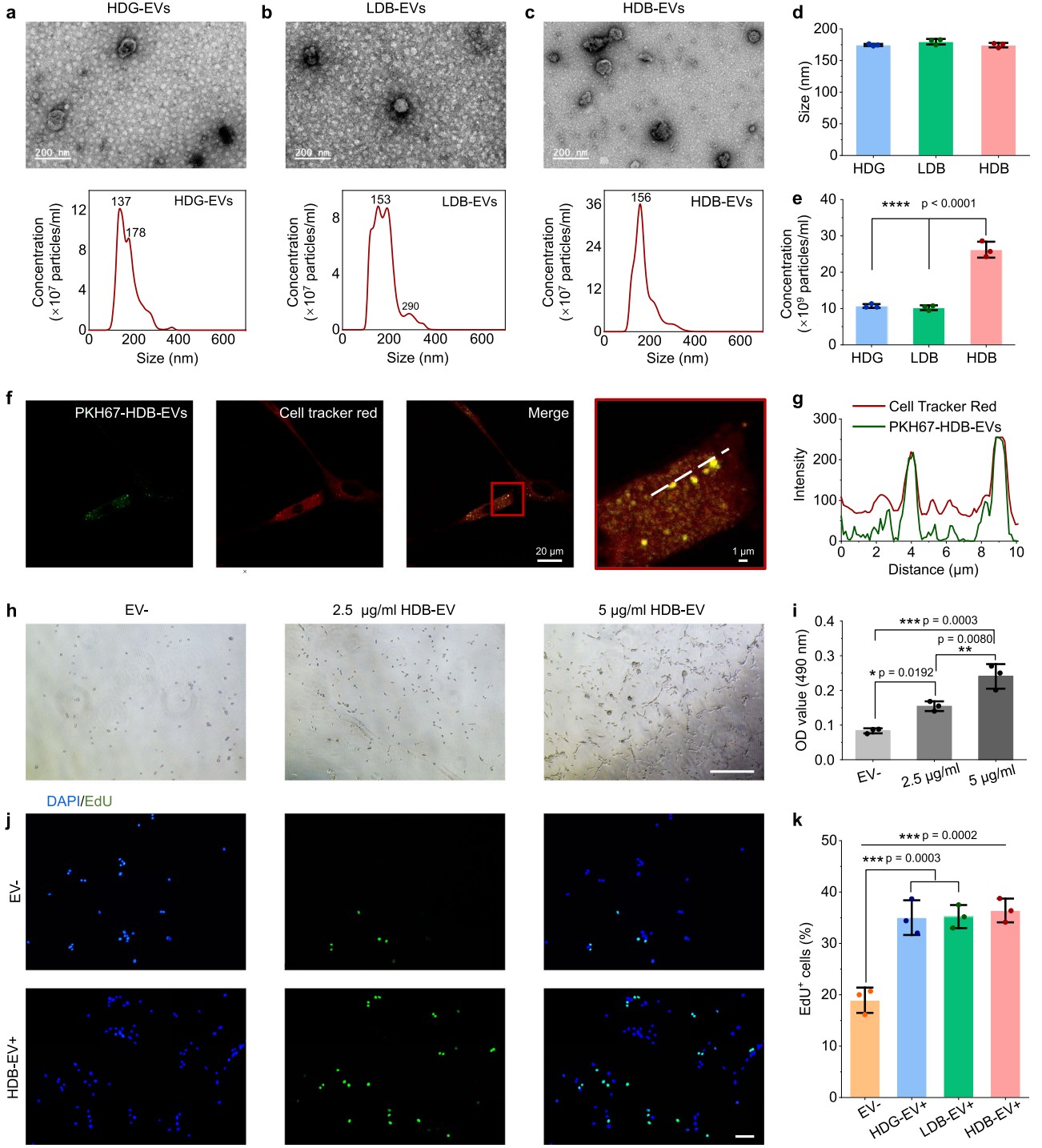

**Fig. 5 | Matrigel HDB group augments MSCs production of extracellular vesicles (EVs). a–c** Transmission electron microscopy (TEM) images and nanoparticle tracking analysis (NTA) of HDG-EVs (**a**), LDB-EVs (**b**), and HDB-EVs (**c**). Scale bars, 200 nm. **d** and **e** The size (mean diameter) (**d**) and particle concentration (**e**) of HDG-EVs, LDB-EVs and HDB-EVs. $n = 3$. **f** Colocalization between HDB-EVs and C2C12 cells. PKH67-labeled HDB-EVs were incubated with C2C12 for 48 h. Cells were stained with Dil cell membrane tracker. **g** Intensity profiles of two fluorescent channels (green, EVs; red, cell membrane) along the dashed line. **h** and

**i** Representative bright-field images (**h**) and OD values of MTT assays for cell proliferation (**i**) of C2C12 cells without EV treatment (EV⁻) and with increased doses of HDB-EVs treatment for 48 h. $n = 3$. Scale bar, 500 μm. **j** and **k** Representative images of EdU staining (**j**) and quantification of EdU-positive C2C12 cells (**k**) without EV treatment (EV⁻) and with 10 μg/mL HDG-EVs, LDB-EVs and HDB-EVs treatment for 48 h. $n = 3$. Scale bar, 100 μm. **d**, **e**, **i**, **k** The data are represented as mean ± SD. The significant difference is determined by one-way ANOVA, followed by Tukey's test. *$p < 0.05$; **$p < 0.01$; ***$p < 0.001$; ****$p < 0.0001$.

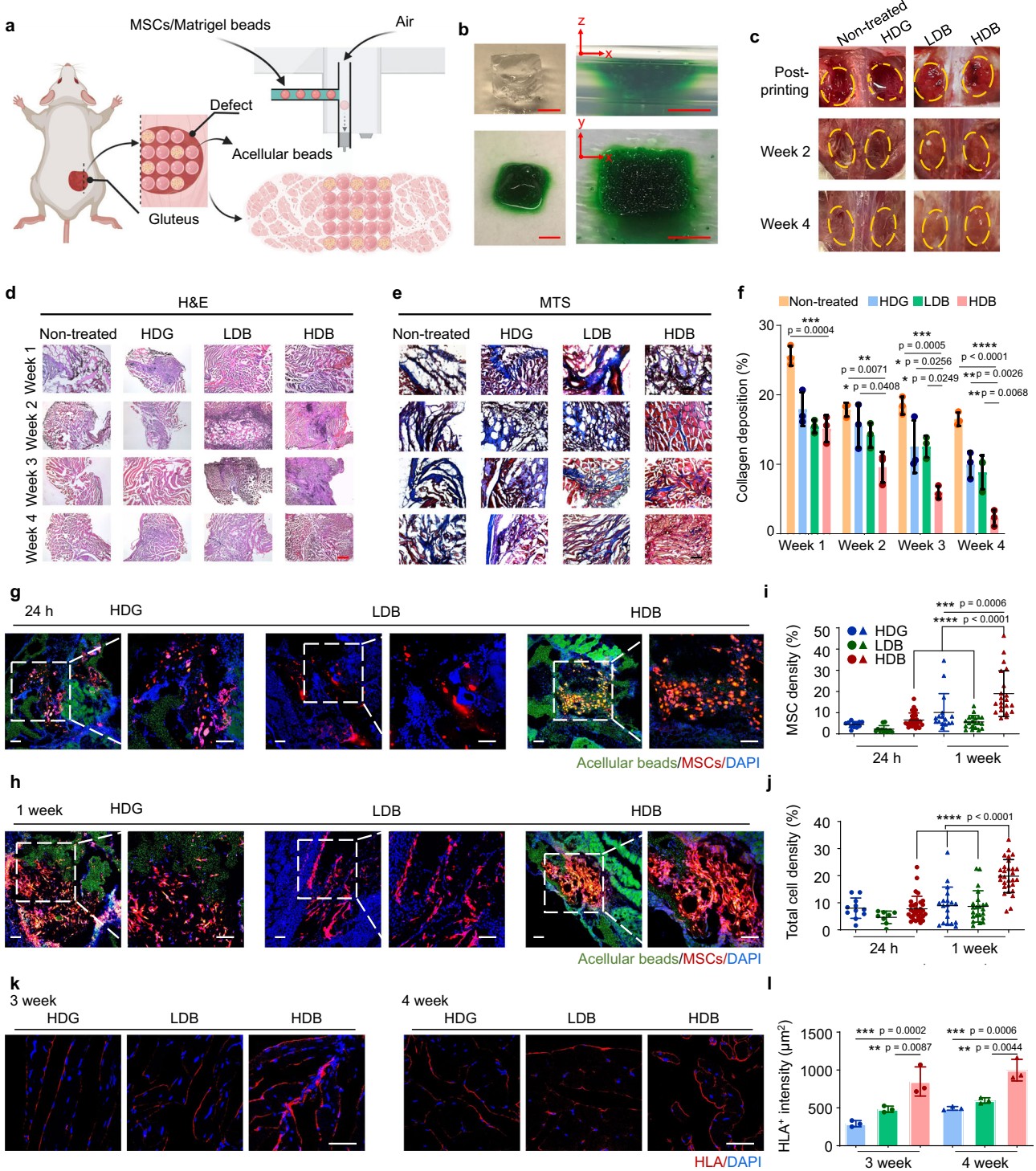

The collagen deposition was further quantified. The fibrosis collagen abundance was the least in the Matrigel HDB-MSCs group (Fig. 6f). Among these groups, transplantation of acellular beads and MSCs suspension injection resulted in heavier fibrosis and less muscle regeneration (Fig. S31b), implying that MSCs were the augmented factor in injury repair when transplanted together with gel beads.

Next, retention, growth, and distribution of MSCs in regenerated muscle tissue were further investigated after transplantation for 1 day, 1, 3, and 4 weeks. MSCs were labeled using the red Dil cell membrane tracker (Meilunbio). Acellular beads and acellular bulk gel were loaded with monodispersed green fluorescent polystyrene microspheres (1 μm, 1:20 diluted in Matrigel solution). At 24 h after transplantation,

the HDB-MSCs remained as integral clusters (Fig. 6g), but displayed extensively spread distribution at 1-week post-transplantation (Fig. 6h). Significantly higher proportion of MSCs was observed in the HDB groups than in the LDB and HDG groups (Fig. 6i). The in vivo result demonstrated that the cell-laden beads (red) were distributed among acellular beads (green) after 1-week transplantation (Fig. S33). Further, MSCs in the HDB group may have gained significantly higher proliferation rates, suggested by the higher density and thickness of MSCs fluorescence at 1-week post-transplantation. The total cell density including endogenous cells and transplanted MSCs in the regenerated muscle tissues were analyzed, which was significantly higher in the HDB group than in other groups (Fig. 6j). This may suggest both

**Fig. 6 | Matrigel HDB-MSCs augment skeletal muscle regeneration in mice volumetric muscle loss (VML) located in mice gluteus maximus (GM).**
**a** Schematic illustration of Matrigel HDB-MSCs transplantation using bead-jet printing. VML was modeled in mice GM muscle. The MSCs-laden HDB beads were printed in sparse patterns to widen the coverage of HDB distribution in the defect. The gaps were filled with acellular beads. **b** Bead-jet printing and ordered assembly in an irregular mold that mimics the irregular in vivo defect. **c** Photographs of VML trauma of immediately post-printing of MSCs-laden LDB and HDB beads (or post-casting of MSCs-laden HDG) and after regeneration of muscle tissues for 2 weeks and 4 weeks in the defect. **d**–**f** H&E (**d**) and MTS (**e**) staining of regenerated skeletal muscle tissues at 4 weeks post-printing. H&E: hematoxylin and eosin. MTS: Masson's trichome staining. Scale bar, 500 μm (**d**), 100 μm (**e**). Quantification of collagen deposition percentage according to the MTS staining results (**f**). n = 3. For

each data, at least six imaging windows are randomly selected. **g**–**j** Representative images of MSCs in the Matrigel HDG, LDB, and HDB groups after transplantation for 24 h (**g**) and 1 week (**h**) in muscle tissues. MSCs were labeled using the red Dil cell membrane tracker (Meilunbio). Acellular beads and acellular bulk gel were loaded with monodispersed green fluorescent polystyrene microspheres (Aladdin, 1 μm, 1:20 diluted in Matrigel solution). Quantification of MSCs (**i**) and total cell density (**j**) per section after transplantation for 24 h and 1 week. n = 3. Scale bar, 100 μm. **k** and **l** Immunofluorescence for human leukocyte antigen (HLA) to quantify the MSCs retention rates at 3 weeks and 4 weeks post-printing (or post-casting) of MSC complexes (**k**). Quantification of HLA⁺ area per section at 3 weeks and 4 weeks (**l**). n = 3 per group. Scale bar, 50 μm. All data are represented as mean ± SD. The significant difference is determined by one-way ANOVA, followed by Tukey's test. *p < 0.05; **p < 0.01; ***p < 0.001; ****p < 0.0001.

MSCs and other endogenous cells were important for muscle regeneration in the initial stage.

Long-term residence of MSCs was also investigated using immunostaining of human leukocyte antigen (HLA) in the regenerated tissues at week 3 and week 4. At 3–4 weeks post-transplantation, the distribution of HLA⁺ signals changed to myofiber-like features, implying that the MSCs had differentiated (Figs. 6k and S34, 35). Likewise, the HLA⁺ intensity was the highest in the Matrigel HDB-MSCs transplantation group (Fig. 6l). Abundant HLA and desmin signals were observed at a large scale in the HDB-MSCs treated tissues, displaying relatively uniform profiles over 3-mm across (Fig. S35).

## Histology characterization of regenerated muscles

Newly formed myofibers, muscle resident cells, vascularization, neural integration, and immunomodulatory effects were explored in the retrieved skeletal muscle tissues at 4 weeks post-transplantation using immunofluorescent analyses. To investigate the de novo muscle fiber formation and maturation, the regenerated tissues were stained for desmin and myosin heavy chain 7 (MYH7), markers for new myofibers. The highest proportion of neo-muscle fibers was observed in the group treated by Matrigel HDB-MSCs, whereas only sparse fibers were detected in the non-treated group. The myofiber area stained using desmin in the Matrigel HDB-MSCs group reached nearly 60% coverage, which was significantly higher than the Matrigel HDG-MSCs group (43.1%), the Matrigel LDB-MSCs group (39.6%) and the non-treated group (32.2%) (Fig. 7a). New myofibers area stained by MYH7 in the Matrigel HDB-MSCs group reached nearly 25% coverage, which was significantly higher than the coverage rates in the Matrigel HDG-MSCs group (9.1%), the Matrigel LDB-MSCs group (6.8%) and the non-treated group (3.4%) (Fig. 7b). Muscle resident cell changes were also investigated by using PAX7, a typical marker for muscle resident cell, and compared across groups. The expression of PAX7 was significantly higher in the HDB-MSCs group than in other groups (Fig. 7c). This may indicate that muscle resident cells gained higher invasion ability in the transplantation of high-density MSCs by bead-encapsulation. Matrigel HDB-MSCs treatment also promoted angiogenesis in the regenerated tissue, as judged from the extent of positive staining of CD31, the endothelial marker (Fig. 7d). Neo-vascularization was further confirmed by immunostaining of α-smooth muscle actin (α-SMA), a marker of functional arterioles[46]. The density of α-SMA⁺ blood vessels was markedly higher in the Matrigel HDB-MSCs group than in the other groups (Figs. 7e and S31c). The retrieved muscle tissues were also immunostained for neurofilament (NF) and βIIIT to investigate the extent of innervation. The most significant high-density distribution of NF and βIIIT were detected in the Matrigel HDG-MSCs group (Fig. 7f, g). MSCs immunomodulatory effects were also explored by using CD68 to stain M1 macrophage and CD3 to stain T-cells. Compared with the non-treat group, the expression levels of CD68 and CD3 were both decreased in the treatment groups in regenerated muscle tissues,

with the HDB-MSCs transplantation possessing the highest reduction (Fig. 7h, i). It verified that MSCs moderated or suppressed the inflammatory activity.

Next, the gene expression profiles of retrieved muscle tissues using RNA-seq analysis were determined. KEGG and GO biological process, cellular component, and molecular function analysis on the sequencing data of regenerated muscle tissues were performed. GO analysis showed that highly differentiated genes were enriched in terms related to an extracellular matrix organization, smooth muscle cell proliferation, axonogenesis, immune response, muscle cell differentiation, extracellular matrix assembly, and skeletal muscle organ development (Figs. S36 and S37). The analysis of highly differentiated genes reached an agreement with the results from the IHC examination (Fig. 7). Specific genes related to the malignant transformation of cancer genes in regenerated muscle tissue were also analyzed. Compared with the normal group, i.e., the healthy tissue, the gene expressions were not showing significant differences among the experimental and control groups (Fig. S38). Thus, no malignant transformation occurred in the regenerated muscle tissues. Specific genes related to MSCs secretome were also analyzed. The gene expressions across different groups showed no significant difference in the regenerated muscle tissues (Fig. S39).

## Matrigel HDB-MSCs promote hair follicle generation in skin wound healing

Different scales of skin wounds, including 5 mm × 5 mm (~2% removal of dorsal skin), 10 mm × 10 mm (~15% removal of dorsal skin) wounds, and 15 mm × 15 mm (~35% removal of dorsal skin) wounds, were created and repaired using our bead-jet printing platform. The skin wounds were treated using Matrigel HDG-MSCs, Matrigel LDB-MSCs, and Matrigel HDB-MSCs (Fig. S40 and Video S6). Non-treated (defect only), treatment by acellular beads only, and MSCs suspension injection were used as the control groups. Previous studies reported that small skin wounds (smaller than 1 × 1 cm) are more difficult to regenerate hair follicles and tend to form scars when compared with larger wounds, where wound-induced hair follicle neogenesis occurs[3,47,48]. Thus, the regeneration of over 5-mm-diameter excisional skin was mainly explored. In the control group, the self-healed wounds displayed a thin dermis with horizontally oriented collagen fibers, a flattened epidermis and were devoid of hair follicles and sebaceous glands (Figs. 8a–d, S41–43). Transplantation of cast or printed Matrigel HDG-MSCs, LDB-MSCs, and HDB-MSCs enabled the generation of hair follicles and sebaceous glands in the healed wounds. A significantly thicker dermis and the highest abundance of hair follicles were observed in the Matrigel HDB-MSCs group (Fig. 8a–d). Wound closure area calculation revealed the Matrigel HDB-MSCs accelerated healing when compared with the control groups (Fig. S40).

Retention rates of MSCs in regenerated skin tissues were investigated after transplantation for 1 week. MSCs were labeled using the red Dil cell membrane tracker. At only after 1-week post-transplantation, hair-follicle-like structures were observed in the Matrigel HDB-MSCs

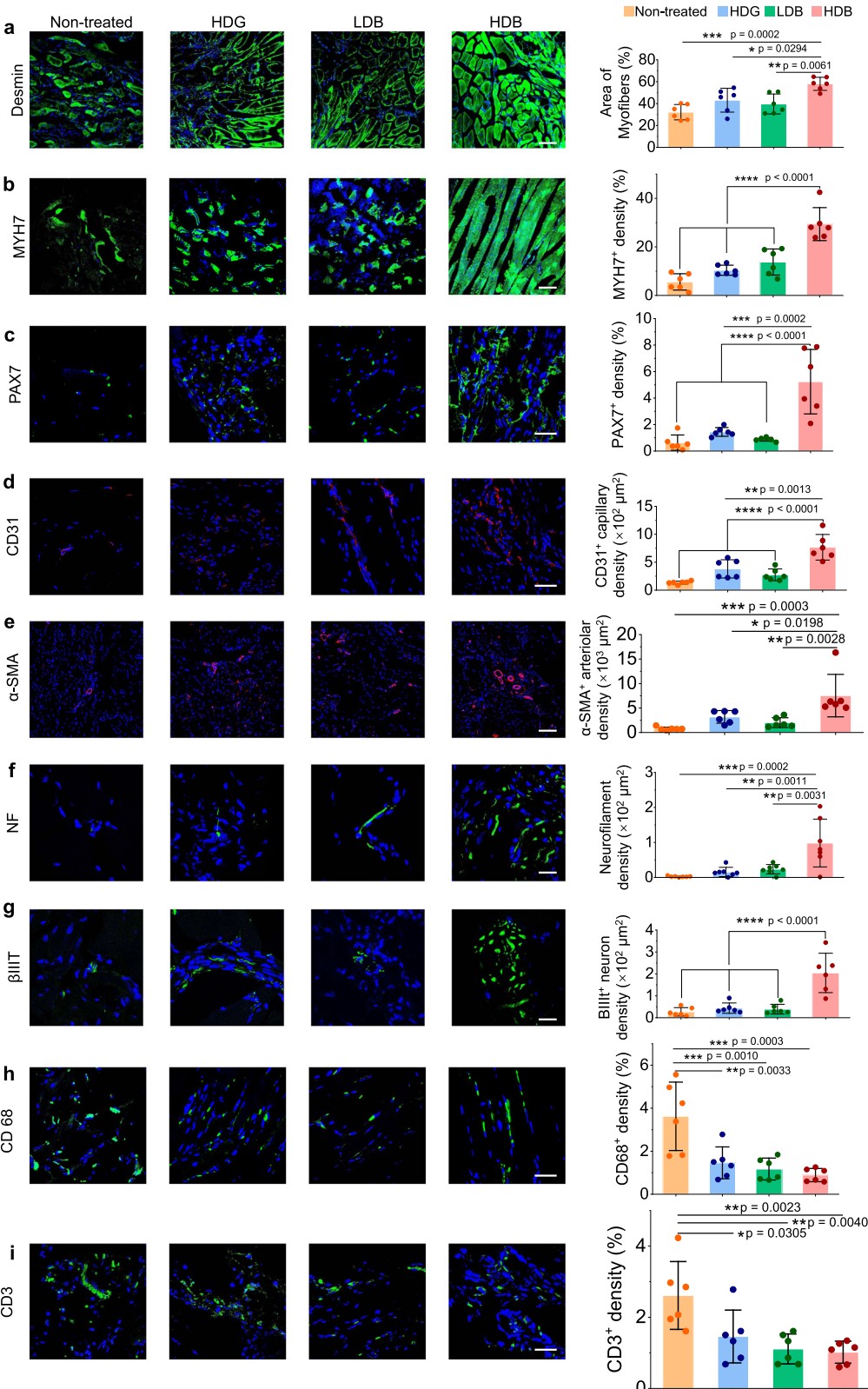

group, but not in other groups (Fig. 8e). However, there was no significant difference in MSCs density (Fig. 8f) and the total cell density that included both the transplanted MSCs and the endogenous cells (Fig. 8g) between the HDB-MSCs group and other groups.

The healed tissues were further examined for Ki67[+] cells, a marker of proliferating cells (Fig. 8h). 13.8% Ki67[+] cells were found in the treatment using Matrigel HDB-MSCs, which was markedly higher than

using Matrigel HDG-MSCs (4.5%), Matrigel LDB-MSCs (6.5%) and the control group (2.0%) (Fig. 8i). Next, angiogenesis was examined in the retrieved skin tissues. The capillary density was distinctively different among the four groups (Fig. 8j). The self-healing group exhibited a minimal amount of CD31[+] cells, whereas the HDB-MSCs treated tissue were distributed with abundant CD31[+] capillaries (Fig. 8k). These results indicated that the printed Matrigel HDB-MSCs accelerated the

**Fig. 7 | Investigation of myofiber maturation, muscle resident cells, angiogenesis, neural integration, and MSCs immunomodulatory effects in regenerated muscle tissues. a, b** Immunofluorescence for desmin (**a**) and MYH7 (**b**) to determine the myofiber maturation and their quantification by fluorescence area. **c** Immunofluorescence for PAX7 to determine the population of muscle resident cells and their quantification by fluorescence area. **d** and **e** Immunofluorescence for CD31 (**d**) and α-SMA (**e**) and to evaluate angiogenesis. α-SMA positive arterioles and CD31 positive capillaries are quantified by their fluorescence area. **f** and **g** Immunofluorescence for neurofilament (NF) (**f**) and βIIIT (**g**) to observe the neural integration. Neurofilament amount and βIIIT magnitude are quantified by their fluorescence area. **h-i**, Immunofluorescence for CD68 (macrophage) (**h**) and CD3 (T-cells) (**i**) to explore the MSCs immunomodulatory effects in muscle regeneration. The signal intensity of CD68 and CD3 were quantified by their fluorescence area. Scale bar, 100 μm (**a**, **b**, **e**), 40 μm (**c**, **d**, **h**, **i**), 20 μm (**f**, **g**). All data are represented as mean ± SD. $n = 6$ per group at each time point. For each data, at least six imaging windows are randomly selected. Significant difference is determined by one-way ANOVA, followed by Tukey's test. $*p < 0.05$; $**p < 0.01$; $***p < 0.001$; $****p < 0.0001$.

formation of hair follicles and associated sebaceous glands, promoted the proliferation of keratinocytes and augmented de novo angiogenesis. MSCs immunomodulatory effects in skin tissues were also explored by using CD68 and CD3 staining. Compared with the non-treat group, the expression of CD68 and CD3 was decreased in the treatment groups in regenerated skin tissues (Fig. S44). It verified that MSCs had moderated or suppressed the inflammatory activity. It was consistent with the finding in the regenerated muscle tissues.

For large-scale (10 mm × 10 mm and 15 mm × 15 mm) skin wound healing, significantly higher amounts of the hair follicles and sebaceous glands were observed in the Matrigel HDB-MSCs group than in the acellular group (Figs. S45 and S46). Gene expression profiling of retrieved skin tissues using RNA-seq analysis was performed to investigate the mechanism of HDB-MSCs in accelerating hair follicle formation. The KEGG and GO biological process, cellular component, and molecular function were analyzed on the sequencing data of regenerated skin tissues. GO analysis showed that highly differentiated genes were enriched in terms related to skin development, skin morphogenesis, wound healing, endothelial cell migration, sprouting angiogenesis, extracellular matrix, hair follicle cell proliferation, and keratinocyte migration (Figures S47 and S48). The analysis of highly differentiated genes reached an agreement with the results from the IHC examination (Fig. 8). Specific genes related MSCs secretome were also analyzed in the regenerated skin tissues. The gene expression of Vcam1, Ccl5, Il12b, Cxcl10, and Cxcl9, etc. were higher in the Matrigel HDB-MSCs group, compared with the other groups (Fig. S49).

## Discussion

This study aims to augment MSCs-based regenerative therapy, which, although has been proven effective in both laboratory and clinical trials, faces challenges in approaching the maximum capacity of transplantable MSCs[13]. Traditionally, the transplantation of MSCs is expensive and of low efficacy due to a dramatic reduction in cell retention after transplantation. Moreover, it remains unresolved how to achieve the full capacity of transplanted cells. In this study, we designed a bead-jet printing system that sparsely patterns high-density cell-laden beads. The regional (intra-bead) high cell density expands cell capacity, and meanwhile, sparse patterning enlarges the transplantation coverage. Therefore, it provides a solution for large-scale injury transplantation with a definite amount of available MSCs. In this study, the bead-jet printing platform accelerated the regeneration of skeletal muscle tissues with reduced fibrosis in mice VML injuries (Figs. 6 and 7), and thickened dermis with increased hair follicles in skin wounds (Fig. 8). The improvements were remarkable when compared with parallel MSCs transplantation conditions, including homogeneous MSCs-laden bead and bulk gel distribution, acellular bead support and cell suspension injection. In the sparse-dense MSCs bead transplantation, significantly elevated MSCs retention rate, MSCs proliferation and differentiation, angiogenesis, and innervation were obtained (Fig. 7).

Precise deposition of microgels in scaled-up production poses a big challenge in this field. Though high-throughput production of microgels has been achieved, the efficiency and throughput of traditional microbead printing are rather low, disabling to generate large

and heterogeneous microtissue assembly[22,23,38]. Bead-jet printing platforms could fabricate large-scale, planar, or spatial, structures in defined homogeneous and heterogeneous organizations. For planar structures, the current in-house bead-jet printing machine can print up to 100 cm² of a planar structure or 2 cm × 2 cm × 3 cm cubic structures (Figs. S4 and S5). Bead-jet printing also broadens the applicability of intraoperative bioprinting. Compared with conventional 3D in-situ printing techniques, including intravital 3D bioprinting[49], directly printing[50], droplet-based bioprinting[51] and in situ 3D bioprinting with bioconcrete bioink[52], bead-jet printing is specialized for stem cell therapy with improved loading efficiency and retention rate of stem cells in vivo. Sparsely patterning of high-density MSCs-laden beads among acellular beads is able to enlarge the transplantation coverage, thus enabling therapy of large-scale traumatic injury with a limited number of MSCs.

Though some prevalent biomaterials were proven their suitability in the bead-jet printing system, including Gelatin, GelMA, HAMA, etc., they failed to present analogous performance in supporting MSCs growth, proliferation, and differentiation as in Matrigel (Figs. 3, S17 and S18). For example, cell viability and metabolic activity were dramatically reduced in HAMA, compared with MSCs in Matrigel products. However, though the cytocompatibility of Matrigel is favorable, the clinical promise is limited due to its tumor-derived, ill-defined, variable composition and low availability for lab-used conditions[53,54]. The potential for antigenicity is one of the inherent limitations. Recent studies about synthetic hydrogels may offer a potential alternative to natural biomaterials, as biochemical and biophysical properties can be modularly altered, and engineered materials do not suffer the risk of antigenicity or other inherent biological properties. For example, Han et al. engineered a synthetic hydrogel-based matrix and applied the matrix for the regeneration of mice's skeletal muscle[55]. Therefore, for future clinical translation of this strategy, synthetic biomaterials with high biocompatibility, cell-ligation ability, biodegradability, and appropriate viscoelastic and apathogenic properties should be innovated and commercialized to provide appropriate alternatives to Matrigel[56].

There is no consensus on the specific mechanism of skeletal muscle regeneration in VML by transplanted MSCs. By tracking the retention, growth, and distribution of MSCs in regenerated muscle tissues, suggests that the transplanted MSCs may first proliferate within 1 week in the initial stage, followed by differentiation from the 3rd week and onward in the later stage (Fig. 7). In the initial stage, the proliferative MSCs and endogenous cells, e.g., muscle resident cells, might both contribute to the neo-tissue growth (Figs. 6 and 7). In the later stage, the Matrigel HDB-MSCs with high cell retention differentiate toward skeletal myofibers and accelerate muscle regeneration. In regenerated skin tissues, transplanted MSCs may differentiate into hair-follicle-like structures within 1 week after transplantation. MSC secretome function, e.g., by producing EVs in vivo, could also stimulate regeneration (Figs. 5 and S49). The presence of Matrigel HDB-MSCs in the early stage may initiate the EV-dependent paracrine signaling, which promotes the regeneration process from native tissues, including promoting cell proliferation and resolving inflammation[41,57]. Therefore, we tend to believe that both MSCs themselves, EVs and

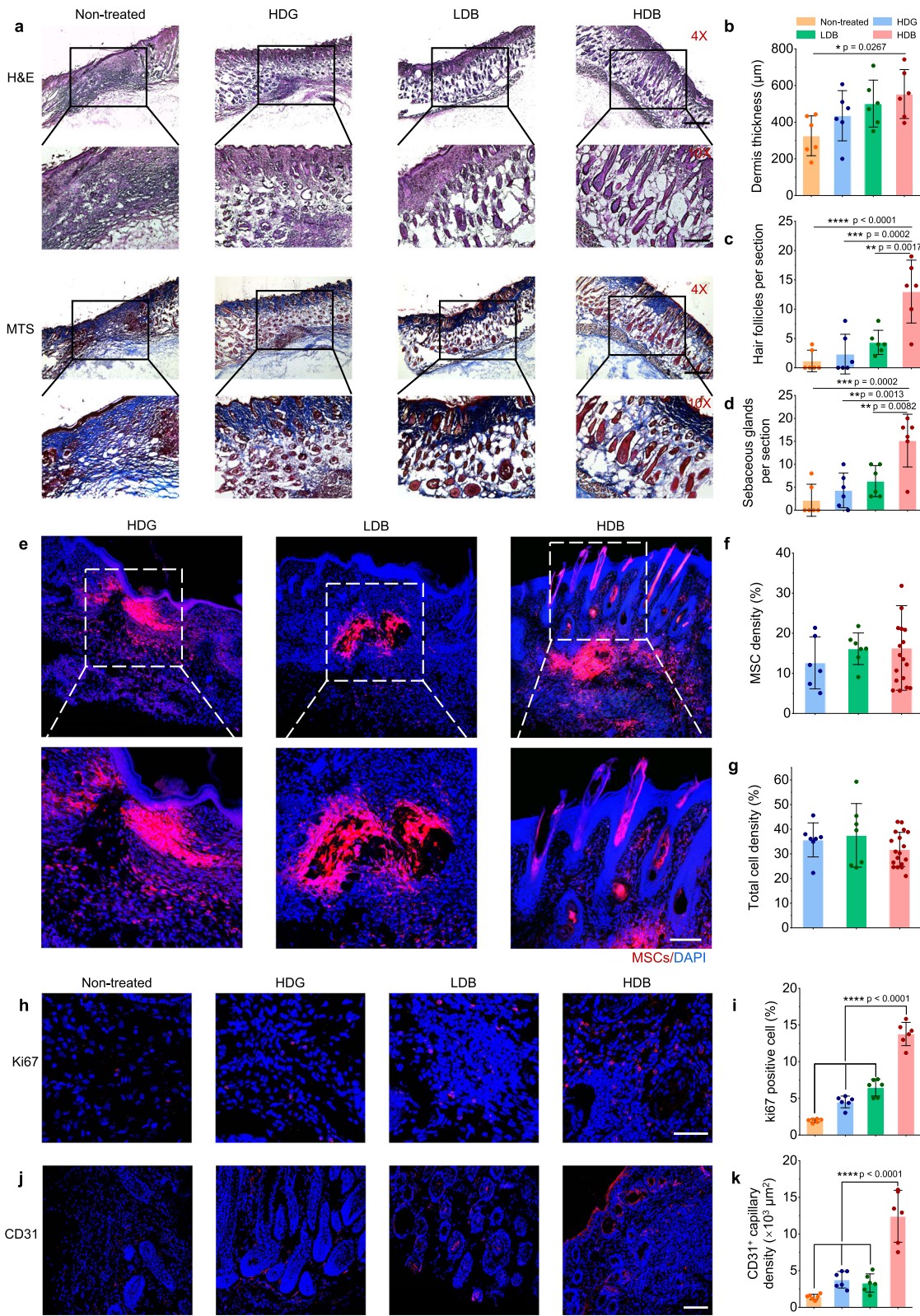

endogenous cells promote skeletal muscle regeneration and, likewise, skin regeneration. However, it remains challenging to distinguish their respective contributions.

It is also reported that the regenerative potential of skeletal muscle is maintained by the heterogeneous population of muscle-resident stem cells (MuSC)[58]. Compared with MSCs, MuSCs suffer from quick cell senescence and a rapid decrease resulting from the shortening of telomeres in each cell cycle[59]. Also, MuSCs after activation

need strong paracrine support from their niche, without which they will not be able to survive[60]. Thus, MSCs therapy has unique advantages in muscle repair and other tissue regeneration, including but not limited to self-renewal ability, multipotentiality, and immunomodulation effects.

In conclusion, we presented a new bead-jet printing system that enables scale-up production and manipulation of cell-laden Matrigel beads. The successes in the demonstration of VML repair and skin

**Fig. 8 | Matrigel HDB-MSCs augment skin wound healing and hair follicle regeneration. a** Representative images of H&E staining and MTS of regenerated skin tissues at 3 weeks post-printing (or post-casting) of MSCs loaded in Matrigel HDG, LDB beads, and HDB beads, respectively. For the HDB group, the MSCs-laden HDB beads were printed in sparse patterns, and the empty space was filled with acellular Matrigel beads. Scale bars: 500 μm (4×, upper row), 200 μm (10×, lower row). **b–d** Quantification of dermis thickness (**b**) and hair follicle (**c**) and sebaceous glands (**d**) counts per section in the regenerated area. $n = 6$. Six sections are selected. **e–g** Representative images of MSCs in the Matrigel HDG, LDB, and HDB groups after transplantation for 1 week (**e**) and quantification of MSCs intensity (**f**) and total cell density (**g**) per section in skin tissues. MSCs were labeled using the red Dil cell membrane tracker (Meilunbio). $n = 3$. Scale bar, 100 μm. **h, i** Immunostaining for Ki67 to view the proliferating cells (**h**) and quantification of Ki67 positive cells per section (**i**). Scale bar, 50 μm. $n = 6$. **j, k**, Immunofluorescence for CD31 to view angiogenesis (**j**) and quantification of CD31 positive capillaries (**k**). Scale bar, 100 μm. $n = 6$. For each data, at least six imaging windows are randomly selected. All data are expressed as mean ± SD. Significant difference is determined by one-way ANOVA, followed by Tukey's test. *$p < 0.05$; **$p < 0.01$; ***$p < 0.001$; ****$p < 0.0001$.

wound healing suggest that the strategy of sparse-dense distribution of MSCs, and perhaps other stem cells, might be extended to a wide range of regenerative therapies. Additionally, the current printing system requires further improvement to increase the feasibility of intraoperative bioprinting, by designing a synergistic system combined with an injury scanning system and a positioning control system for microgels. Eventually, the bead-jet printing system might be applied in versatile clinical and laboratory operations in regenerative medicine.

## Methods

### Ethical regulations
All the animal procedures were approved by the Animal Experimentation Ethics Committee at Tsinghua Shenzhen International Graduate School, Tsinghua University [Project 2020/19]. All the acquisitions of human samples were approved by the Ethic Committees of the Shenzhen Maternity & Child Healthcare Hospital, Project SFYLS[2020]061. Two donors (ref number 10011399 and 10016808) at Shenzhen Maternity and Child Healthcare Hospital volunteered to give the human umbilical cords for this study with informed consent.

### Study design
Three types of MSCs behavior and regeneration efficacy were tested in vitro and in vivo. The MSCs were loaded in high cell density ($1.0 \times 10^7$ cells per mL) beads (HDB), low cell density ($2.0 \times 10^6$ cells per mL) beads (LDB), high cell density ($1.0 \times 10^7$ cells per mL) bulk gel (HDG) with the initial cell number consistent across the parallel groups. When culturing in 12-well plates in vitro, 50 HDBs, 250 LDBs, 7.5 μl HDG were printed or cast in each well individually. When transplanted in vivo, balb/c mice (25-30 g) received the same amount of MSCs in the traumatic injury.

### Cell culture
Human umbilical cord tissues were obtained from full-term births after normal vaginal delivery at the delivery room of the Shenzhen Maternity and Child Healthcare Hospital as approved by the ethics committee. The tissues (length >10 cm) were collected in a sterile jar containing PBS supplemented with penicillin (100 U mL⁻¹) (Gibco) and streptomycin (100 μg mL⁻¹) (Gibco). The samples were transported on ice and processed within 12 h. Blood clots were removed by rinsing with PBS supplemented with penicillin (100 U mL⁻¹) and streptomycin (100 μg mL⁻¹), the cords were cut into pieces (2–3 cm), and blood vessels were removed. The Wharton's jelly was chopped into fragments (1 mm³) using scissors and placed into cell culture flasks (25 cm²) for culture expansion at 37 °C and 5% CO₂. After isolation, the MSCs were cultured in a medium comprising Dulbecco's modified Eagle's medium (DMEM)/F12 (Gibco, China) supplemented with 10% fetal bovine serum (FBS) (Gibco, USA) and 1% penicillin/streptomycin (P/S). When cells had reached 80% confluence, the umbilical cord tissues were removed and cells were transferred into culture flasks (75 cm²). In this study, MSCs between passages 3 and 10 (P3–P10) were used. Cells were verified in vitro for the MSC markers CD90 and CD105, as suggested by the International Society for Cell Therapies (ISCT) (Fig. 3i). C2C12 mouse muscle cells (GDC0175), and NIH3T3 mouse embryonic

fibroblast cells (GDC0030) were purchased from China Center for Type Culture Collection and used in P5 to P20. SW620 human colon cancer cells (CL-0225) were purchased from Procell Life Science Technology Co. Ltd. NIH3T3 mouse embryonic fibroblast cells, C2C12 mouse muscle cells, and SW620 human colon cancer cells culture media composed of DMEM (Gibco, China) supplemented with 10% FBS and 1% penicillin-streptomycin. All cells were maintained at 37 °C in a humidified CO₂ incubator. Cell culture media was refreshed every 2 to 3 days.

### Bead-jet printing
A customized bead-jet printing platform was designed in this study. This platform comprises a bead formulation module and a bead printing (jetting) module. The two modules operate in synchronization. In the formulation module, Matrigel (Corning, USA), gelatin (Sigma, USA), Gelatin Methacryloyl (GelMA) (EFL Inc., China), and Hyaluronic acid Methacryloyl (HAMA) (EFL Inc., China) were used to produce the microbeads. To fabricate Matrigel beads, the Matrigel liquid phase at 4 °C was loaded into a 1 mL syringe (Yuekang) connected with a fluid injection pump (LEAD FLUID) stored in a 4 °C refrigerator. The flow rate was set at 20 μL/min. The HFE 7000 oil was loaded in a 10 mL syringe (Yuekang) connected to another fluid injection pump (LEAD FLUID) and the flow rate was set at 300 μL/min. The Matrigel droplets were produced and collected in a long connection tubing (Woer, inner diameter (ID) = 0.56 mm), through a 3-way PDMS cube, and then gelled at 37 °C for 30 min. To fabricate gelation beads, the 5% (w/v) gelatin solution mixed with the saturated and filtered TG (Dongsheng Biotech, China) solution at the mixing ratio of 5:1 was loaded in a 1 mL syringe and placed at room temperature. Then the gelatin droplets were formulated under the periodic shear effect from the oil flow and gelled in the collection tubing at 37 °C for 15 min. To generate GelMA beads, 5% (w/v) GelMA was dissolved in 0.25% (w/v) LAP in PBS buffer at 60 °C and maintained for 1 h in a dark place. Then samples were centrifugated to remove the air bubble. Then the GelMA droplets were formulated as described above. The tubing was exposed to UV light at 405 nm for 1 min to crosslink GelMA microgel. The production of HAMA microbeads was identical to the practice of GelMA. Briefly, 5% (w/v) HAMA was dissolved in 0.25% LAP in PBS buffer at room temperature and maintained for 1 h in a dark place. Then samples were centrifugated to remove the air bubble. Then the HAMA droplets were formulated as described above, and the tubing was exposed to UV light at 405 nm for 1 min to crosslink HAMA microgel.

The tubing outlet is connected to a 3-port connector (PDMS shaped), with a side channel (width = 1 mm) open to the compression air (5 psi). The microbeads are then ejected through a vertical channel onto the culture plate or wound substrate by the compressed air flow at the intersection point of the nozzle. The air jet shoots a bead onto the substrate along a straight path and the beads are patterned in a defined manner controlled by the motion of the *XYZ* axis.

### Rheology test
To evaluate the rheological property difference between bulk gel and microbeads, 5% (w/v) TG-crosslinked gelatin and Matrigel was used

throughout this study and tested by a rotational rheometer (MCR302, Anton Paar, Austria). Briefly, the solidified microbeads were collected in the cubes. Then the microbeads were pelleted in the tube and centrifuged at 93×$g$ for 5 min. After that, the oil was aspirated and removed from the tube. The gel–oil complex was washed, centrifuged, and aspirated for multiple steps until all oil droplets were removed. In the rheological test, a 25-mm measuring plate PP025 was used in this study. An approximate volume of microbead assembly was placed onto the center of the rheometer plate using a spatula. Then the geometry was lowered to the desired gap height (1 mm used in this study). Redundant microbeads were removed using a scraper. Oscillatory strain sweeps (0.1–500%, 1 Hz) were carried out at room temperature. Oscillatory frequency sweeps (0.1–100 Hz, 1% strain) were performed at room temperature. Time sweeps (0.5% strain, 1 Hz, 2 min) were performed at room temperature to determine the storage ($G'$, Pa) and loss ($G''$, Pa) moduli of microbead assemblies and bulk gel. Shear ramp (0.01–100 s$^{-1}$) was used to study the relationship between viscosity and shear rate. All experiments were repeated at least three times.

### Granular microgels porosity

To characterize the granular gel porosity, 5% (w/v) gelatin, Matrigel, 5% (w/v) GelMA, and 5% (w/v) HAMA microbead assemblies from different microbeads ($D = 450$ and $650\,\mu$m) were produced. Bead samples were centrifugated to achieve different packing degrees (low-packing gel, 93×$g$ for 5 min; high-packing gel, 20879 $g$ for 5 min) and infiltrated with 10 mg/mL FITC-dextran (2 MDa) in PBS solution. Each microgel pellet was transferred to a PDMS holder (a 5-mm hole punched in the center) and adhered to a glass slide. A confocal microscope (Nikon A1) was used to visualize the FITC-dextran within interstitial pores.

### Live dead staining

The viability of cells in the hydrogel, bulk, and microbeads, was accessed using a LIVE/DEAD staining kit (Yeasen, China). The cell-laden hydrogels were washed three times with 1×PBS buffer. After that, cells were immersed with $2.0 \times 10^{-6}$ M Calcein AM (the live staining dye) and $4.5 \times 10^{-6}$ M PI (the dead staining dye) for 30 min, followed by gentle 1×PBS buffer washing. All microscopic images were captured under a fluorescent microscope (Nikon Eclipse Ts2r). The viability ratio was quantified by counting the number of live (green) and dead (red) cells in the images.

### Cell dissociation and cell counting

The cell growth of MSCs, NIH3T3, and SW620 cultured in Matrigel was detected by counting the growing cell numbers using collagenase type I (Sigma-Aldrich). 50 HDBs, 250 LDBs, 7.5 μL HDG were printed or cast in 12-well plates individually and further cultured. Then the Matrigel was dissolved using 1 mg/mL collagenase type I in PBS for 1 h. The dissolved cell was counted using hemocytometer (Qiujing, China). The live cell number was obtained by multiplying the total cell number and cell viability.

### LDH-CyQuant assay

MSCs growth in Matrigel was determined using the LDH Cytotoxicity assay kit (Yeasen, China) according to the manufacturer's protocol. Substrate mix, iodonitrotetrazolium (INT) substrate solution, INT dilution buffer, and enzyme solution were mixed together according to the volume ratio from the protocol and formulated the working solution. Cells were dissociated using 1 mg/mL collagenase type I in PBS for 1 h and incubated with cell lysis solution at 37 °C for 1 h. Then the supernatant was incubated with the working solution without light at 25 °C for 30 min. Finally, the optical density (OD) value was measured at 490 nm by Synergy H1 microplate reader (Bio-Teck, Winooski, VT, USA).

### Metabolic activity assay

The metabolic activity of cells embedded in the hydrogel was determined using the Alamar Blue assay (Thermo Fisher Scientific). Briefly, samples were washed with PBS buffer and then cultured using MSC cell culture medium mixed with the Alamar Blue reagent at the ratio of 1:9 for 24 h. Finally, the optical density (OD) value of the supernatant was measured at 570 nm by Synergy H1 microplate reader (Bio-Teck, Winooski, VT, USA).

### Measurement of DNA concentration and real-time quantitative PCR (RT-qPCR)

Genomic DNA and total RNA from Matrigel HDG-MSCs, LDB-MSCs, and HDB-MSCs were isolated using a DNA/RNA Isolation kit (TIANGEN, China). DNA and RNA concentration was determined using the Nanodrop 2000 (Thermo, US). 2 μg RNA was reverse transcribed to cDNA by the PrimeScript™ RT reagent Kit (TAKARA). cDNA was used for qPCR using the GoTaq® qPCR Master Mix (Promega, USA) with RT-qPCR instrument (Bio-Rad, CA). The primers used for the qPCR reaction were as follows: HGF, 5′- TCCAAGGTCAAGGAGAAGGCTACAG-3′ and 5′-CAGGAGTCATGTCATGCTCGTGAG-3′; TEK, 5′-TGCAGAGAACAACATAGGGTCAAGC-3′ and 5′-CAGGTCATTCCAGCAGAGCCAAG-3′; VEGFA, 5′-AGGCTCCAGGGCATTAGAC-3′ and 5′- AGGCTCCAGGG-CATTAGAC-3′; EPOR, 5′-CTTCTGGTGTTCGCTGCCTA-3′ and 5′-CCTCGTAGCGGATGTGAGAC-3′; glyceraldehyde phosphate dehydrogenase (GAPDH), 5′-CGACCCCTTCATTGACCTC-3′and 5′-CTCCAC-GACATACTCAGCACC-3′. The reaction mixture (25 μL) consisted of 12.5 μL SYBR Green, 2 μl template DNA, 1 μL each of the forward and reverse primers (1 nmol/mL), and 8.5 μL distilled water. The PCR protocol was as follows: 95 °C, 3 min, 95 °C, 30 s; 57 °C, 45 s; 39×. The relative gene expression level was measured by using the $2^{-\Delta\Delta CT}$ way and was normalized to GAPDH.

### 3-(4,5-dimethyl-2-thiazolyl)-2,5-diphenyl-2H-tetrazolium bromide (MTT) assay

The cell growth of MSCs was evaluated using MTT (Sigma-Aldrich) assay. For MSCs encapsulated in the hydrogel, 20 MSCs-laden HDBs, 100 MSCs-laden LDBs, or 3 μL MSCs-laden HDG were printed or cast in a 24-well plate individually. MTT reagent (5 mg/mL) was added to each well, then these cells were kept at 37 °C in the dark for 4 h. Dimethyl sulfoxide (DMSO) was added to dissolve formazan crystals. Finally, the optical density (OD) value was measured at 490 nm by Synergy H1 microplate reader (Bio-Teck, Winooski, VT, USA).

### 5-Ethynyl-2′-deoxyuridine (EdU) labeling

The proliferation rate of MSCs encapsulated in hydrogels was evaluated using an EdU Imaging Kit (Yeasen, China). For MSCs encapsulated in the hydrogel, 5 MSCs-laden HDBs, 25 MSCs-laden LDBs, or 0.75 μl MSCs-laden HDG were printed or cast in a 96-well plate. EdU staining was carried out according to the manufacturer's protocol. Briefly, the cells were incubated with $1.0 \times 10^{-6}$ M for 24 h. Then the culture media was removed and washed with PBS twice. 4% Polyformaldehyde (PFA) was used to fix the cell for 30 min and neutralized with 2 mg/mL glycine for 5 min. After washing with 3% bovine serum albumin (BSA) in PBS, cells were permeabilized with 0.5% Triton X-100 in PBS. Then the cells were stained with Yefluor 488Azide (1:500) for 30 min at room temperature. The nuclei were stained with 1:5000 4′,6-diamidino-2-phenylindole (DAPI) (Sigma-Aldrich) for 10 min, followed by washing with PBS for 3 times and visualized under a fluorescent microscope (Nikon Eclipse Ts2r). The quantification of EdU labeling was done by calculating the proportion of positive cells in the imaging field.

### Crystal violet staining

The migration ability of MSCs was tested using crystal violet staining cultured in a transwell chamber (6.5 mm width, 8 μm pore size,

Corning). 5 MSCs HDBs, 25 MSCs LDBs or 0.75 μl MSCs HDG were printed or cast into the upper chamber of the transwell. Then the transwell chambers were plated into a 24-well plate. After 3 and 7 days, the cells in the chamber were fixed with methanol for 30 min and stained for 30 min with 1% crystal violet (C0775, Sigma-Aldrich). Non-migrated cells inside of transwell were wiped off by a cotton swab. The images of migrated cells were captured by a microscope (Nikon Eclipse Ts2r). The crystal violet stained in migrated MSCs is dissolved by 33% ethylic acid and measured at 570 nm by Synergy H1 microplate reader (Bio-Teck, Winooski, VT, USA).

## Flow cytometry

MSCs from the third generation were cultured in 2 days condition, Matrigel HDG, LDB, and HDB groups for 4 days and then harvested in Cell Staining Buffer (420201, BioLegend) and stained with fluorescent-conjugated primary antibodies, phycoerythrin (PE)-conjugated human antibody CD90 (5 μL per million cells in 100 μl staining volume, Bio-Legend, Cat #328123) and allophycocyanin (APC)-conjugated human antibody CD105 (5 μL per million cells in 100 μL staining volume, Bio-Legend, Cat #323225), for 30 min at 4 °C in the dark. Processed cells were then washed twice with wash buffer and analyzed by flow cytometry (BC16129, Beckman Coulter). The flow cytometry data were analyzed with FlowJo software (version 10). The details of gating strategies could be obtained from Fig. S50.

## Osteogenic differentiation protocol

The MSCs were induced to differentiate osteogenically using an OriCell Human Umbilical Cord Mesenchymal Stem Cell Osteogenic Differentiation Kit (Cyagen Biosciences) following the manufacturer's instructions. MSCs-laden microbeads were induced to differentiate either immediately post-formulation, or after being cultured using DMEM/F12 (Gibco, China) supplemented with 10% (w/v) FBS and 1% (w/v) P/S for 5 days. For cell suspension differentiation, MSCs were differentiated when 70% confluency was reached. Osteogenic differentiation tests were tracked for 3 weeks and stained with Alizarin Red S staining.

## Adipogenic differentiation protocol

The MSCs were induced to differentiate adipogenically using an Ori-Cell Human Umbilical Cord Mesenchymal Stem Cell Osteogenic Differentiation Kit (Cyagen Biosciences) following the manufacturer's instructions. MSCs-laden microbeads were induced to differentiate either immediately post-formulation, or after being cultured using DMEM/F12 (Gibco, China) supplemented with 10% FBS and 1% P/S for 5 days. For cell suspension differentiation, cells were differentiated when 100% confluency was reached. Adipogenic differentiation tests were tracked for 3 weeks and stained with Alizarin Red S staining.

## Western blot

Cells cultured in Matrigel were collected and lysed with RIPA buffer (MCE) containing protease and phosphatase inhibitors (MCE). Protein concentrations were determined by the BCA Protein Assay Kit (23227, Thermo Fisher Scientific). Western blotting was conducted according to a standard protocol. Briefly, the proteins were loaded on 10% SDS-polyacrylamide gels and electrophoretically transferred to a PVDF membrane (Millipore). The primary antibodies were purchased as follows: p-Akt (1:1000, ABclonal, AP0637), Akt (1:1000, ABclonal, A17909), p-GSK3β (1:1000, ABclonal, AP1088), GSK3β (1:1000, ABclonal, A6164), β-CATENIN (1:1000, ABclonal, A19657), β-actin (1:1000, ABclonal, AC026), anti-CD9 (1:1000, Abcam, ab263019), anti-CD81 (1:1000, Abcam, ab109201), anti-Hsp70 (1:1000, Abcam, ab181606), anti-TSG101 (1:1000, Abcam, ab125011), anti-Calnexin (1:1000, Abcam, ab133615), anti-CD63 (1:1000, Abcam, ab134045). Detection was performed using a Chemiluminescent Western Blot detection kit (4AW012-1000, 4A Biotech). The western blot results were analyzed using ImageJ (version 1.52). Uncropped and unprocessed scans of the blots could be obtained from the source data file.

## Immunofluorescence staining of 3D MSCs

MSCs were cultured in Matrigel HDG, LDB, and HDB groups for 1 week and then fixed using 4.0% PFA for 1 h at room temperature followed by three times washing using PBS. Then the MSCs were permeabilized with 0.5% Triton X-100 for 30 min at room temperature and further incubated with 5% BSA for 1 h. After that, MSCs were incubated with primary antibodies against pAKT (1:100, Cell Signaling, Cat #4060S), AKT (1:100, Cell Signaling, Cat # 4685S) and β-CATENIN (1:100, Cell Signaling, Cat #8480S) at 4 °C overnight. After three-time washing using PBS, the samples were incubated with Alexa Fluor 488 conjugated secondary antibody (1:500, Abcam, Cat #ab150073) at 4 °C overnight. Finally, the samples were stained with DAPI. The images were captured under a fluorescent microscope (Nikon Eclipse Ts2r).

## EV Isolation

MSCs were cultured in Matrigel HDG, LDB, and HDB groups containing complete DMEM/F12 medium for 5 days and further cultured in complete medium containing EV-depleted FBS (C38010100, Vivacell) for 96 h. The medium of MSCs was collected and centrifuged at 2000×g for 10 min to remove cell debris followed by centrifugation at 10,000×g to remove microvesicles (>500 nm), and then passed through a 0.22-μm filter to remove the cellular debris. Afterward, the supernatant was added to a 38.5 mL polypropylene ultracentrifuge tube (Beckman) and centrifuged at 120,000×g for 70 min at 4 °C. Then, the supernatant was removed, and EV pellets were washed in PBS followed by centrifugation at 120,000×g for 70 min at 4 °C. The final EV pellet was resuspended in PBS and stored at −80 °C.

## Protein quantification and size characterization of EVs

The protein concentration of EVs was assayed by a BCA Protein Assay Kit (23227, Thermo Fisher Scientific). EVs isolated from complete median containing EV-depleted FBS by ultrafiltration were analyzed by nanoparticle tracking analysis (NTA) to determine vesicle concentration and size distribution using NanoSight NS300 instrument (Malvern, Worcestershire, UK) with scientific CMOS sensor. The samples were diluted in filtered PBS and three 40 s videos were recorded. The data were analyzed using NTA software 3.0 with the detection threshold 7.

## Morphology characterization of EVs using TEM

The Matrigel HDG-EVs, LDB-EVs, and HDB-EVs samples (10 μl) were deposited on formvar-carbon-coated grids for 2 min and gently blotted on filter paper. Next, the grids were stained with 3% uranyl acetate for 1 min at room temperature and quickly blotted on filter paper. After drying, the grids were examined in Tecnai G2 Spirit transmission electron microscopy at 80 kV.

## Tracking of EVs uptake

The Matrigel HDG-EVs, LDB-EVs, and HDB-EVs were labeled using PKH67 dye according to the manufacturer's protocol to complete the colocalization. Briefly, the isolated EVs were diluted using Diluent C, then mixed with PKH67 dye by gentle pipetting and let stand at room temperature for 5 min. The staining reaction was quenched by adding EV-depleted FBS. Next, the superfluous dyes were removed by centrifugation. And the EVs were resuspended in 1× PBS by gentle pipetting. C2C12 cells were stained with Dil cell membrane Tracker Red (MB4240-1, Meilunbio) and treated with 10 μg/mL PKH67-labled HDG-EVs, LDB-EVs, and HDB-EVs in a complete medium containing EV-depleted FBS for 2 days. Then the colocalization of EVs and cells were visualized using a confocal microscope (Nikon A1) at ×100 magnification. Fluorescence intensity was obtained by ImageJ software. Pearson's correlation coefficient was calculated from randomly six regions.

## C2C12 growth

C2C12 cells were seeded at $2.0 \times 10^4$ cells/cm² in a 24-well plate and treated with Matrigel HDB-EVs (2.5 and 5 µg/mL), then cultured in a complete medium containing EV-depleted FBS for 48 h. The control group was treated without any EVs. After 2 days, the cell growth was determined by MTT assay as same as described previously and visualized under a microscope (Nikon Eclipse Ts2r).

## C2C12 proliferation

C2C12 cells were seeded at $2.0 \times 10^4$ cells/cm² in a 96-well plate and treated with 10 µg/mL Matrigel HDG-EVs, LDB-EVs, and HDB-EVs. The control group was treated without any EVs. Afterward, the cells were cultured in a complete medium containing EV-depleted FBS for 2 days. The proliferation rate of C2C12 was detected using the EdU labeling method as same as described previously and visualized under a fluorescent microscope (Nikon Eclipse Ts2r).

## The mouse VML defect model

All the animal experiments were performed under the ethical regulation of Shenzhen International Graduate School (SIGS) of Tsinghua University. All the mice were housed in a pathogen-free environment with the temperature maintained at $23 \pm 2\,°C$ and relative humidity at 50–65% under a 12 h/12 h light/dark cycle with free access to food and water, in accordance with the National Institutes of Health guidelines. The skeletal muscle defect in GM muscle was created in balb/c mice (male, 4–6 weeks old, Guangdong Medical Laboratory Animal Center, China). Mice were anesthetized with 2–5% isoflurane, disinfected with serial washes of povidone–iodine and ethanol, and then received a 2-3 cm incision on the dorsal skin of the gluteus. After that, 60% volume of GM muscle was removed from either side of gluteus. The printed or cast constructs were implanted in the defect zone. Six groups were carried out at 1 week, 2 weeks, 3 weeks, and 4 weeks: (1) non-treated group (defect only, without any treatment); (2) acellular bead group; (3) cell injection group; (4) Matrigel HDG-MSCs group; (5) Matrigel LDB-MSCs group; (6) Matrigel HDB-MSCs group. $9 \times 10^4$ cells were injected, cast, or printed into the defect site, including 9 µL HDGs and 36 µl acellular gel, 300 LDBs, and 60 HDBs and 240 acellular beads (Table S2). After implantation, the fascia was closed and the skin was sutured using a transparent dressing (Vbon).

## Mouse excisional wound-healing model

Briefly, the balb/c mice (male, 4–6 weeks old, Guangdong Medical Laboratory Animal Center, China) were anesthetized with 2–5% isoflurane and disinfected with serial washes of povidone–iodine and ethanol. 5, 10, or 15 mm rounded full-thickness excisional wounds were created along the dorsal midline. The printed or cast constructs were implanted in the wound region. Six groups were performed at each time point: (1) non-treated group (defect only, without any treatment); (2) acellular bead group; (3) cell injection group; (4) Matrigel HDG-MSCs group; (5) Matrigel LDB-MSCs group; amnd (6) Matrigel HDB-MSCs group. For 5 mm, rounded full-thickness excisional wounds, $2.1 \times 10^4$ cells were injected, cast, or printed into the defect site, including 2.1 µL HDGs and 8.4 µl acellular gel, 70 LDBs, and 14 HDBs and 56 acellular beads. For 10 mm, rounded full-thickness excisional wounds, $8.4 \times 10^4$ cells were printed into the defect site, including 56 HDBs and 224 acellular beads. For 15 mm, rounded full-thickness excisional wounds, $1.8 \times 10^5$ cells were printed into the defect site, including 120 HDBs and 480 acellular beads (Table S2). After treatment, the skin was sutured using a transparent dressing (Vbon). The wound closure was recorded on days 0, 7, 10, 15, and 20. The dynamic wound healing process was analyzed using ImageJ. At the culmination of the wound-healing experiment (day 21), the mice were killed using an overdose of isoflurane and the tissue was taken out from the transplantation region.

## MSCs retention rate investigation in vivo

Retention, growth, and distribution of MSCs in regenerated muscle tissue were tracked using fluorescence-labeled MSCs. Briefly, MSCs were labeled using the red Dil cell membrane tracker (Meilunbio, 1:1000). Acellular beads and acellular bulk gel were loaded with monodispersed green fluorescent polystyrene microspheres (Aladdin, 1 µm, 1:20 diluted in Matrigel solution). Then Matrigel HDG-MSCs, LDB-MSCs, and HDB-MSCs were transplanted in the injury sites. After a certain time, the mice were euthanized and samples were harvested. ImageJ was used to analyze cell density per section. MSCs density was calculated as the fraction area of red fluorescence per section. Total density was calculated as the fraction area of blue fluorescence per section.

## Histologic and immunostaining analysis

For histological and immunostaining evaluation, all harvested tissue samples were fixed using 4% PFA overnight, followed by 1× PBS washing 3 times. Then the tissues were dehydrated using a 30% saccharose solution for 12 h. The samples were immersed in an optimal cutting temperature (OCT) compound and quickly frozen in liquid nitrogen. The samples were cryosectioned into 10-µm slides. H&E and Masson's trichrome staining was performed according to the manufacturer's instructions. Images were captured on a Leica DM1000 microscope.

For immunofluorescence, the slides were blocked using 5% Albumin BSA in PBS supplemented with 1% Triton X-100 for 1 h at room temperature. Then the sides were incubated at 4 °C for 24 h with rabbit anti-desmin (1:500, Servicebio, GB11081), rabbit anti-MYH7 (1:400, Servicebio, GB111857), mouse anti-human/mouse/rat/chicken Pax7 (5 µg/mL; R&D Systems, MAB1675), rabbit anti-CD31 (1:200, Servicebio, GB11063-3), rabbit anti-α-SMA (1:100, Servicebio, GB111364), rabbit anti-NF (1:100, Abcam, ab223343), rabbit anti-βIIIT (1:100, Abcam, ab229590), rat anti-CD68 (1:100, Invitrogen, Cat# 14-0681-82), rabbit anti-CD3 (1:100, Abcam, ab16669), rabbit anti-Ki67 (1:600, Servicebio, GB111141), rabbit anti-HLA (1:100, Abcam, ab52922). Slides were then washed by 1× PBS washing for 3 times and incubated in the secondary antibody such as anti-rabbit secondary antibody (1:400) (Alexa Fluor 488, Abcam), anti-rabbit secondary antibody (1:400) (Alexa Fluor 594, Abcam), anti-mouse secondary antibody (1:400) (Alexa Fluor 488, Abcam), anti-rat secondary antibody (1:400) (Alexa Fluor 488, Abcam) and anti-rat secondary antibody (1:400) (Alexa Fluor 568, Abcam) at 4 °C for 24 h. Finally, the slides were mounted using DAPI (1:5000) and the images were acquired under a confocal microscope (Nikon A1). Negative control was performed in parallel.

## RNA-seq analysis

The mRNA sequencing of cells, animal samples, and EV samples was performed in GENEWIZ (Suzhou) Co., Ltd. China (for cells) and BGI-Shenzhen, China (for EV and animal samples). Briefly, the cells were cultured for 7 days and send to GENEWIZ (Suzhou). The GM muscle sample was taken out after 4-week implantation and sent to BGI-Shenzhen. The skin tissue sample was taken out after 4-week implantation and sent to BGI-Shenzhen. The sequencing process was carried out according to the company's protocol. EVs were collected from cells cultured in different conditions and were sent to BGI-Shenzhen for small RNA sequencing. DESeq2 package was used to identify differential expression genes. Gene expression value was normalized by Fragments Per Kilobase of transcript per Million mapped reads (FPKM). The Kyoto Encyclopedia of Genes and Genomes (KEGG) pathway was analyzed online (https://david.ncifcrf.gov/).

## Statistics and reproducibility

All the statistical analysis was performed using Origin 2021b (OriginLab Co, Northampton, MA) and Prism 6 (GraphPad, Inc.) software. Significant differences between parallel groups were calculated using

unpaired two-failed Student's *t*-test (comparing two experimental groups) or one-way ANOVA (more than two groups involved) following Tukey's multiple comparisons test. Unless indicated otherwise, the data are represented as mean ± SD and a significant difference was considered as $p < 0.05$. All representative fluorescence images, TEM images, H&E staining, and MTS images, osteogenic and adipogenic differentiation images, and bright field images shown in the main text and supplementary information were repeated at least three times independently with similar results.

### Reporting summary

Further information on research design is available in the Nature Portfolio Reporting Summary linked to this article.

## Data availability

The RNA-seq data described in this article are available at Gene Expression Omnibus (GEO) under accessions "GSE214868" and "GSE197356". The small RNA-seq data are available under "GSE215294". All other data needed to support the conclusions in the paper are presented in the paper and/or the supplementary information or from the corresponding author upon reasonable request. Source data are provided with this paper.

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

## Acknowledgements

The work was supported by the National Natural Science Foundation of China (Grant Numbers: 61971255 and 82111530212), the Natural Science Foundation of Guangdong Province (Grant Number: 2021B1515020092), the Shenzhen Science and Technology Innovation Commission (Grant Numbers: GJHZ20210705143001004, RCYX20200714114736146, WDZC20200821141349001, KCXFZ20201221173207022, KCXFZ20200201101050887), and the Shenzhen Bay Laboratory Fund (Grant Number: SZBL2020090501014). All schematics shown in the main text and supplementary figures were created with biorender.com.

## Author contributions

S.M. and Y.C. conceived and designed this work. Y.C. set up the bead-jet printer prototype and completed the data analysis. Y.C., J.T., T.D., Y.H., J.Z., and Z.H. completed the cell experiment. Y.C. and H.Z. performed the animal experiment. B.X. and Z.W. provided help with animal experiment design. J.L. and K.H. provided help with figure visualization. S.M. supervised this work. Y.C. and S.M. wrote the manuscript. Y.C., S.M., P.E., and Y.W. revised and edited the manuscript with input from all authors.

## Competing interests

The authors declare no competing interests.
