## [Peer Review File · Nature Communications]

REVIEWER COMMENTS

Reviewer #1 (Remarks to the Author):

The authors have devised a microfluidic setup to generate microbeads containing high (1E7/ml) and low (1E6/ml) density of MSCs. In comparison with casted MSCs, the authors have shown that the high density beads have higher proliferation and cell migration when comparing among the 3 different groups. The authors have also highlighted the need to use high cell density microspheres in order to achieve higher efficacy of wound closure in the animal, while also showcasing the potential for using the technology for in situ printing.

Similar microbeads fabrication techniques have been proposed using microfluidic to produce microgels by shearing dispersed phase with a continuous phase [Feng, Q., Li, Q. T., Wen, H. J., Chen, J. X., Liang, M. H., Huang, H. H., Lan, D. X., Dong, H., Cao, X. D., Injection and Self-Assembly of Bioinspired Stem Cell-Laden Gelatin/Hyaluronic Acid Hybrid Microgels Promote Cartilage Repair In Vivo. *Adv. Funct. Mater.* 2019, 29, 1906690. <https://doi.org/10.1002/adfm.201906690>; de Rutte, J. M., Koh, J., Di Carlo, D., Scalable High-Throughput Production of Modular Microgels for In Situ Assembly of Microporous Tissue Scaffolds. *Adv. Funct. Mater.* 2019, 29, 1900071. <https://doi.org/10.1002/adfm.201900071>]. Authors should cite these articles and compare the advantageous and disadvantageous between these techniques to highlight the strength of in situ printing microbeads rather than injecting/casting. As shown in Figure 2h, additional print layers result in deformation of the structure that resembles a pyramidal shape. The authors should evaluate the structural integrity of printed structure using methods as described in [Lee, J. M., and Yeong, W. Y. Engineering macroscale cell alignment through coordinated toolpath design using support-assisted 3D bioprinting. *Journal of the Royal Society Interface* 17.168 (2020): 20200294. <https://doi.org/10.1098/rsif.2020.0294>] to determine the integrity of the printed microbeads layer. The authors have to justify the need for printing these microbeads since previous works have shown that microgels can be delivered to the implant site through injection.

Specific comments

- How were the microbeads loaded onto the rheometer for measurement? How did the authors ensure complete contact of microbeads between the parallel plates? Can the authors explain the observation of $G'' > G'$ at lower shear strain % for microbeads? Also, can the author clarify whether 10 Hz or 10 rad/s was used for the strain sweep test?
- It is advisable for the authors to elaborate on the details regarding the animal model. How did the authors determine the amount of cells to be delivered at the wound site? with the animal model study, what was number of cells and volume of materials loaded in the cast group?

Reviewer #2 (Remarks to the Author):

"Bead-Jet" printing of high-density mesenchymal stem cell-laden Matrigel beads augments skeletal muscle and hair follicle regeneration

The authors propose a bead jet printer to print high and low density MSC loaded Matrigel beads in order to effectively improve regenerative outcomes of skeletal muscle and hair follicles. The proposed system was tested both in-vitro and in-vivo and shows good promise

in causing successful regeneration of intended tissue. Authors have also shown that the encapsulation of MSCs into the Matrigel beads using the novel bead jet printing technology does not affect the extracellular vesicle production capabilities of the MSC and might still maintain its therapeutic potential. Bead printing parameters were first set using gelatin as the bioink and viability of printed MSC loaded beads was assessed. Using these conditions, various combination of MSC loaded Matrigel beads (high density vs low density) as well as bulk gel formulations were tested for their viability and EV formation in-vitro. The authors also went a step further and tested the viability of NIH3T3 fibroblasts and SW620 when bioprinted. In-vivo studies in mice showed that printed microbeads were able to cause functional restoration of muscle mass even after 60% removal as evidenced by H&E and MTS staining for collagen and fibrosis followed by RNA-seq analysis of retrieved tissue. Ability of the newly regenerated muscle fibers to undergo skeletal integration was assessed by IHC for desmin (myofiber marker), alpha-SMA (artery development), CD31 (endothelial marker), neurofilament and β IIIIT (neural integration). While the presented study addresses the important consideration of spatial patterning of MSCs required to ensure successful regeneration, there are several gaps in the experimental studies that are outlined below. Overall, this study provides a good foundation to the application of bead printing technology to ensure spatial patterning of MSCs in order to improve the regenerative success in case of irregular sized large defects.

Accept with revisions

Major Concerns:

- 1) The lab has previously published an article in Advanced biology titled "Injectable Mesenchymal Stem Cell-Laden Matrigel Microspheres for Endometrium Repair and Regeneration" using the same bead composition and similar technology. It is unclear what the difference between this printer and the microfluidic device used in that article is. Can the authors elaborate on the advantage of a printer over the microfluidic device? In previous work the simpler setup does work equally well and is more easily adaptable to customizations.
- 2) What is the size limit of the injury that can be filled with the MSC loaded beads using bead jet printing while still maintaining sterility and successful regeneration?
- 3) The distinction between the LDB, HDB and HDG is still unclear given that the initial cell seeding density is the same. Can the authors clarify the logic behind this further? What is the number of MSCs loaded per microsphere for each of these bead groups since that will also affect regeneration? MSCs per sphere or MSCs per unit volume needs to be calculated since it will affect in-vitro and in-vivo outcomes.
- 4) The authors talk about gel bead assemblies maintaining structural integrity at low shear, but being prone to deformation under increased shear. While advantageous for in terms of motion adaptation, wouldn't it result in increase degradation of beads leading to sub-optimal performance? Some larger sized defects may require longer residence of cells.
- 5) The initial cell seeding density of HDG, LDB and HDB seems to be very high, lowest being 2×10^6 cells in LDB. What is the average number of cells per bead? The high cell number could be causing metabolic stress on the MSCs leading to altered functions. Normalization of cells/bead based on DNA content or number of nuclei needs to be done.
- 6) Because of the high cell density used, the MTT assay is always going to give high viability (False positive). Viability data needs to be tested for rigor with one more viability tests such LDH-CyQuant assay which allows for normalization with cell number/DNA content. Also, viability must be assessed over longer time points at least till Day 14.
- 7) Characterization of the EV cargo needs to be performed to ensure that the MSC derived EVs produced after encapsulation in Matrigel beads still possess therapeutic anti-inflammatory potential and are not just producing EVs because of cell stress, which is a common occurrence.
- 8) While IHC shows upregulation of markers for successful innervation, reendothelialization and vascularization of defects, the RNA-Seq data does not reflect the same. GO analysis needs to be performed again and specific terms related to angiogenesis, follicle

development, ECM remodeling etc need to be highlighted that are relevant to the effect being seen. Upregulation of Ribosome related pathway is a very broad term and cannot be used since it shows up as one of the top regulated processes in any type of RNA-seq analysis with any cell/tissue type.

9) The errors in most of the bar graphs in Figure 8 are very large especially for HDB. How was the quantification done? Was normalization to total tissue area performed?

Minor Concerns:

1) Matrigel inherently has a lot of batch-to-batch variation resulting in poor control over bead micro-architecture. How does the bead jet printing address this issue?

2) Matrigel is known to cause malignant transformation of certain cell types, like fibroblasts. Have any assays been done to check this doesn't happen?

3) More TEM images of EVs should be analyzed for size distribution, at least 15-20.

Reviewer #3 (Remarks to the Author):

This manuscript by Cao et al. reports a bead-jet printing technique that deposits cell-laden and acellular Matrigel beads directly onto wound sites. This study is motivated by a need to optimize cell-based therapies for tissue regeneration by using biomaterials and biofabrication approaches to enhance cell function. For translational applications, an intra-operative technique is desirable and one that allows deposition of therapeutic material in a scalable and high-throughput manner even more so. Cao et al. report that MSCs encapsulated at a high density within Matrigel beads and deposited in a pre-defined pattern stimulate muscle and skin regeneration. They attribute these outcomes to the granular nature of their beads, the specific pattern in which they are printed, MSCs EV production and other behaviors including migration, proliferation, and trans-differentiation.

Although this manuscript aimed to tackle an important topic in the field, there are many scientific and technical concerns around claims made about the used materials, printing technique, biological outcomes observed, and conclusions derived. The authors make numerous conjectures while discussing their data without performing the experiments necessary to rigorously test their hypothesis.

Major concerns are listed below for the authors' consideration.

Engineering limitations/concerns:

1. The use of Matrigel dampens any biomaterials-related enthusiasm for this printing technique and outcomes described. Out of the many synthetic and naturally derived biomaterials available and in use, Matrigel is perhaps one that has the most limited future potential in terms of translational applications. The field has recently focused on producing designer synthetic matrices and moving away from Matrigel - despite its historical use and utility - due to well-known issues centering around batch to batch variability (see Nguyen et al. Nat. Biomed. Eng. 2017), xenogeneic origin, and inability to tune biophysical and biochemical properties. None of this is discussed. More alarmingly, the authors cite two of their own papers to show the applicability of Matrigel in tissue engineering applications. The authors should provide a rationale for using Matrigel as their cell-encapsulating material and explore tunability of its properties and related cell outcomes. Availability of Matrigel is also a concern not addressed by authors - Matrigel is currently backordered in commercial supplier stocks even for lab-based use. How can this material be translated into a commercial product for use in patients?

2. For broad applicability to the field, the authors should demonstrate that their bioprinting technique is compatible with other soft polymer materials that have a higher chance of

translation or that are already being used in pre-clinical and clinical studies e.g., PEG, alginate, and hyaluronic acid.

3. The authors claim that their bead-jet technique is specialized for high-throughput printing, but fail to define what high throughput means. Their movies show a rather slow (bead-by-bead) deposition of the materials into the animal wound site. Furthermore, the methods describe a gelation step that requires 15-30 minutes prior to being deposited. This is far away from "high-throughput". As an example, in the field of microfluidics where similar cell-encapsulating beads are a focus of research, high throughput typically means Liters or kilograms worth of material per hour. Authors should demonstrate that they can print materials and fill large clinically relevant voids (102-103 cm³) within surgically relevant time scales (i.e. 101-102 minutes).

4. Patterning high-cell density beads among acellular ones is not rationalized or based on any optimized layout. The manuscript claims that one of the advantages of their bead-jet printing technique is the patterned deposition of cellular and acellular materials. However, this aspect is not experimentally explored or discussed at all. For example, how is the deposition pattern optimized? Was a design criteria established or various patterns tested for their biological advantage? Do endogenous cells invade and cluster around cell-laden beads? Is there a heterogenous HLA signal intensity based on where cell laden beads are present compared to acellular regions?

5. Evidence to support patterned deposition of beads in wound models is missing. The authors should provide an early time point (e.g. 24 hours) histological image of the wound site where beads were deposited in the HDB group. They should demonstrate that cell-laden beads are present among acellular beads.

6. Regarding patterning and printing precision, authors show a simplistic macro-deposition. More complex patterns using beads have been reported recently by others including Xin et al. Adv. Funct. Mater. 2020 and Li et al. Adv. Mater. 2018 1803475. In comparison, this manuscript does not demonstrate any heterogenous 3D patterns despite their entire therapeutic strategy based on these patterns.

7. It is claimed that neighboring beads adhere/bond to each other but the mechanism of how this occurs is not described. Clearly there is no chemical functionalization performed on Matrigel. If it is solidified and crosslinked, then it can no longer fuse with adjacent material.

8. Authors claim that printing a 15-layer 3D assembly of beads retains its structural integrity. It is not clear how this is concluded from data that shows a drooping of the material and change in angle from 90 degrees to 50 degrees.

9. Rheology performed to demonstrate extrudability is in a low strain range (0.1-1%) whereas standard in the field for these types of experiments is 0.1-500%. This experiment should be repeated and yield strain (cross-over G' with G'') should be reported.

10. A limitation of this technique is that it can print either cell-laden or acellular materials in a batch-wise manner. This means that if a few cell-laden beads are deposited into the wound site, then they need to stay in place until the new acellular material is setup and deposited into empty areas. It is hard to imagine how this is an optimal approach with such a time-lag. More explanation and discussion should be included.

11. Authors cite recent work related to granular hydrogels and motivate their use of beads by citing advantages of granular systems including microporosity and extrudability.

However, the results are not discussed in the context of these benefits. Even though the beads are expected to degrade quite quickly (being Matrigel), there is clearly an advantage over the bulk gel (HDG) group. This should be explored by early-stage (1-3 days) immunohistochemical analysis of beads in the wound site. At a minimum, in vitro models could be used to highlight differences due to the bead morphology. Authors should characterize the degree of bead packing and interstitial porosity.

12. At least two relevant negative control groups are missing in the in vivo experiments. 1) Acellular beads only – this is expected to elicit some degree of biological/regenerative response since Matrigel is a naturally derived matrix. 2) Cell injection without a material – since cell delivery as a bolus is the gold standard in the clinic, this should be included to demonstrate superiority over the biomaterial groups.

13. Another technical concern is the deposition of oil at the site of injury. Oil and surfactant removal is one of the most important steps in other techniques e.g. batch emulsions and microfluidics that are used to fabricate beads and microgels for tissue implantation. The authors claim that the oil used (Novac 7500) is a volatile liquid that evaporates when exposed to air should not be an excuse to ignore the potential concerns over oil in the tissue site. Especially for scaling up bead deposition, it is easy to imagine how rate of oil evaporation would be masked by rate of bead +oil deposition leaving behind more oil in the wound site.

14. It is concerning that the authors compare two very different materials i.e. Matrigel and gelatin, which vary not only in physical features but chemical compositions, and claim that Matrigel is better than gelatin as an extracellular scaffold based on viability and proliferation experiments. These types of comparisons and statements should be removed and the authors should revisit and reconsider their approach to fairly comparing biomaterials.

Biological limitations/concerns:

15. Dosage determination is not optimized. It is unclear what is the basis for using the cell density employed. For a manuscript that is motivated by clinical challenges of high-dose therapies, this is an important aspect. Authors should investigate a dose-response relationship within their HDB and LDB groups. The manuscript also contains seemingly conflicting statements such as “sparse patterning of dense MSCsenabling therapy of large scale traumatic injury with a limited number of MSCs.” where the solution to the problem of high cell doses is the encapsulation of high cell doses in fewer materials.

16. For the skeletal muscle work, the authors show data that suggests MSCs (HLA) differentiate and integrate into muscle fibers. Although MSCs may integrate structurally into the tissue, satellite cells and other muscle-resident cells are most likely key players in the regeneration process. However, this is neither discussed nor explored. Further, is differentiation the common mechanism through which regeneration is achieved in the muscle and skin models?

17. The muscle injury model is defined as a ~60% removal of tissue by volume, but how this was achieved (e.g., with a biopsy punch or other) is not described. Considering the small size of mice muscles, this information about the technique will be relevant for the field.

18. Authors should clarify if the transplanted MSCs adopt roles in skeletal muscle that are similar to those performed by fibro/adipogenic progenitors, which are also a stromal cell population. Single cell sequencing or other similar techniques could be performed to

determine how the muscle-resident cell population changes as a result of treatment.

19. Muscle regeneration is a complex process that involves diverse cell types including immune cells. The authors should perform a more rigorous analysis of how the immune compartment (macrophages, T-cells, neutrophils) is altered in response to MSC therapy – especially since MSCs are known for their immunomodulatory effects.

20. Authors use desmin as a marker for regenerated fibers. This is not the standard in the field and is not exclusively used as a marker for new fibers. Authors should perform embryonic myosin heavy chain staining, or quantify centrally nucleated myofibers. Both of these are established markers of muscle fiber regeneration.

21. Although HLA is shown as a marker for cell presence, no negative controls are provided and the signal intensity is very low in relation to the number of cells transplanted. Authors should perform in vivo imaging with labeled cells atleast in the HDB group to show the duration of cell presence at the site of injury.

22. Authors make many conjectures related to the mode of action of MSCs in their models. For example, they claim that “muscle regeneration may be attributed to MSCs proliferation and EVs production in the initial stage and MSCs differentiation in the later stage”. Also “we tend to believe that both MSCs themselves and EVs promoted skeletal muscle regeneration and, likewise, skin regeneration”. There is no evidence provided to support these claims. MSCs’ ability to secrete EVs and also differentiate into multiple lineages is well-established in vitro but that does not necessarily mean that it’s how they stimulate regeneration. At a minimum, more experiments should be performed to characterize what these cells are producing in vivo (e.g., RNA seq of explanted cells). Other conjecture includes low MSC proliferation and migration due to intercellular communication and low concentration of secreted molecules. This is not backed up by any evidence.

23. The authors should revisit their explanation for why beads in HDB group are breaking down more rapidly than those in LDB group. The manuscript states that this implies a higher activity of cells in the HDB group. However, it is likely that this is due to a obvious significant difference in the number of cells that secrete enzymes such as MMPs. The authors could also perform cell metabolic activity experiments to test their hypothesis.

24. The authors describe a loss in stemness of MSCs in LDB group by quantifying the expression of surface markers. Performing multi-lineage differentiation experiments would help contextualize this loss of stemness.

25. The authors perform RNA-sequencing and observe many differentially expressed genes between groups. However, they then pick some of these genes and attribute differences in cell behavior to this expression pattern. For example, they implicate the PI3K-AKT pathway – which is an ubiquitous signaling pathway involved in many biological processes. They do not discuss other highly differentially expressed genes e.g. those related to cancer pathways (which might be due to Matrigel being derived from the ECM of a cancerous cell line). Further, their analysis of the western blots of the AKT pathways needs revision as they do not normalize or quantify pAKT to AKT expression which, when performed, will likely remove any differences between groups.

26. Histological images are very small making it very challenging to see any structural details. Some of these seem to be from smaller regions of interest within larger wound sites. The authors should provide higher resolution images and indicate which part of the tissue the regions of interest are extracted from.

Writing and References:

27. Authors do not provide a rationale for using Matrigel and fail to discuss its well known drawbacks, as discussed in many recent papers including Nguyen et al. Nat. Biomed. Eng. 2017, and Aisenbrey and Murphy Nat. Rev. Mater. 2020). Authors should provide an objective and well-balanced discussion on the use of Matrigel and future implications on their technique and outcomes.

28. Relevant research articles related to 3D intra-tissue printing have not been cited. Authors should compare their printing technique with recently reported approaches including intravital 3D bioprinting (Urciuolo et al. Nat. Biomed. Eng. 2020) and direct in situ printing into muscle tissues (Russell et al. ACS Appl. Biomater. 2020).

29. Authors do not discuss their proposed therapy in relation to others involving the use of hydrogels e.g. Han et al. Sci. Adv. 2018. They also do not discuss the merits of mesenchymal stromal cell therapy compared to transplantation of muscle stem cells (i.e. satellite cells) see Judson et al. NPG Regen. Med. 2020.

REVIEWER COMMENTS

Reviewer #1

The authors have devised a microfluidic setup to generate microbeads containing high ($1E7/ml$) and low ($1E6/ml$) density of MSCs. In comparison with casted MSCs, the authors have shown that the high density beads have higher proliferation and cell migration when comparing among the 3 different groups. The authors have also highlighted the need to use high cell density microspheres in order to achieve higher efficacy of wound closure in the animal, while also showcasing the potential for using the technology for in situ printing.

Similar microbeads fabrication techniques have been proposed using microfluidic to produce microgels by shearing dispersed phase with a continuous phase [Feng, Q., Li, Q. T., Wen, H. J., Chen, J. X., Liang, M. H., Huang, H. H., Lan, D. X., Dong, H., Cao, X. D., Injection and Self-Assembly of Bioinspired Stem Cell-Laden Gelatin/Hyaluronic Acid Hybrid Microgels Promote Cartilage Repair In Vivo. *Adv. Funct. Mater.* 2019, 29, 1906690. <https://doi.org/10.1002/adfm.201906690>; de Rutte, J. M., Koh, J., Di Carlo, D., Scalable High-Throughput Production of Modular Microgels for In Situ Assembly of Microporous Tissue Scaffolds. *Adv. Funct. Mater.* 2019, 29, 1900071. <https://doi.org/10.1002/adfm.201900071>].

Authors should cite these articles and compare the advantageous and disadvantageous between these techniques to highlight the strength of in situ printing microbeads rather than injecting/casting. As shown in Figure 2h, additional print layers result in deformation of the structure that resembles a pyramidal shape. The authors should evaluate the structural integrity of printed structure using methods as described in [Lee, J. M., and Yeong, W. Y. Engineering macroscale cell alignment through coordinated toolpath design using support-assisted 3D bioprinting. *Journal of the Royal Society Interface* 17.168 (2020): 20200294. <https://doi.org/10.1098/rsif.2020.0294>] to determine the integrity of the printed microbeads layer. The authors have to justify the need for printing these microbeads since previous works have shown that microgels can be delivered to the implant site through injection.

Thanks for the constructive comments on our submission.

First, the two papers included in the comment has been cited in revision:

22. Feng Q, et al. *Injection and self-assembly of bioinspired stem cell-laden gelatin/hyaluronic acid hybrid microgels promote cartilage repair in vivo. Adv. Funct. Mater.* 29, 1906690 (2019).

23. de Rutte JM, Koh J, Di Carlo D. *Scalable high-throughput production of modular microgels for in situ assembly of microporous tissue scaffolds. Adv. Funct. Mater.* 29, 1900071 (2019).

Second, compared with the microbead injection technique proposed in the papers above, the bead-jet printing system possesses many advantages, including on-demand patterning of building blocks, i.e., beads, personalized treatment, and enlarged therapeutic area/volume when the beads are sparsely patterned, etc.

a) The bead-jet printing system is capable of printing different structures, in either homogeneous or heterogeneous arrangement of microbeads. Most natural organs are composed of heterogeneous organization of cells and the extracellular components. Therefore, heterogeneous printing can offer a better solution to mimic tissue microenvironment, by patterning multiple types of cells and tissue elements within one engineered architecture. Some of the homogeneous and heterogeneous structures are presented below by tailored printing (**Figure R1**).

Figure R1. (left) Homogenous structures of printed gelatin beads, including a THU alphabetical sparse pattern, and rectangular, triangular and hexagonal dense patterns. (right) Heterogeneous structures of printed gelatin beads of two different colours. Scale bar, 5 mm.

b) The bead-jet printing platform can distribute beads into large-scale and defined patterns. For planar printing, the printed pattern could cover nearly 100 cm² in this current in-house machine. The printing scale could be further increased by improved the locomotion ability, i.e., the *x-y* movement range, of the printing nozzle (**Figure R2**). For three-dimensional (3D) patterning, the bead-jet printing could fabricate up to 2 cm × 2 cm × 3 cm structures using the current in-house printer (**Figure R3**).

Figure R2. Printing of microbeads into (a) a 12-cm long straight line, (b) a rectangular shape (10 cm \times 1 cm), (c) a square shape (3 cm \times 3 cm), and (d-f) large-scale shapes (d, 5 cm \times 7 cm; e, 8 cm \times 12 cm; f, 9 cm \times 9 cm).

Figure R3. Printing of 3D heterogeneous structures. **a**, Schematic of heterogeneous printing. **b**, 3D printed heterogeneous structures of gelatin beads on sacrificial gelatin support. The printed assembly had up to 50 layers. Scale bar, 1 cm.

c) Comparison with other bead-printing technique:

The throughput of traditional microbead printing techniques is very low. For example, the aspiration-assisted bioprinting, reported in *Science Advances* in 2021¹, took nearly 30 s to process one microbead. Our bead-jet printing platform only takes approximately 1 s to print one microbead, nearly satisfying the requirement for large-scale printing in either continuous printing or on-demand printing manners.

d) Characterization of structural stability:

We deeply appreciate the comment on examining the robustness and integrity of printed structures. The comment suggested us to follow the method described in [Lee, J. M., and Yeong, W. Y. Engineering macroscale cell alignment through coordinated toolpath design using support-assisted 3D bioprinting. *Journal of the Royal Society Interface* 17.168 (2020): 20200294. <https://doi.org/10.1098/rsif.2020.0294>]. However, this method required grid structures to be embedded in the scaffold, which cannot be copied in our study, because the bead-printing in our work was conducted in bead-only manners, i.e., no extra structural support and adaptive to printing-on-irregular substrates. Therefore, we adopted other mechanical interruption methods to evaluate the structural integrity, including tweezer-clamping, shaking in PBS buffer, and phase-transfer of printed planar patterns (single layer and multiple layers) and 3D cube structures. After mechanical interruption, all structures remained intact, verifying the integrity and robustness of printed structures (**Figure R4**). The stability was attributed to the bead-bead interface fusion that glued the building blocks (**Figure R5**). The interface fusion was formed when two beads were in contact before fully gelled.

Figure R4. Evaluation of structural integrity of printed architectures. **a**, Clamping of planar structure (single layer). **b**, Planar structure (single layer) immersed in PBS buffer. **c**, Planar structure (multiple layers) immersed in PBS buffer. **d**, 3D cube structure transferred from a 10-cm culture dish to a beaker. **e**, A 3D cube structure immersed in PBS buffer in a beaker. Scale bar, 1cm.

Figure R5. Mono-layer printed gelatin beads preserved the bead-to-bead interface and formed bead-bonds on a sacrificial gelatin support. Scale bar, 200 μm .

Specific comments

- How were the microbeads loaded onto the rheometer for measurement? How did the authors ensure complete contact of microbeads between the parallel plates? Can the authors explain the observation of $G'' > G'$ at lower shear strain % for microbeads? Also, can the author clarify whether 10 Hz or 10 rad/s was used for the strain sweep test?

In the first submission, the details of the rheological test were listed as below:

To evaluate the rheological property difference of bulk gel and printed beads as a patterned bulk assembly, 5% (w/v) gelatin crosslinked by TG was used throughout this study. The hydrogels were evaluated by a rotational rheometer (MCR302, Anton Paar, Austria). A measuring plate PP08 with a diameter of 8 mm was used in all measurements. Oscillatory strain sweeps (0.1 to 1%) were carried out at room temperature using a frequency of 10 Hz. Oscillatory angular frequency sweeps (0.1-25 rad/s) were performed at room temperature using a strain of 0.1%.

The printed microbeads assembly were used and loaded onto the rheometer. However, during revision, we found there were shortcomings of this method. The packing degree of printed microbeads assembly is quite low, that couldn't satisfy the measurement requirement of rheometer. Therefore, we re-performed the rheological test using both gelatin and Matrigel according to the protocol as described in the literature²⁻³. Frequency sweep, strain sweep, time sweep and shear ramp measurements were performed to evaluate the rheological properties of granular microgels and bulk gels (**Figure R6**).

The details of the new rheological test are updated as below:

Rheology test

To evaluate the rheological property difference of bulk gel and microbeads, 5% (w/v) TG-crosslinked gelatin and Matrigel was used throughout this study and tested by a rotational rheometer (MCR302, Anton Paar, Austria). Briefly, the solidified microbeads were collected in the cubes. Then the microbeads were pelleted in the tube and centrifuged at 1000 rpm for 5 min. After that, the oil was aspirated and removed from the tube. The gel-oil complex was washed, centrifuged and aspirated for multiple steps until all oil droplets were removed. In the rheological test, a 25-mm measuring plate PP025 was used in this study. An approximate volume of microbead assembly was placed onto the center of the rheometer plate using a spatula. Then the geometry was lowered to the desired gap height (1 mm used in this study). Redundant microbeads were removed using a scraper. Oscillatory strain sweeps (0.1 to 500%, 1 Hz) were carried out at room temperature. Oscillatory frequency sweeps (0.1-100 Hz, 1% strain) were performed at room temperature. Time sweeps (0.5% strain, 1 Hz, 2 min) were performed at room temperature to determine the storage (G' , Pa) and loss (G'' , Pa) moduli of microbead assemblies and bulk gel. Shear ramp (0.01-100 s^{-1}) were used to study the relationship between viscosity and shear rate. All experiments were repeated at least three times.

Figure R6. Procedures of the rheological test of microbead assembly. a, Schematic of microbeads assembly preparation for the rheological test. **b,** Experimental procedures of the rheological test of microbead assembly.

Figure R7. Rheological characterization of gelatin and Matrigel granular and bulk gels. a, Frequency sweep performed from 0.1 to 100 Hz of gelatin (i) and Matrigel (ii) granular and bulk gels. **b,** Shear-thinning properties of gelatin (i) and Matrigel (ii) granular and bulk gels. **c,** Strain sweep performed from 0.1 to 500% of gelatin (i) and Matrigel (ii) granular and bulk gels. **d,** Yield

strains of gelatin (i) and Matrigel (ii) granular and bulky hydrogel. e, Storage modulus G' of gelatin (i) and Matrigel (ii) granular and bulk gels. (** $P < 0.01$)

Frequency sweep showed that storage modulus of gelatin and Matrigel microbead assembly and their bulk counterparts at low frequency exhibits linear trends (**Figure R7a**). Shear-thinning behaviors were also observed across four groups by reduction in viscosity with increasing shear rates (**Figure R7b**). Strain sweep test showed that all samples yielded ($G'' > G'$) at high strains (**Figure R7c**). The yield strain of bulk gelatin was significantly higher than the granular gelatin (**Figure R7d, i**). However, no significant difference of yield strains of Matrigel between its bulk and granular configurations was observed (**Figure R7d, ii**). There was no significant difference of storage moduli of gelatin and Matrigel in their granular and bulk configurations (**Figure R7e**).

- It is advisable for the authors to elaborate on the details regarding the animal model. How did the authors determine the amount of cells to be delivered at the wound site? with the animal model study, what was number of cells and volume of materials loaded in the cast group?

Basically, the number of cells delivered at each wound site was determined according to the defect size (**Figure R8**). As the scheme below shows, the total cell number delivered to defect is calculated based on LDB group since there is no need to include other acellular materials. The number of acellular beads and volume of acellular bulk gel are calculated according to the cell density and cell numbers loaded in microsphere.

Figure R8. HDG-MSCs, LDB-MSCs, HDB-MSCs are printed or casted to fill the defect volume.

Cells in the microsphere has been calculated according to the volume and listed as following.

Table R1. Number of cells in microbeads and in bulk gel

Hydrogel formation	HDG	LDB	HDB
Cell density	1×10^7 cells/mL	2×10^6 cells/mL	1×10^7 cells/mL
Cells / bead volume	1×10^4 cells/ μ L	300 cells/bead	1500 cells/bead

For the mouse VML defect models, 60% volume (~ 75 mg, 50 mm^3) of gluteus maximus (GM) muscle tissue was removed. Afterward, 300 LDBs were printed to fill in this defect volume. To

keep the number of cells consistent across these groups, 60 HDBs and 240 acellular beads, 9 μL HDGs and 36 μL acellular gel were printed or casted. Acellular beads and acellular gel were used to fill in the gap between the cell-laden scaffold and the native tissue. The total number of delivered cells and the defect sizes were kept consistent across all groups.

For the skin wound healing models, different size of skin wounds, including 5 mm \times 5 mm, 10 mm \times 10 mm and 15 mm \times 15 mm, were created. The number of cells and microbeads, and the volume of bulk gel are provided in the table below.

Table R2. Number of cells delivered at the wound sites

Tissue types	Volume and size	Total cell number	HDG group	LDB group	HDB group
VML GM tissues	60% volume (~75mg, 45 μl)	9×10^4 cells	9 μL HDGs, 36 μl acellular gel	300 LDBs	60 HDBs, 240 acellular beads
Skin wounds	5 mm \times 5 mm	2.1×10^4 cells	2.1 μL HDGs, 8.4 μL acellular gel	70 LDBs	14 HDBs, 56 acellular beads
	10 mm \times 10 mm	8.4×10^4 cells	8.4 μL HDGs, 33.6 μl acellular gel	280 LDBs	56 HDBs, 224 acellular beads
	15 mm \times 15 mm	1.8×10^5 cells	18 μL HDGs, 72 μl acellular gel	600 LDBs	120 HDBs, 480 acellular beads

Reviewer #2

"Bead-Jet" printing of high-density mesenchymal stem cell-laden Matrigel beads augments skeletal muscle and hair follicle regeneration

The authors propose a bead jet printer to print high and low density MSC loaded Matrigel beads in order to effectively improve regenerative outcomes of skeletal muscle and hair follicles. The proposed system was tested both in-vitro and in-vivo and shows good promise in causing successful regeneration of intended tissue. Authors have also shown that the encapsulation of MSCs into the Matrigel beads using the novel bead jet printing technology does not affect the extracellular vesicle production capabilities of the MSC and might still maintain its therapeutic potential. Bead printing parameters were first set using gelatin as the bioink and viability of printed MSC loaded beads was assessed. Using these conditions, various combination of MSC loaded Matrigel beads (high density vs low density) as well as bulk gel formulations were tested for their viability and EV formation in-vitro. The authors also went a step further and tested the viability of NIH3T3 fibroblasts and SW620 when bioprinted. In-vivo studies in mice showed that printed microbeads were able to cause functional restoration of muscle mass even after 60% removal as evidenced by H&E and MTS staining for collagen and fibrosis followed by RNA-seq analysis of retrieved tissue. Ability of the newly regenerated muscle fibers to undergo skeletal integration was assessed by IHC for desmin (myofiber marker), alpha-SMA (artery development), CD31 (endothelial marker), neurofilament and β IIIIT (neural integration). While the presented study addresses the important consideration of spatial patterning of MSCs required to ensure successful regeneration, there are several gaps in the experimental studies that are outlined below. Overall, this study provides a good foundation to the application of bead printing technology to ensure spatial patterning of MSCs in order to improve the regenerative success in case of irregular sized large defects.

Accept with revisions

Major Concerns:

1) The lab has previously published an article in Advanced biology titled "Injectable Mesenchymal Stem Cell-Laden Matrigel Microspheres for Endometrium Repair and Regeneration" using the same bead composition and similar technology. It is unclear what the difference between this printer and the microfluidic device used in that article is. Can the authors elaborate on the advantage of a printer over the microfluidic device? In previous work the simpler setup does work equally well and is more easily adaptable to customizations.

The author appreciates the reviewer for raising this question. The bead-jet printing platform gains lots of advantages over traditional microfluidic device or microsphere injection method, including printing speed, types of printed structures, printing coverage area and so on.

a) The bead-jet printing system is capable of printing different structures, in either homogeneous or heterogeneous arrangement of microbeads. Most natural organs are composed of heterogeneous organization of cells and the extracellular components. Therefore, heterogeneous printing can

offer a better solution to mimic tissue microenvironment, by patterning multiple types of cells and tissue elements within one engineered architecture. Some of the homogeneous and heterogeneous structures are presented below by tailored printing (**Figure R9**).

Figure R9. (left) Homogeneous structures of printed gelatin beads, including a THU alphabetical sparse pattern, and rectangular, triangular and hexagonal dense patterns. (right) Heterogeneous structures of printed gelatin beads of two different colours. Scale bar, 5 mm.

b) The bead-jet printing platform can distribute beads into large-scale and defined patterns. For planar printing, the printed pattern could cover nearly 100 cm² in this current in-house machine. The printing scale could be further increased by improved the locomotion ability, i.e., the *x-y* movement range, of the printing nozzle (**Figure R9**). For three-dimensional (3D) patterning, the bead-jet printing could fabricate up to 2 cm × 2 cm × 3 cm structures using the current in-house printer (**Figure R10**).

Figure R10. Printing of microbeads into (a) a 12-cm long straight line, (b) a rectangular shape (10 cm × 1 cm), (c) a square shape (3 cm × 3 cm), and (d-f) large-scale shapes (d, 5 cm × 7 cm; e, 8 cm × 12 cm; f, 9 cm × 9 cm).

Figure R11. Printing of 3D heterogeneous structures. **a**, Schematic of heterogeneous printing. **b**, 3D printed heterogeneous structures of gelatin beads on sacrificial gelatin support. The printed assembly had up to 50 layers. Scale bar, 1 cm.

c) Comparison with other bead-printing technique:

The throughput of traditional microbead printing techniques is very low. For example, the aspiration-assisted bioprinting, reported in Science Advances in 2021¹, took nearly 30 s to process one microbead. Our bead-jet printing platform only takes approximately 1 s to print one microbead, nearly satisfying the requirement for large-scale printing in either continuous printing or on-demand printing manners.

2) What is the size limit of the injury that can be filled with the MSC loaded beads using bead jet printing while still maintaining sterility and successful regeneration?

Thanks for raising this question. We have tested our bead-jet printing platform on large-scale mice skin wounds, including 10 mm × 10 mm (~15% removal of dorsal skin) wounds and 15 mm × 15 mm (~35% removal of dorsal skin) wounds. The results as below indicate that the bead-jet printing system could repair nearly 35% removal of skin tissues (**Figures R12, R13**).

To investigate the capacity of bead-jet printing, the patterning was performed on planar substrates. the in-house bead-jet printing machine could produce 30 mm × 30 mm square and 150 mm × 10 mm rectangular structures. It suggested that this platform could be applied for regenerative therapy of large-scale injury.

Figure R12. Matrigel HDB-MSCs augment skin wound (10 mm × 10 mm) healing and hair follicle regeneration. **a-b**, The photographs of wound closure area (**a**) and quantification (**b**) at day 0 (D0), D3, D7, D10, D14, and D21, respectively. $n = 3$ at each time point. Scale bar, 1 cm. **c**, Representative images of H&E staining and MTS of regenerated skin tissues at 4 weeks post-printing. Scale bar, 500 μm. **d-f**, Quantification of total skin thickness (**d**), hair follicle (**e**), and sebaceous gland counts (**f**) per section in the regenerated area. $n = 3$. * $p < 0.05$. $n = 3$ for each group. The data are represented as mean ± SD. Significant difference is determined by unpaired two-tailed student t-test. * $P < 0.05$.

Figure R13. Matrigel HDB-MSCs augment LARGE skin wound (15 mm × 15 mm) healing and hair follicle regeneration. a-b, The photographs of wound closure area (a) and quantification (b) at D0, D3, D7, D10, D14 and D21, respectively. n = 3 at each time point. Scale bar, 1 cm. c, Representative images of H&E staining and MTS of regenerated skin tissues at 4 weeks post-printing. Scale bars, 500 µm. d-f, Quantification of total skin thickness (d), hair follicle (e), and sebaceous gland counts (f) per section in the regenerated area. n = 3. n = 3 for each group. The data are represented as mean ± SD. Significant difference is determined by unpaired two-tailed student t-test. *P < 0.05.T

For large-scale skin wound healing, Matrigel HDB-MSCs exhibited significantly better regeneration ability than the acellular group. Significantly higher amount of hair follicle and sebaceous glands were observed in the Matrigel HDB-MSCs group, as regenerative treatment of both 10 mm × 10 mm, 15 mm × 15 mm skin wounds. It indicated that Matrigel HDB-MSCs could accelerate hair follicle formation.

3) The distinction between the LDB, HDB and HDG is still unclear given that the initial cell seeding density is the same. Can the authors clarify the logic behind this further? What is the number of MSCs loaded per microsphere for each of these bead groups since that will also affect regeneration? MSCs per sphere or MSCs per unit volume needs to be calculated since it will affect

in-vitro and in-vivo outcomes.

The cell density of HDB and HDG was identical, and both were 5-fold higher than the density of LDB.

The number of cells in the microsphere has been calculated according to the volume and listed as following.

Table R3. Number of cells in microbeads and in bulk gel

Hydrogel formation	HDG	LDB	HDB
Cell density	1×10^7 cells/mL	2×10^6 cells/mL	1×10^7 cells/mL
Cells / bead volume	1×10^4 cells/ μ L	300 cells/bead	1500 cells/bead

For the *in vitro* study, to keep the cell seeding number consistent at the beginning when cultured in the 12 well plate, 50 HDBs, 250 LDBs, and 7.5 HDG were printed or casted into the culture medium. The total cell number across the three groups were 7.5×10^4 cells at day 0.

For the mouse VML defect models, 60% volume (~ 75 mg, 50 mm^3) of gluteus maximus (GM) muscle tissue was removed. Afterward, 300 LDBs were printed to fill in this defect volume. To keep the number of cells consistent across these groups, 60 HDBs and 240 acellular beads, $9 \mu\text{L}$ HDGs and $36 \mu\text{L}$ acellular gel were printed or casted. Acellular beads and acellular gel were used to fill in the gap between the cell-laden scaffold and the native tissue. The total number of delivered cells and the defect sizes were kept consistent across all groups.

For the skin wound healing models, different size of skin wounds, including $5 \text{ mm} \times 5 \text{ mm}$, $10 \text{ mm} \times 10 \text{ mm}$ and $15 \text{ mm} \times 15 \text{ mm}$, were created. The number of cells and microbeads, and the volume of bulk gel are provided in the table below.

Table R4. Number of cells delivered at the wound sites

Tissue types	Volume and size	Total cell number	HDG group	LDB group	HDB group
VML GM tissues	60% volume (~ 75 mg, $45 \mu\text{l}$)	9×10^4 cells	$9 \mu\text{L}$ HDGs, $36 \mu\text{l}$ acellular gel	300 LDBs	60 HDBs, 240 acellular beads
Skin wounds	$5 \text{ mm} \times 5 \text{ mm}$	2.1×10^4 cells	$2.1 \mu\text{L}$ HDGs, $8.4 \mu\text{L}$ acellular gel	70 LDBs	14 HDBs, 56 acellular beads
	$10 \text{ mm} \times 10 \text{ mm}$	8.4×10^4 cells	$8.4 \mu\text{L}$ HDGs, $33.6 \mu\text{L}$ acellular gel	280 LDBs	56 HDBs, 224 acellular beads
	$15 \text{ mm} \times 15 \text{ mm}$	1.8×10^5 cells	$18 \mu\text{L}$ HDGs,	600 LDBs	120 HDBs,

	cells	72 μ l acellular gel	480 acellular beads
--	-------	--------------------------	---------------------

4) The authors talk about gel bead assemblies maintaining structural integrity at low shear, but being prone to deformation under increased shear. While advantageous for in terms of motion adaptation, wouldn't it result in increased degradation of beads leading to sub-optimal performance? Some larger sized defects may require longer residence of cells.

Thanks for raising this concern. The degradation of non-cellular components in regenerative therapy is a major concern in designing therapeutic agents and their configurations. To address this comment, the stability of Matrigel in both bead-assembly and bulk gel were compared over 1 week when immersed in DMEM medium and incubated at 37 °C (**Figure R14**). As the picture below shows, the edge of Matrigel beads remained clear and sharp in both the bead group and the bulk group, indicating that the bead configuration did not accelerate Matrigel degradation within 1 week. However, comprehensive understanding of Matrigel degradation remains challenging, especially in cell-presence manners (to mimic in-vivo environment) and over long term.

Figure R14. Degradation test of bulk Matrigel and acellular Matrigel beads in vitro at D1, D3 and D7. Scale bar, 200 μ m.

In addition, we further investigated the retention, growth and distribution of MSCs in regenerated muscle tissue after transplantation (**Figure R15**). The results proved that the morphology and organization of MSCs in the HDB group were clear to observe compared with MSCs in the HDG or LDB group after 1 week post implantation. Significantly higher proportion of MSCs were observed in the HDB groups than in the LDB and HDG groups (**Figure R15, i**). The *in vivo* result also proved that cell-laden beads (the bead configuration) did not decrease the time of residence of cells compared with the bulk configuration. Instead, MSCs in the HDB group may have gained significantly higher proliferation rates, suggested by the higher density and thickness of MSCs fluorescence at 1-week.

Figure R15. Representative images of MSCs in the Matrigel HDG, LDB and HDB groups after transplantation for 24 h (g) and 1 week (h) in muscle tissues. MSCs were labeled using the red Dil cell membrane tracker (Meilunbio). Acellular beads and acellular bulk gel were loaded with monodispersed green fluorescent polystyrene microspheres (Aladdin, 1 μm , 1:20 diluted in Matrigel solution). Quantification of MSCs (i) and total cells density (j) per section after transplantation for 24 h and 1 week. Scale bar, 100 μm .

5) The initial cell seeding density of HDG, LDB and HDB seems to be very high, lowest being 2×10^6 cells in LDB. What is the average number of cells per bead? The high cell number could be causing metabolic stress on the MSCs leading to altered functions. Normalization of cells/bead based on DNA content or number of nuclei needs to be done.

Cells in the microsphere has been calculated according to the volume and listed as following.

Table R5. Number of cells in microbeads and in bulk gel

Hydrogel formation	HDG	LDB	HDB
Cell density	1×10^7 cells/mL	2×10^6 cells/mL	1×10^7 cells/mL
Cells / bead volume	1×10^4 cells/ μL	300 cells/bead	1500 cells/bead

For the HDB groups, there was nearly 1500 cells per microbead. For the LDB group, there were nearly 300 cells per microbead. The initial cell number delivered to different experimental groups were identical. For the in vitro study, to keep the cell seeding number consistent at the beginning when cultured in the 12 well plate, 50 HDBs, 250 LDBs, and 7.5 HDG were printed or casted into the culture medium. The total cell number across the three groups were 7.5×10^4 cells at day 0.

The suggestion on normalizing cell number with the DNA content is highly appreciated. We calculated the cell number and measured the DNA content by dissociating cells from Matrigel using collagenase type I (Figure R16). At the initial stage, cell numbers and DNA content were nearly identical across these groups. However, at the later stages (D7 and D14), the cell numbers and DNA content were significantly varied, presenting higher values in the Matrigel HDB-MSCs

group compared with the other two groups. A 3-fold increase of live cell number was achieved in the Matrigel HDB group after 1 week, which was significantly higher than the Matrigel LDB groups (87.2% increase) and the Matrigel HDG (48.6% increase). A 12-fold increase of live cell number was achieved in the Matrigel HDB group after 2 weeks, significantly higher than the Matrigel LDB groups (440% increase) and the Matrigel HDG (360% increase).

A 2-fold increase of DNA content was achieved in the Matrigel HDB group after 1 week, significantly higher than the Matrigel LDB groups (84.6% increase) and the Matrigel HDG (109% increase). A 27-fold increase of DNA content was achieved in the Matrigel HDB group after 1 week, significantly higher than the Matrigel LDB groups (670% increase) and the Matrigel HDG (385% increase).

Figure R16. Quantification of Cell number and DNA content of MSCs cultured in the Matrigel-HDG, LDB and HDB groups at D0, D3, D7 and D14. d, Live cell counts using hemocytometer. e, DNA content acquisition. n = 3 for each group. *p < 0.05; **p < 0.01; ***p < 0.001.

6) Because of the high cell density used, the MTT assay is always going to give high viability (False positive). Viability data needs to be tested for rigor with one more viability tests such LDH-CyQuant assay which allows for normalization with cell number/DNA content. Also, viability must be assessed over longer time points at least till Day 14.

As suggested by the comment, we performed the LDH-Cy Quant assay to test cell growth (**Figure R17**). Both the LDH-Cy Quant assay and live/dead staining were employed to compare different doses of MSCs till day 14, including 150, 300, 1500, 3000, 6000 cells per microbead. LDH-Cy Quant assay showed that high MSC density gained significantly higher proliferation rates and accelerated MSCs growth compared with low MSC density and MSCs in bulk gel.

Figure R17. a, The schematic illustration of MSCs growth in the HDG, LDB and HDB groups. MSCs in HDG and microbeads of different cell densities, including 150, 300, 1500, 3000, 6000 cells per microbead, were characterized at D0, D3, D7 and D14. Live and dead cells were stained with calcein-AM (green) and propidium iodide (red), respectively. The initial cell numbers were kept consistent among all groups. **b**, OD values of MSCs growth at 490 nm in Matrigel HDG, and in Matrigel beads comprising different cell densities at day 1 (D1), D3, D7 and D14, determined using the LDH-Cy Quant assay. **c**, Metabolic activity of MSCs at D1, D3, D7 and D14, determined using the Alamar Blue assay. * $p < 0.05$; ** $p < 0.01$; *** $p < 0.001$; **** $p < 0.0001$.

We also quantified cell viability across Matrigel HDG, and Matrigel beads comprising different cell densities using the Calcein-AM/PI staining over long time. The result indicated that at extremely high cell encapsulation densities, e.g., 3000 and 6000 cell per bead, might decrease cell viability within the microsphere (**Figure R18**). It might be resulted from the limited access to the cell culture medium and higher metabolic stress in the microbeads. Therefore, our HDB group were chosen at 1500 cell per bead.

Figure R18. Quantification of MSCs viability in Matrigel HDG, and Matrigel beads comprising different MSCs densities at D1, D4, D7 and D14. n = 3. The data is represented as mean \pm SD. Significant difference is determined by one-way ANOVA, followed by Tukey's test. *p < 0.05; **p < 0.01; ***p < 0.001; ****p < 0.0001.

7) Characterization of the EV cargo needs to be performed to ensure that the MSC derived EVs produced after encapsulation in Matrigel beads still possess therapeutic anti-inflammatory potential and are not just producing EVs because of cell stress, which is a common occurrence.

To further verify that EVs derived by MSCs still possessed the therapeutic anti-inflammatory potential, we performed gene sequencing analysis of EVs derived from Matrigel HDG, Matrigel LDB, and Matrigel HDB groups and 2D-cultured MSC-EVs that were proven to possess therapeutic anti-inflammatory effects⁴⁻⁸ (**Figure R19**).

Genes related to anti-inflammatory activity were selected from the following publications.

Yu, Hui, et al. "Exosomes derived from hypertrophic cardiomyocytes induce inflammation in macrophages via miR-155 mediated MAPK pathway." *Frontiers in immunology* 11 (2021): 606045.

Ti, Dongdong, et al. "LPS-preconditioned mesenchymal stromal cells modify macrophage polarization for resolution of chronic inflammation via exosome-shuttled let-7b." *Journal of translational medicine* 13.1 (2015): 1-14.

Geiger, Adolf, Audrey Walker, and Erwin Nissen. "Human fibrocyte-derived exosomes accelerate wound healing in genetically diabetic mice." *Biochemical and biophysical research communications* 467.2 (2015): 303-309.

Zhao, Xige, et al. "Immunomodulation of MSCs and MSC-derived extracellular vesicles in osteoarthritis." *Frontiers in Bioengineering and Biotechnology* 8 (2020): 575057.

Figure R19. Heatmap of genes expression related to anti-inflammatory activities in EVs derived from 2D-culture, and the Matrigel HDG, Matrigel LDB, Matrigel HDB groups.

By comparing the gene expression of 2D EVs, and Matrigel HDG, Matrigel LDB, Matrigel HDB-EVs, we found there was significant difference on gene expression across these groups. Therefore, we believe Matrigel HDG, Matrigel LDB, Matrigel HDB-EVs possessed the equivalent potential of therapeutic anti-inflammatory abilities.

8) While IHC shows upregulation of markers for successful innervation, reendothelialization and vascularization of defects, the RNA-Seq data does not reflect the same. GO analysis needs to be performed again and specific terms related to angiogenesis, follicle development, ECM remodeling etc need to be highlighted that are relevant to the effect being seen. Upregulation of Ribosome related pathway is a very broad term and cannot be used since it shows up as one of the top regulated processes in any type of RNA-seq analysis with any cell/tissue type.

Thanks for the suggestion. We have performed the KEGG and GO biological process, cellular component, and molecular function analysis on the sequencing data of regenerated skin and muscle tissues. For skin tissues, the GO analysis showed that highly differentiated genes were enriched in terms related to skin development, skin morphogenesis, wound healing, endothelial cell migration, sprouting angiogenesis, extracellular matrix, hair follicle cell proliferation and keratinocyte migration (**Figures R20 and R21**). For muscle tissues, the GO analysis showed that highly differentiated genes were enriched in terms related to extracellular matrix organization, smooth muscle cell proliferation, axonogenesis, immune response, muscle cell differentiation, extracellular matrix assembly and skeletal muscle organ development (**Figures R22 and R23**). The analysis of highly differentiated genes reached agreement with the results from the IHC examination.

Figure R20. KEGG and GO analysis of regenerated skin tissues. 10-mm round full-thickness excisional wounds were regenerated by transplanting acellular beads (ACB) and MSCs-HDB.

Figure R21. KEGG and GO analysis of regenerated skin tissues. 15-mm rounded full-thickness excisional wounds were regenerated by transplanting acellular beads (ACB) and MSC-HDB.

Figure R22. KEGG and GO analysis of regenerated muscle tissues. Injured muscle tissues were repaired by transplanting MSCs in HDB and HDG.

Figure R23. KEGG and GO analysis of regenerated muscle tissue. Injured muscle tissues were repaired by transplanting MSCs in HDB or self-repaired (the non-treated group).

9) The errors in most of the bar graphs in Figure 8 are very large especially for HDB. How was the quantification done? Was normalization to total tissue area performed?

Thanks for this question. The bar difference among HDG-MSCs, LDB-MSCs, and HDB-MSCs may be attributed to the influence of surgical implementation condition and animal individual-variance.

To reduce the sample difference, we repeated six samples and created four 5-mm-diameter excisional skin wounds for transplantation, including non-treated, Matrigel HDG-MSCs, LDB-MSCs and HDB-MSCs groups. The HE and Masson staining of all samples were provided. (Figure R24)

Two sections from two different slides per sample were selected to decrease the measurement error. The skin thickness, and the counts of hair follicle and the sebaceous gland, were measured per section for each time. Then the average of statistical data from all sections and slides were used to plot the data. For immunostaining, at least 6 imaging windows were randomly selected

and the sample size for each antibody measurement was 6.

Some sample, e.g., Sample 2 in Figure R24, may present significant difference in appearance with others by non-biological reasons, causing high errors in plotted graphs.

Figure R24. The H&E and MTS staining of 6 skin tissue samples after self-healing (non-treated), or treated with Matrigel LDB-MSCs and HDB-MSCs at 3 weeks post-printing (or post-casting of Matrigel HDG-MSCs). Scale bar, 500 μm .

Minor Concerns:

1) Matrigel inherently has a lot of batch-to-batch variation resulting in poor control over bead micro-architecture. How does the bead jet printing address this issue?

The author thanks the reviewer raising this question. The batch-to-batch variation of Matrigel is the inherent properties that couldn't be solved in this work. The rational design for choosing Matrigel as the scaffolding material is given below:

a) There are multiple types of growth factor secreted by cell including transforming growth factor (TGF- β), insulin like growth factor (IGF) and fibroblast growth factor (FGF), and cytokines

within Matrigel that could facilitate cell migration, proliferation and angiogenesis, inducing the integration of printed or casted scaffolds and native tissues⁹⁻¹⁰.

b) The mechanical properties, including both viscoelasticity and porosity, of Matrigel are close to the decellularized skin extracellular matrix. The could provide ideal growth environment for MSCs, in particular, nature-like mechanotransduction cues to regulate MSCs behaviours¹¹. Synthetic or hybrid gels cannot fully replace Matrigel in providing appropriate mechano-regulatory cues as Matrigel³⁴.

c) Matrigel presents suitable rheological properties for our customized bead-jet printing system. It undergoes gelation at body temperature and the bead-assembly shows shear-thinning properties. (Figure R25) Matrigel could be gelled before implantation and there is no any further crosslinker required¹².

Figure R25. Rheological characterization of gelatin and Matrigel granular and bulk gels. a, Frequency sweep performed from 0.1 to 100 Hz of gelatin (i) and Matrigel (ii) granular and bulk gels. **b,** Shear-thinning properties of gelatin (i) and Matrigel (ii) granular and bulk gels. **c,** Strain sweep performed from 0.1 to 500% of gelatin (i) and Matrigel (ii) granular and bulk gels. **d,** Yield strains of gelatin (i) and Matrigel (ii) granular and bulky hydrogel. **e,** Storage modulus G' of gelatin (i) and Matrigel (ii) granular and bulk gels. (** $P < 0.01$)

d) Matrigel is particularly well-suited for the *ex vivo* culture and study of epithelial cells that typically interact with a basement membrane *in vivo*. It may provide bioactive cues that are not naturally found in the microenvironment of other progenitor and tissue-resident cells¹³.

Overall, the main reason for choosing Matrigel was its favorable support for cell proliferation and migration. However, there are also some inherent shortcomings of Matrigel, including the batch-to-batch variation, the concerns over use for clinical and translational research its

xenogenic origin¹⁴.

For future clinical translation of this strategy, synthetic biomaterials with high biocompatibility and comparable or tailorable mechanical and apathogenic properties should be developed to provide appropriate alternatives to Matrigel.

2) Matrigel is known to cause malignant transformation of certain cell types, like fibroblasts. Have any assays been done to check this doesn't happen?

We evaluated genes expression related to malignant transformation of cancer genes in regenerated muscle tissue by transplanting MSCs in HDG, LDB, HDB or self-repaired (the non-treat group). Compared with the normal group, i.e., the healthy tissue, the genes expressions were not showing significance in difference among the experimental and control groups. Therefore, we might draw the conclusion that malignant transformation of certain cell types didn't happen in regenerated muscle tissues. (Figure R26)

Figure R26. Heatmap of genes expression related to malignant transformation of RNA-seq of regenerated muscle tissues. Injured muscle tissues were repaired by transplanting MSCs in HDG, LDB, HDB or self-repaired, i.e., the non-treat group.

3) More TEM images of EVs should be analyzed for size distribution, at least 15-20.

Thanks for this suggestion. Size distribution of EV was calculated based on nanoparticle tracking

analysis (NTA) using the NanoSight NS300 instrument (Malvern, Worcestershire, UK) with a scientific CMOS sensor. This chart was generated by the software automatically. (Figure R27)

Figure R27. Size distribution of EVs based on the nanoparticle tracking analysis (NTA) using NanoSight NS300 instrument with a scientific CMOS sensor.

Following the review comment, we also measured the EV particle size distribution and plotted curves based on the TEM images. The result is in agreement with the NTA result. (Figure R28)

Figure R28. Transmission electron microscopy (TEM) images and corresponding particle size distribution curves for HDB-EVs (a), HDG-EVs (b) and LDB-EVs (c).

Reviewer #3

This manuscript by Cao et al. reports a bead-jet printing technique that deposits cell-laden and acellular Matrigel beads directly onto wound sites. This study is motivated by a need to optimize cell-based therapies for tissue regeneration by using biomaterials and biofabrication approaches to enhance cell function. For translational applications, an intra-operative technique is desirable and one that allows deposition of therapeutic material in a scalable and high-throughput manner even more so. Cao et al. report that MSCs encapsulated at a high density within Matrigel beads and deposited in a pre-defined pattern stimulate muscle and skin regeneration. They attribute these outcomes to the granular nature of their beads, the specific pattern in which they are printed, MSCs EV production and other behaviors including migration, proliferation, and trans-differentiation.

Although this manuscript aimed to tackle an important topic in the field, there are many scientific and technical concerns around claims made about the used materials, printing technique, biological outcomes observed, and conclusions derived. The authors make numerous conjectures while discussing their data without performing the experiments necessary to rigorously test their hypothesis.

Major concerns are listed below for the authors' consideration.

Engineering limitations/concerns:

1. The use of Matrigel dampens any biomaterials-related enthusiasm for this printing technique and outcomes described. Out of the many synthetic and naturally derived biomaterials available and in use, Matrigel is perhaps one that has the most limited future potential in terms of translational applications. The field has recently focused on producing designer synthetic matrices and moving away from Matrigel - despite its historical use and utility – due to well-known issues centering around batch to batch variability (see Nguyen et al. Nat. Biomed. Eng. 2017), xenogeneic origin, and inability to tune biophysical and biochemical properties. None of this is discussed. More alarmingly, the authors cite two of their own papers to show the applicability of Matrigel in tissue engineering applications. The authors should provide a rationale for using Matrigel as their cell-encapsulating material and explore tunability of its properties and related cell outcomes. Availability of Matrigel is also a concern not addressed by authors – Matrigel is currently backordered in commercial supplier stocks even for lab-based use. How can this material be translated into a commercial product for use in patients?

Thanks for raising this critical concern. We definitely agree with the review comment that Matrigel cannot be directly applied in regenerative medicine for clinical trials, but for technology development, Matrigel is still a good candidate to test new principles, such as sparse patterning of crowded MSCs in viscoelastic scaffold with extremely low stiffness.

To be brief, we used Matrigel in this study because it has the optimized properties to promote MSCs growth and behave. But meanwhile, we also have the plan to replace Matrigel in the future

development when this technology is to be translated. However, up to now, there has been no commercial supply of Matrigel replacement that overcome its limitation but not sacrificing its strength in supporting cell growth and behavior.

The composition of Matrigel is similar to ECM for its basement membrane-like nature, thus it was selected as the biomaterial in this research. Matthias Lutolf et al. noted that Matrigel has advantages over artificial biomaterials in directing stem-cell rate³⁵. The growth support of Matrigel on dermal papilla and sweat glands has also been verified in previous studies³⁶. The interaction of skin-derived stem cells with Matrigel acts in concert to give rise to a series of spatially and temporally coordinated events that regulate the stem-cells fate, specifically, directional differentiation into epidermis and dermis with necessary appendages.

Matrigel contains several major ECM proteins and specific amino acid sequences among them provide multiple adhesion sites for the attachment of stem cells and promote differentiation and angiogenesis, thereby beneficial for MSCs¹¹. Also, there are multiple types of growth factor secreted by cells, including transforming growth factor (TGF- β), insulin like growth factor (IGF) and fibroblast growth factor (FGF), and cytokines within Matrigel that could facilitate cell migration, proliferation and angiogenesis, inducing the integration of printed or casted scaffolds and native tissue^{9, 10}.

Additionally, Matrigel presents suitable rheological properties for our customized bead-jet printing technique (**Figure R25**). It undergoes gelation at body temperature and shows shear-thinning properties. Matrigel could be gelled before implantation and there is no any further crosslinking reactions required as in photo- or chemical-crosslinking gels¹².

Moreover, Matrigel is particularly well-suited for the *ex vivo* culture and study of epithelial and mesenchymal cells that typically interact with a basement membrane and extracellular scaffold *in vivo*. It may provide bioactive cues that are not provided in other synthetic or engineered hydrogels¹³.

Figure R25. Rheological characterization of gelatin and Matrigel granular and bulk gels. a, Frequency sweep performed from 0.1 to 100 Hz of gelatin (i) and Matrigel (ii) granular and bulk gels. **b,** Shear-thinning properties of gelatin (i) and Matrigel (ii) granular and bulk gels. **c,** Strain sweep performed from 0.1 to 500% of gelatin (i) and Matrigel (ii) granular and bulk gels. **d,** Yield strains of gelatin (i) and Matrigel (ii) granular and bulky hydrogel. **e,** Storage modulus G' of gelatin (i) and Matrigel (ii) granular and bulk gels. (**P < 0.01)

The main reason for choosing Matrigel was its favorable support for cell proliferation and migration, compared with the commercially available materials, e.g., alginate, collagen, HAMA, gelatin. However, there are also some inherent shortcomings of Matrigel, including the batch-to-batch variation, the concerns over use for clinical and translational research its xenogeneic origin¹⁴.

The shortcoming of Matrigel has also been discussed in the manuscript and listed as following:

However, though the cytocompatibility of Matrigel is favorable, the clinical promise is limited due to its tumor-derived, ill-defined, variable composition and low availability for lab-used condition⁵⁴⁻⁵⁵. The potential for antigenicity is one of the inherent limitations.

For future clinical translation of this strategy, synthetic biomaterials with high biocompatibility, cell-ligation ability, biodegradability and appropriate viscoelastic and apathogenic properties should be innovated and commercialized to provide appropriate alternatives to Matrigel.

The cited articles are as below:

54. Nguyen EH, et al. Versatile synthetic alternatives to Matrigel for vascular toxicity screening and stem cell expansion. Nat. Biomed. Eng. 1, 1-14 (2017).

55. Isenberg EA, Murphy WL. Synthetic alternatives to Matrigel. Nat. Rev. Mater 5, 539-551 (2020).

In addition, the purpose of this research is to propose the concept of microbead in situ bioprinting for functional skin/muscle repair and provide inspiration for the research of intraoperative 3D bioprinting. Thus, the primary demand for biomaterials should be the most favorable biocompatibility, which led us to choose Matrigel. Synthetic biomaterials with high biocompatibility and stable mechanical and apathogenic properties would be developed as replacement to Matrigel, and the "bead-jet printing" strategy would be transferred to the new gel.

2. For broad applicability to the field, the authors should demonstrate that their bioprinting technique is compatible with other soft polymer materials that have a higher chance of translation or that are already being used in pre-clinical and clinical studies e.g., PEG, alginate, and hyaluronic acid.

The author thanks for this suggestion.

HAMA is a viscoelastic material widely used to encapsulate cells in 3D. It suits the customization of our bead-jet printing system, whereas other materials have been proved with good performance in shape-manufacturing using established techniques. Therefore, HAMA was chosen as a new scaffolding material and multiple tests have been performed (**Figure R26**). 5% HAMA was

crosslinked using 0.25% LAP solution as the UV crosslinker. Live/dead staining showed that cell did not migrate out of the microbeads or spread out. Cell viability test was performed MSCs in HAMA microbeads and bulk gel. Over 90% cell viability rates and microbead configuration were maintained at day 0, suggesting the suitability of our technique in HAMA bead formulation and manipulation. However, due to the poor biocompatibility of HAMA, e.g., lack of peptide sequences for cell ligation and MMP-degradation, cells could not grow, expand and morphogenesis in HAMA gel or beads. It also resulted in decreased cell viability rates. Among the groups, LDB-MSCs had the lowest viability at day 7. Based on the molecular properties of alginate and PEG, we would expect a similar result in the growth and behaviors of encapsulated cells.

Significantly lower proportion of EdU positive cells appeared in the HAMA LDB group compared with the other groups. Both live/dead staining and EdU test elucidated that low cell density might be detrimental to cell growth

Metabolic activity of MSCs was also investigated. Cell metabolic activity in the HAMA LDB group was significantly lower at day 5 and day 7 compared with the HDG and HDB groups.

Further, compared with the metabolic activity of MSCs in HAMA, the metabolic activity of MSCs in Matrigel beads or bulk was significantly higher. (Figures R26, e and Figure R27, c) It also revealed the advantaged of Matrigel as the cell encapsulation scaffold, though it remained with the challenge of clinical translation.

Figure R26. Characterization of MSC viability, migration and proliferation in the HAMA HDG, LDB and HDB groups. a, b, Representative live/dead fluorescence images (a) and viability rate quantification (b) of MSCs in the HAMA HDG, LDB and HDB groups at day 0 (D0), D3 and D7. c-d, Representative EdU staining images (c) and quantification (d) of MSCs in the HAMA HDG, LDB and HDB groups at D0, D3 and D7. e, Metabolic activity of MSCs in the HAMA HDG, LDB and HDB groups at D1, D3, D5 and D7, determined using the Alamar Blue assay. n = 3. The data are represented as mean ± SD. Significant difference is determined by

one-way ANOVA, followed by Tukey's test. * $p < 0.05$; ** $p < 0.01$; **** $p < 0.0001$. Scale bar, 100 μm in all panels.

3. The authors claim that their bead-jet technique is specialized for high-throughput printing, but fail to define what high throughput means. Their movies show a rather slow (bead-by-bead) deposition of the materials into the animal wound site. Furthermore, the methods describe a gelation step that requires 15-30 minutes prior to being deposited. This is far away from “high-throughput”. As an example, in the field of microfluidics where similar cell-encapsulating beads are a focus of research, high throughput typically means Liters or kilograms worth of material per hour. Authors should demonstrate that they can print materials and fill large clinically relevant voids (102-103 cm^3) within surgically relevant time scales (i.e. 101-102 minutes).

Thanks for raising this comment. High-throughput is always among the key concerns in translational technology development. Here, “high-throughput” has two meanings. First, compared with the reported “microbead printing” techniques, such as the “aspiration-based printing” reported in Science Advances 2021¹, where the throughput was only up to ten spheroids per manipulation, here our system improved the printing frequency to approaching 1Hz, i.e. one bead per second. This was only performed in a prototype machine. The “bead printing” frequency shall not be directly compared with “bead formulation” frequency as in microfluidics. Even one bead per second allows the printing rate reaching 5 million cells per hour (1500 cells per bead, 3600 beads per hour). If further accelerating the printing rate by a few folds, it would reach the

clinical needs.

Second, the gelation time of “15-30 min” was correct but it could be accomplished when the droplets (or beads) were circulating in tubing. Then, once the printing end started to work, the printing became continuously ~1-bead-per-second. Our previous studies proved that circulating beads in tubing would not be detrimental to encapsulated cells, including iPSCs and primary cells, or alter their transcription. Please refer to “*Jiang, Shengwei, et al. "An automated organoid platform with inter-organoid homogeneity and inter-patient heterogeneity." Cell Reports Medicine 1.9 (2020): 100161.*”

The bead-jet printing platform can distribute beads into large-scale and defined patterns. For planar printing, the printed pattern could cover nearly 100 cm² in this current in-house machine. The printing scale could be further increased by improved the locomotion ability, i.e., the *x-y* movement range, of the printing nozzle (**Figure R28**). For three-dimensional (3D) patterning, the bead-jet printing could fabricate up to 2 cm × 2 cm × 3 cm structure using the current in-house printer (**Figure R29**).

Figure R28. Printing of microbeads into (a) a 12-cm long straight line, (b) a rectangular shape (10 cm \times 1 cm), (c) a square shape (3 cm \times 3 cm), and (d-f) large-scale shapes (d, 5 cm \times 7 cm; e, 8 cm \times 12 cm; f, 9 cm \times 9 cm).

Figure R29. Printing of 3D heterogeneous structures. a, Schematic of heterogeneous printing. **b,** 3D printed heterogeneous structures of gelatin beads on sacrificial gelatin support. The printed assembly had up to 50 layers. Scale bar, 1 cm.

4. Patterning high-cell density beads among acellular ones is not rationalized or based on any optimized layout. The manuscript claims that one of the advantages of their bead-jet printing technique is the patterned deposition of cellular and acellular materials. However, this aspect is not experimentally explored or discussed at all. For example, how is the deposition pattern optimized? Was a design criteria established or various patterns tested for their biological advantage? Do endogenous cells invade and cluster around cell-laden beads? Is there a heterogeneous HLA signal intensity based on where cell laden beads are present compared to acellular regions?

Thanks for raising this question. Acellular materials are used to fill in the gaps between implanted scaffold and native tissue. As the scheme below shows, the total cell number delivered to defect is calculated based on LDB group since there is no need to include other acellular materials. The number of acellular beads and volume of acellular bulk gel are calculated according to the cell density and cell numbers loaded in microsphere.

Basically, the number of cells delivered at each wound site was determined according to the defect size. **(Figure R30)**

Figure R30. HDG-MSCs, LDB-MSCs, HDB-MSCs as printed or casted to fill the defect

volume.

The number of cells in the microsphere has been calculated according to the volume and listed as following.

Table R6. Number of cells in microbeads and in bulk gel

Hydrogel formation	HDG	LDB	HDB
Cell density	1×10^7 cells/mL	2×10^6 cells/mL	1×10^7 cells/mL
Cells / bead volume	1×10^4 cells/ μ L	300 cells/bead	1500 cells/bead

For the *in vitro* study, to keep the cell seeding number consistent at the beginning when cultured in the 12 well plate, 50 HDBs, 250 LDBs, and 7.5 HDG were printed or casted into the culture medium. The total cell number across the three groups were 7.5×10^4 cells at day 0.

A design criterion of delivering cellular beads and acellular beads has been established. For the mouse VML defect models, 60% volume (~ 75 mg, 50 mm^3) of gluteus maximus (GM) muscle tissue was removed. Afterward, 300 LDBs were printed to fill in this defect volume. To keep the number of cells consistent across these groups, 60 HDBs and 240 acellular beads, 9μ L HDGs and 36μ L acellular gel were printed or casted. Acellular beads and acellular gel were used to fill in the gap between the cell-laden scaffold and the native tissue. The total number of delivered cells and the defect sizes were kept consistent across all groups.

For the skin wound healing models, different size of skin wounds, including $5 \text{ mm} \times 5 \text{ mm}$, $10 \text{ mm} \times 10 \text{ mm}$ and $15 \text{ mm} \times 15 \text{ mm}$, were created. The number of cells and microbeads, and the volume of bulk gel are provided in the table below.

Table R7. Number of cells delivered at the wound sites

Tissue types	Volume and size	Total cell number	HDG group	LDB group	HDB group
VML GM tissues	60% volume (~ 75 mg, 45μ l)	9×10^4 cells	9μ L HDGs, 36μ l acellular gel	300 LDBs	60 HDBs, 240 acellular beads
Skin wounds	$5 \text{ mm} \times 5 \text{ mm}$	2.1×10^4 cells	2.1μ L HDGs, 8.4μ L acellular gel	70 LDBs	14 HDBs, 56 acellular beads
	$10 \text{ mm} \times 10 \text{ mm}$	8.4×10^4 cells	8.4μ L HDGs, 33.6μ l acellular gel	280 LDBs	56 HDBs, 224 acellular beads
	$15 \text{ mm} \times 15 \text{ mm}$	1.8×10^5 cells	18μ L HDGs, 72μ l acellular gel	600 LDBs	120 HDBs, 480 acellular beads

We also characterized MSCs *in vivo* by using Dil cell membrane tracker after 24 h and 1 week post implantation. From the fluorescent images, we could find that endogenous cells (DAPI signal, excluding MSCs positive signal) invade and cluster around cell-laden beads. The main observation was that the implanted MSCs expanded and proliferated.

Figure R31. Representative images of MSCs in the Matrigel HDG, LDB and HDB groups after transplantation for 24 h (g) and 1 week (h) in muscle tissues. MSCs were labeled using the red Dil cell membrane tracker (Meilunbio). Acellular beads and acellular bulk gel were loaded with monodispersed green fluorescent polystyrene microspheres (Aladdin, 1 μm , 1:20 diluted in Matrigel solution). Quantification of MSCs (i) and total cells density (j) per section after transplantation for 24 h and 1 week. Scale bar, 100 μm .

Figure R32 shows the heterogenous signal distribution of cells in a few HDB, each bead was surrounded by multiple acellular beads.

Figure R32. Matrigel MSCs-HDB complex and acellular beads were distributed in the muscle defect at D7 after implantation. Matrigel MSCs-HDB was surrounded by acellular beads. MSCs were labeled using the red Dil cell membrane tracker (Meilunbio). Acellular beads and acellular bulk gel were loaded with monodispersed green fluorescent polystyrene microspheres (Aladdin, 1 μm). Scale bar, 100 μm .

5. Evidence to support patterned deposition of beads in wound models is missing. The authors should provide an early time point (e.g. 24 hours) histological image of the wound site where beads were deposited in the HDB group. They should demonstrate that cell-laden beads are present among acellular beads.

Thanks for this suggestion. We labeled MSCs using red Dil cell membrane tracker (Meilunbio). Acellular beads and acellular bulk gel were loaded with monodispersed green fluorescent polystyrene microspheres (Aladdin, 1 μm , 1:20 diluted in Matrigel solution). The sectioned images showed that the cell-laden beads were present among acellular beads.

Figure R33. Matrigel MSCs-HDB complex and acellular beads were distributed in the muscle defect at D7 after implantation. Matrigel MSCs-HDB was surrounded by acellular beads. MSCs were labeled using the red Dil cell membrane tracker (Meilunbio). Acellular beads and acellular bulk gel were loaded with monodispersed green fluorescent polystyrene microspheres (Aladdin, 1 μm). Scale bar, 100 μm .

6. Regarding patterning and printing precision, authors show a simplistic macro-deposition. More complex patterns using beads have been reported recently by others including Xin et al. Adv. Funct. Mater. 2020 and Li et al. Adv. Mater. 2018 1803475. In comparison, this manuscript does not demonstrate any heterogeneous 3D patterns despite their entire therapeutic strategy based on these patterns.

Thanks for this suggestion. Now more heterogeneous and complex patterns were printed and presented. **(Figure R34)**

Figure R34. (left) Homogenous structures of printed gelatin beads, including a THU alphabetical sparse pattern, and rectangular, triangular and hexagonal dense patterns. **(right)** Heterogeneous structures of printed gelatin beads of two different colours. Scale bar, 5 mm.

7. It is claimed that neighboring beads adhere/bond to each other but the mechanism of how this occurs is not described. Clearly there is no chemical functionalization performed on Matrigel. If it is solidified and crosslinked, then it can no longer fuse with adjacent material.

Among Matrigel beads, they formed fused interface because the beads were not 100% solidified upon printing. The post-printing incubation would further solidify the interface that glued Matrigel beads. The fused interface was not as robust as bulk Matrigel, but it maintained the structural integrity at static (**Figure R35**). The interface-fusion mechanism also applied in the *in vivo* printing, when the substrate became the native tissues.

The following description was supplemented to the main text:

Sacrificial support has positive roles in bead-jet printing. It anchors the printed building blocks toward larger architecture on demanded positions. The printing bead bonds with the support or the neighbouring beads by forming a robust interface-gel layer.

Figure R35. Representative images of acellular Matrigel beads *in vitro* post-fabrication. Scale bar, 200 μm .

Fusion or bond formation among cell-laden beads is common. We tested cellular beads fabricated from Matrigel suspended with $1.5 \times 10^7 \text{ mL}^{-1}$ NIH3T3/GFP and $1.5 \times 10^7 \text{ mL}^{-1}$ OVCAR-5/RFP. The confocal images clearly indicated that cells formed connection across the bead interfaces.

Figure R36. Zoom-in images of spheroids merging at day 2 after assembly. The spheroids were fabricated from Matrigel suspended with $1.5 \times 10^7 \text{ mL}^{-1}$ NIH3T3/GFP and $1.5 \times 10^7 \text{ mL}^{-1}$ OVCAR-5/RFP. Scale bar, 250 μm .

8. Authors claim that printing a 15-layer 3D assembly of beads retains its structural integrity. It is not clear how this is concluded from data that shows a drooping of the material and change in angle from 90 degrees to 50 degrees.

It is true that with the accumulation of beads along the z axis, the advancing angle decreased from 90 degree to 50 degrees, because the “bead-jet printing” on existing layers had “collision” effect, which may result in “sliding” of printed beads before they were glued to the structure. However, the ability to pattern beads upward was already among the best performance in both speed and structural maintenance. It was at least 10-fold faster than “aspiration-based printing”, and overcome the limitation of “ink-jet printing”, when the liquid building blocks were splashed once printed.

Figure R37. i, 3D Printing of multi-layer gelatin beads on a sacrificial gelatin support. The printed assembly had 15 layers. Scale bar, 1 cm. **j,** Advancing angle measurement of 3D printed bead assemblies with increasing layers. $n = 3$ for each measurement. Data are represented as mean \pm SD.

We performed additional experiment to prove that the printed structures were stable and robust against mechanical deformation caused by clamping, shaking in PBS buffer, and phase-transferred of both planar patterns (single layer and multiple layers) and 3D cubic structures. After mechanical interruption, all structures maintained stable, which verified that the integrity of printed structures was stable, owing to the bead-to-bead bonding formation.

Figure R38. Evaluation of structural integrity of printed architectures. a, Clamping of planar structure (single layer). **b,** Planar structure (single layer) immersed in PBS buffer. **c,** Planar structure (multiple layers) immersed in PBS buffer. **d,** 3D cube structure transferred from a 10-cm culture dish to a beaker. **e,** A 3D cube structure immersed in PBS buffer in a beaker. Scale bar, 1cm.

9. Rheology performed to demonstrate extrudability is in a low strain range (0.1-1%) whereas standard in the field for these types of experiments is 0.1-500%. This experiment should be repeated and yield strain (cross-over G' with G'') should be reported.

Thanks for this suggestion. We have redone the rheological test. Both granular and bulk gel properties of gelatin and Matrigel were evaluated. The method is described as follows.

To evaluate the rheological property difference of bulk gel and microbeads, 5% (w/v) TG-crosslinked gelatin and Matrigel was used throughout this study and tested by a rotational rheometer (MCR302, Anton Paar, Austria). Briefly, the solidified microbeads were collected in the cubes. Then the microbeads were pelleted in the tube and centrifuged at 1000 rpm for 5 min. After

that, the oil was aspirated and removed from the tube. The gel-oil complex was washed, centrifuged and aspirated for multiple steps until all oil droplets were removed. In the rheological test, a 25-mm measuring plate PP025 was used in this study. An approximate volume of microbead assembly was placed onto the center of the rheometer plate using a spatula. Then the geometry was lowered to the desired gap height (1 mm used in this study). Redundant microbeads were removed using a scraper. Oscillatory strain sweeps (0.1 to 500%, 1 Hz) were carried out at room temperature. Oscillatory frequency sweeps (0.1-100 Hz, 1% strain) were performed at room temperature. Time sweeps (0.5% strain, 1 Hz, 2 min) were performed at room temperature to determine the storage (G' , Pa) and loss (G'' , Pa) moduli of microbead assemblies and bulk gel. Shear ramp (0.01-100 s⁻¹) were used to study the relationship between viscosity and shear rate. All experiments were repeated at least three times.

Rheological tests were performed to evaluate the granular and bulk gel properties of gelatin and Matrigel. Frequency sweep showed that storage modulus of gelatin and Matrigel microbead assembly and their bulk counterparts at low frequency exhibits linear trends (**Figure R39a**). Shear-thinning behaviors were also observed across four groups by reduction in viscosity with increasing shear rates (**Figure R39b**). Strain sweep test showed that all samples yielded ($G'' > G'$) at high strains (**Figure R39c**). The yield strain of bulk gelatin was significantly higher than the granular gelatin (**Figure R39d, i**). However, no significant difference of yield strains of Matrigel between its bulk and granular configurations was observed (**Figure R39d, ii**). There was no significant difference of storage moduli of gelatin and Matrigel in their granular and bulk configurations (**Figure R39e**)

Figure R39. Rheological characterization of gelatin and Matrigel granular and bulk gels. a, Frequency sweep performed from 0.1 to 100 Hz of gelatin (i) and Matrigel (ii) granular and bulk gels. **b,** Shear-thinning properties of gelatin (i) and Matrigel (ii) granular and bulk gels. **c,** Strain sweep performed from 0.1 to 500% of gelatin (i) and Matrigel (ii) granular and bulk gels. **d,** Yield

strains of gelatin (i) and Matrigel (ii) granular and bulky hydrogel. e, Storage modulus G' of gelatin (i) and Matrigel (ii) granular and bulk gels. (** $P < 0.01$)

10. A limitation of this technique is that it can print either cell-laden or acellular materials in a batch-wise manner. This means that if a few cell-laden beads are deposited into the wound site, then they need to stay in place until the new acellular material is setup and deposited into empty areas. It is hard to imagine how this is an optimal approach with such a time-lag. More explanation and discussion should be included.

The printing of acellular materials is to keep total cell number identical across these groups. As the scheme below shows, the total cell number delivered to defect is calculated based on LDB group since there is no need to include other acellular materials. The number of acellular beads and volume of acellular bulk gel are calculated according to the cell density and cell numbers loaded in microsphere.

Figure R40. HDG-MSCs, LDB-MSCs, HDB-MSCs as printed or casted to fill the defect volume.

In real case, acellular materials were deposited into empty areas first and waiting for the deposition of cellular beads. Then cellular beads were printed into the wound site by changing another tubing loaded with the cellular beads. The process of microbeads printing or bulk gel deposition was quite short, less than 5 min for both the skins wound (70 beads for 5-mm-diameter excisional skin wounds, 1 bead per second) and VML models (300 beads for filling the defect site, 1 bead per second). For example, in this study, the skin wounds were 5-mm-diameter, and only 14 HDB beads were needed for repair of each wound. Therefore, the deposition of acellular materials and cellular materials had minimal influence on cell viability or time-lag.

For future clinical applications, when the trauma area/volume becomes many folds larger, the printing may take longer time. By then, the in-house printer could be upgraded to implemented printing at increased throughput, and integrated multi-nozzle printing.

11. Authors cite recent work related to granular hydrogels and motivate their use of beads by citing advantages of granular systems including microporosity and extrudability. However, the results are not discussed in the context of these benefits. Even though the beads are expected to degrade quite quickly (being Matrigel), there is clearly an advantage over the bulk gel (HDG) group. This should be explored by early-stage (1-3 days) immunohistochemical analysis of beads

in the wound site. At a minimum, *in vitro* models could be used to highlight differences due to the bead morphology. Authors should characterize the degree of bead packing and interstitial porosity.

Thanks for this constructive comment.

We first examined the degradation course of bulk Matrigel and Matrigel beads *in vitro* for over 7 days. There was no difference found. As the picture below shows, the edge of Matrigel beads remained clear and sharp in both the bead group and the bulk group, indicating that the bead configuration did not accelerate Matrigel degradation within 1 week. However, comprehensive understanding of Matrigel degradation remains challenging, especially in cell-presence manners (to mimic *in-vivo* environment) and over long term.

Figure R14. Degradation test of bulk Matrigel and acellular Matrigel beads *in vitro* at D1, D3 and D7. Scale bar, 200 μm .

We then studied the distribution of printed beads and MSCs in VML wounds. We labeled MSCs using Dil cell membrane tracker (Meilunbio, 1:1000). The acellular beads were labeled green by encapsulating monodispersed green fluorescent polystyrene microspheres (Aladdin, 1 μm , 1:20 diluted in Matrigel solution). MSCs positive signal is calculated per section. At 1-week post-implantation, significantly higher proportion of MSC-positive signal appeared in the Matrigel-HDB group, compared with the Matrigel-HDG and Matrigel-LDB groups, indicating that the HDB configuration accelerated cell proliferation *in vivo*. (**Figure R41**)

Figure R41. Representative images of MSCs in the Matrigel HDG, LDB and HDB groups after

transplantation for 24 h (g) and 1 week (h) in muscle tissues. MSCs were labeled using the red Dil cell membrane tracker (Meilunbio). Acellular beads and acellular bulk gel were loaded with monodispersed green fluorescent polystyrene microspheres (Aladdin, 1 μm , 1:20 diluted in Matrigel solution). Quantification of MSCs (i) and total cells density (j) per section after transplantation for 24 h and 1 week. Scale bar, 100 μm .

Next, the interstitial porosity of microbead assemblies of gelatin, Matrigel, HAMA and GelMa was investigated by adopting different bead-sizes (D = 650 μm and 450 μm) and degree of packing (low packing and high packing by different centrifugation rates). The microbead assemblies were infiltrated with FITC-dextran (Sigma-Aldrich, 2M Da, 10 mg/ml in PBS) solution.

Assembly of small gelatin beads (D = 450 μm) at low packing density had a void fraction of $\sim 14.1\%$, significantly higher than the void fraction at high packing density ($\sim 1.3\%$). Assembly of large gelatin beads (D = 650 μm) at low packing density had a void fraction of $\sim 11.5\%$, significantly higher than the void fraction at high packing density ($\sim 1.5\%$) (Figure R42, f-h).

Likewise, porosity of granular Matrigel was investigated (Figures 2i and 2k). Assembly of small Matrigel beads (D = 450 μm) at low and high packing density reached the void fractions of $\sim 3.9\%$ and $\sim 2.4\%$, respectively. Assembly of large Matrigel beads (D = 650 μm) at low and high packing density reached void fractions of $\sim 8.3\%$ and $\sim 1.7\%$, respectively (Figure R42, i-k). Both the void fractions at low-packing and high-packing, as well as the difference resulted from small bead packing and large bead packing, were decreased in granular Matrigel and granular gelatin. The results proved that the porosity of bead assembly was controllable by adjusting the degree of packing.

Figure R42. f-h, Representative confocal 2D (f) and 3D (g) representative images of granular

gelatin microgels assembled from different microbeads ($D = 450 \mu\text{m}$ and $650 \mu\text{m}$) and packing densities. The bead samples were infiltrated with FITC-dextran (2M Da , 10 mg/ml) solution. **(h)** Quantification of granular gelatin porosity. **i-k**, Representative confocal 2D **(i)** and 3D **(j)** representative images of granular Matrigel microgels assembled from different microbeads ($D = 450 \mu\text{m}$ and $650 \mu\text{m}$) and packing densities. The bead samples were infiltrated with FITC-dextran solution. **(k)** Quantification of granular Matrigel porosity. Here, the high-packing and low-packing conditions were reached by centrifugation of the microbead assembly for 5 min at 1000 rpm or at 15, 000 rpm. Scale bar, $200 \mu\text{m}$. The data are represented as mean \pm SD. Significant difference is determined by unpaired two-tailed student t-test. $**P < 0.01$, $***p < 0.001$; $****p < 0.0001$.

Similar results were also obtained by using GelMA and HAMA microgels (**Figure R43**).

Porosity of GelMA and HAMA is also been characterized. Small GelMA granular hydrogels packed by low packing resulted in void space of $\sim 16.0\%$, higher than small GelMA granular hydrogels packed by high packing (void space of $\sim 15.4\%$). Large gelatin GelMA hydrogels packed by low packing resulted in void space of $\sim 13.6\%$, significantly higher than large GelMA granular hydrogels packed by high packing (void space of $\sim 10.5\%$). Small HAMA granular hydrogels packed by low packing resulted in void space of $\sim 11.7\%$, higher than small HAMA granular hydrogels packed by high packing (void space of $\sim 7.2\%$). Large HAMA granular hydrogels packed by low packing resulted in void space of $\sim 13.6\%$, significantly higher than large HAMA granular hydrogels packed by high packing (void space of $\sim 5.6\%$). The result indicates also high degree of packing may decrease the porosity of microgels, which is consistent with the finding above.

Figure R43. Porosity characterization of granular GelMA and HAMA hydrogels. a-c, Representative confocal 2D **(a)** and 3D **(b)** representative images of granular GelMA hydrogels assembled from different microbeads ($D = 450 \mu\text{m}$ and $650 \mu\text{m}$) and packing densities. The bead

assemblies were infiltrated with FITC-dextran solution. (c) Quantification of granular GelMA porosity. **d-f**, Representative confocal 2D (**d**) and 3D (**e**) representative images of granular HAMA hydrogels assembled from different microbeads ($D = 450 \mu\text{m}$ and $650 \mu\text{m}$) and packing densities. The bead assemblies were infiltrated with FITC-dextran solution. (**f**) Quantification of granular HAMA porosity. $n = 3$. Here, the high-packing and low-packing conditions were reached by centrifugation of the microbead assembly for 5 min at 1000 rpm or at 15,000 rpm. Scale bar, $200 \mu\text{m}$. The data is represented as mean \pm SD. * $p < 0.05$; ** $p < 0.01$; Significant difference is calculated by unpaired two-tailed student t-test.

12. At least two relevant negative control groups are missing in the in vivo experiments. 1) Acellular beads only – this is expected to elicit some degree of biological/regenerative response since Matrigel is a naturally derived matrix. 2) Cell injection without a material – since cell delivery as a bolus is the gold standard in the clinic, this should be included to demonstrate superiority over the biomaterial groups.

We now have added the additional control groups, acellular Matrigel beads printing and MSCs suspension injection, as suggested on both VML defect and skin wound models.

Figure R44. Muscle regeneration treated with acellular Matrigel beads and MSCs suspension injection. The acellular beads were densely printed as in the practice of HDB plus acellular beads, and MSCs suspension were injected in the whole VML region. **a**, The photograph of acellular beads and cell injection post-printing and regeneration after 4 weeks. **b**, The MTS staining of muscle tissues harvested at week 4 treated using acellular beads and MSCs injection. **c**, Representative images of immunostaining for desmin, CD31, α -SMA, neurofilament, β IIIIT, MYH7, PAX7, CD68 and CD3. The injected MSCs were identical to the MSCs-HDB implantation in cell counts, with 9×10^4 cells per wound.

Figure R45. Skin wound regeneration treated with acellular Matrigel beads and MSCs suspension injection. The acellular beads were densely printed as in the practice of HDB plus acellular beads and MSCs suspension were injected in the skin wound region. **a-b**, The photographs of wound closure area (**a**) and quantification (**b**) at week 0, week 1, week 2, week 3. ($n = 3$) **c**, The HE and MTS staining of skin tissues harvested at week 3 treated using acellular beads and MSCs injection. **d**, Representative images of immunostaining for CD31, Ki67, CD68, CD3. The injected MSCs were identical to the MSCs-HDB implantation in cell counts, 2.1×10^4 cells per wound.

13. Another technical concern is the deposition of oil at the site of injury. Oil and surfactant removal is one of the most important steps in other techniques e.g. batch emulsions and microfluidics that are used to fabricate beads and microgels for tissue implantation. The authors claim that the oil used (Novec 7500) is a volatile liquid that evaporates when exposed to air should not be an excuse to ignore the potential concerns over oil in the tissue site. Especially for scaling

up bead deposition, it is easy to imagine how rate of oil evaporation would be masked by rate of bead +oil deposition leaving behind more oil in the wound site.

In this study, we did not use any surfactant. The droplets and beads were maintained by the channel geometry and separated by the oil plugs, until being printed at the outlet, when the beads were stable as solids.

We used HFE 7000, instead of 7500. The boiling point of this oil is quite low (b.p. 34 °C). Its evaporation at ambient is fast. This oil is also bio-inert, which have minimal influence on cell growth¹⁵.

We performed the *in vitro* study to prove the fast evaporation and bio-compatibility of HFE7000.

Figure R46. Effects of oil residues on MSCs growth in the Matrigel LDB and HDB groups. The oil was dispersed in the tubing during droplet formation and then ejected into the cell culture media during printing by compressed air. **a**, The parameters of HFE 7000 and HFE 7100 oil. **b**, MSCs cultured in the Matrigel LDB and HDB groups with 7000 and 7100 ejection for 7 days. HFE7000 was used throughout the study.

To accelerate the evaporation of oil, the mice is placed under a heater for 2 min. We also tested oil evaporation by directly dripping oil onto mice and human skin. From the image below, we can see the oil is easy to operate within seconds post dripping.

Figure R47. Oil evaporation test. a, A mouse was placed under a heater for 2 min. **b,** Drip oil onto the dorsal skin of a shaved mouse and human skin.

14. It is concerning that the authors compare two very different materials i.e. Matrigel and gelatin, which vary not only in physical features but chemical compositions, and claim that Matrigel is better than gelatin as an extracellular scaffold based on viability and proliferation experiments. These types of comparisons and statements should be removed and the authors should revisit and reconsider their approach to fairly comparing biomaterials.

The author thanks for this suggestion. However, the method proposed in this study is to set up a new microsphere printing technique. It's common sense that gelatin or its derivatives have been used widely in bioprinting owing to its good structural stability and tunable mechanical properties. Therefore, we compared bead-jet printing technique using Matrigel and gelatin. Traditional printing techniques cannot print Matrigel unless inert structural materials were used, such as PCL, PLGA, alginate, etc. Traditional printing techniques can print gelatin, e.g., microextrusion, but not continuous printing of gelatin beads. We demonstrated the performance of our system by using both materials to prove that it was not limited to one material. Further, the principle of "sparse patterning of high-density MSCs in gel" by using gelatin as the materials has both scientific and engineering perspectives. So, we still believe this part is important to highlight the advantages of our printing technique.

We also tested our bead jet printing platform using HAMA material.

5% HAMA was crosslinked using 0.25% LAP solution is used as the UV crosslinker. Live/dead staining shows that cell won't migrate out from the microsphere and spread out. Cell viability of HAMA microsphere and bulk gel decreases after 1 week culture, significantly lower in HAMA

LDB group. Significant lower proportion of EdU positive cells appeared in HAMA LDB group compared with other groups. Live/dead staining and EdU test all elucidate low cell density may be detrimental for cell growth, reducing the cell activity and cell proliferate rate. This is also consistent with our previous finding. Metabolic activity of MSCs is also performed. Cell metabolic activity in HAMA LDB group is also significantly lower at day 5 and day 7 compared with HDG and HDB group. Compared with cell metabolic activity of MSCs in HAMA and Matrigel, metabolic activity is much higher in Matrigel than HAMA.

Figure R48. Characterization of MSC viability, migration and proliferation in the HAMA HDG, LDB and HDB groups. a, b, Representative live/dead fluorescence images (a) and viability rate quantification (b) of MSCs in the HAMA HDG, LDB and HDB groups at day 0 (D0), D3 and D7. c-d, Representative EdU staining images (c) and quantification (d) of MSCs in the HAMA HDG, LDB and HDB groups at D0, D3 and D7. e, Metabolic activity of MSCs in the HAMA HDG, LDB and HDB groups at D1, D3, D5 and D7, determined using the Alamar Blue assay. n = 3. The data are represented as mean ± SD. Significant difference is determined by one-way ANOVA, followed by Tukey's test. *p < 0.05; **p < 0.01; ****p < 0.0001. Scale bar, 100 µm in all panels.

Biological limitations/concerns:

14. Dosage determination is not optimized. It is unclear what is the basis for using the cell density employed. For a manuscript that is motivated by clinical challenges of high-dose therapies, this is an important aspect. Authors should investigate a dose-response relationship within their HDB and LDB groups. The manuscript also contains seemingly conflicting statements such as “sparse patterning of dense MSCsenabling therapy of large scale traumatic injury with a limited number of MSCs.” where the solution to the problem of high cell doses is the encapsulation of high cell doses in fewer materials.

Thanks for this comment. We have explored cell growth with an increasing dose of MSCs, including 150, 300, 1500, 3000, 6000 cells per microbead using LDH-Cy Quant assay. The LDH-Cy Quant assay showed that high cell density of MSCs could gain significant higher proliferation rate and accelerate MSCs growth compared with low cell density of MSCs and bulk gel of MSCs. After 4, 7 and 14 days in culture, cell growth of 1500-cells-per-microbead was significantly higher than the other groups. High cell density may accelerate the proliferation of cells compared with low cell density. (**Figures R49a and R49b**)

Alamar Blue assay was employed to test cell metabolic activity. After being cultured for 7 days, the cell metabolic activity in high-cell-density groups, including 1500, 3000, 6000-cells-per-bead, were significantly higher than in the low cell density group, including 150 and 300-cells-per-bead (**Figure R49c**).

Figure R49. **a**, The schematic illustration of MSCs growth in the Matrigel HDG, LDB and HDB groups. MSCs in HDG and microbeads of different cell densities, including 150, 300, 1500, 3000, 6000 cells per microbead, were characterized at D0, D3, D7 and D14. Live and dead cells were stained with calcein-AM (green) and propidium iodide (red), respectively. The initial cell numbers were kept consistent among all groups. **b**, OD values of MSCs growth at 490 nm in Matrigel HDG, and in Matrigel beads comprising different cell densities at day 1 (D1), D3, D7 and D14, determined using the LDH-Cy Quant assay. **c**, Metabolic activity of MSCs at D1, D3, D7 and D14, determined using the Alamar Blue assay. * $p < 0.05$; ** $p < 0.01$; *** $p < 0.001$; **** $p < 0.0001$.

We also quantified cell viability across Matrigel HDG, and different doses of Matrigel microspheres using live/dead staining. The result indicated that extremely high cell density, e.g., 3000 – 6000 cells per bead, may decrease cell viability. This may have been resulted from the limited access to the cell culture medium and higher metabolic stress in the microspheres. (**Figure R50**)

Figure R50. Quantification of MSCs viability rates in Matrigel HDG, and different doses of Matrigel microspheres at D1, D4, D7 and D14. * $p < 0.05$; ** $p < 0.01$; *** $p < 0.001$; **** $p < 0.0001$.

We are sorry for this unclear statement that “*sparse patterning of dense MSCsenabling therapy of large scale traumatic injury with a limited number of MSCs*”. In real case, some high cell dose microspheres are printed among the acellular beads.

The manuscript has been corrected as following:

Secondly, sparse patterning of high density MSCs-laden beads among acellular beads also enlarged the transplantation coverage, thus enabling therapy of large-scale traumatic injury with a limited number of MSCs.

15. For the skeletal muscle work, the authors show data that suggests MSCs (HLA) differentiate and integrate into muscle fibers. Although MSCs may integrate structurally into the tissue, satellite cells and other muscle-resident cells are most likely key players in the regeneration process. However, this is neither discussed nor explored. Further, is differentiation the common mechanism through which regeneration is achieved in the muscle and skin models?

After implantation for 1 week, the total cell density in the regenerated muscle tissues was analyzed, including the transplanted MSCs and other endogenous cells (**Figure R51 j**). Total cell density was also significantly higher in the HDB group than in other groups. This may suggest both MSCs and other endogenous cells, are important for muscle regeneration.

Figure R51. Representative images of MSCs in the Matrigel HDG, LDB and HDB groups after transplantation for 24 h (g) and 1 week (h) in muscle tissues. MSCs were labeled using the red Dil cell membrane tracker (Meilunbio). Acellular beads and acellular bulk gel were loaded with monodispersed green fluorescent polystyrene microspheres (Aladdin, 1 μm , 1:20 diluted in Matrigel solution). Quantification of MSCs (i) and total cells density (j) per section after transplantation for 24 h and 1 week. Scale bar, 100 μm .

We also analyzed the populations of muscle-resident cells in the regenerated muscle tissues at week 4 immunostained using PAX7, a marker for muscle resident cells. The PAX7 signal was significantly upregulated in Matrigel MSCs-HDB group, indicating that muscle resident cells also played important role in muscle regeneration.

Figure R51. Immunofluorescence for PAX7 to determine the population of muscle resident cells and their quantification by fluorescence area. *** $p < 0.001$; **** $p < 0.0001$.

We also implanted MSCs labeled with Dil cell membrane tacker to the skin wounds (Figure R52). Interestingly, only at 1 week post-implantation, there appeared hair-follicle like structures in the Matrigel MSCs-HDB group, but not in other groups. It provided further evidence that MSCs-HDB may regenerate tissue through differentiation.

Figure R52. MSCs in the Matrigel HDG, LDB and HDB groups after implantation for 1 week in skin tissues. MSCs were labeled using the Dil cell membrane tracker (Meilunbio). Scale bar, 100 μm .

However, the mechanism of tissue regeneration by MSCs implantation remains unclear in the field of regenerative medicine. The differentiation of MSCs is the one of the major routes directing tissue regeneration^{16, 17, 18, 19}. Beyond any doubt, further studies are demanded to elaborate the mechanism via global and cross-discipline collaboration.

17. The muscle injury model is defined as a $\sim 60\%$ removal of tissue by volume, but how this was achieved (e.g., with a biopsy punch or other) is not described. Considering the small size of mice muscles, this information about the technique will be relevant for the field.

The VML model was performed with reference to reported article²⁰. The VML model was created by using a surgical scissor. Nearly 75 mg, 50 mm^3 of gluteus maximus (GM) muscle tissue was taken out from the native tissues. It accounted for $\sim 60\%$ removal of tissue by volume.

Figure R53. Surgical procedure of GM VML modeling.

18. Authors should clarify if the transplanted MSCs adopt roles in skeletal muscle that are similar to those performed by fibro/adipogenic progenitors, which are also a stromal cell population. Single cell sequencing or other similar techniques could be performed to determine how the muscle-resident cell population changes as a result of treatment.

Thanks for this suggestion.

It is known that in response to injury, skeletal muscle displays a dynamic multicellular response that involves several cell types and discrete regenerative steps, among which the fibro/adipogenic progenitors (FAPs) are actively coordinated²¹. Conventional strategies to study FAPs rely on enrichment by fluorescence-activated cell sorting using a transgenic reporter or prospective isolation markers²². These methods, however, are ill-suited to capture the subtle, continuous cell state transitions which are critical for myogenesis due to a paucity of highly stage-specific cell isolation markers and the rarity of these cells. New technologies such as single-cell RNA-seq revealed the cellular heterogeneity of FAPs and their complex regulatory network during muscle regeneration²³.

So, if we compare the roles of transplanted MSCs with FAPs, single-cell RNA-seq should be applied. However, we consulted the BGI company, a renowned gene sequencing company in China about the commercial single cell RNA sequencing (scRNA-seq) of regenerated muscle tissues. The technician in BGI told us it was difficult for skeletal muscles to perform single cell sequencing based on their current technology, because skeletal muscle cells were too big to satisfy the measurement requirements of single cell sequencing.

Based on the available technology and equipment, we verified the roles of transplanted MSCs adopted in muscle regeneration in terms of myofiber formation, vascularization, neural integration, muscle resident cell changes and immunomodulatory effects.

We stained PAX7 of regenerated muscle tissue, a typical marker for muscle resident cell and compared its expression across groups. The expression of PAX7 was significantly higher in the HDB group than in other groups. Though it's still difficult to distinguish the roles of transplanted MSCs and their EVs, and resident endogenous cells, it provides clues of the intermediate state of tissue regeneration by MSC therapy. To further elaborate the roles of each mechanism among the three, tremendously more work would need to be done with new techniques involved.

Figure R54. Immunofluorescence for PAX7 to determine the population of muscle resident cells and their quantification by fluorescence area. *p < 0.001; ****p < 0.0001.**

18. Muscle regeneration is a complex process that involves diverse cell types including immune cells. The authors should perform a more rigorous analysis of how the immune compartment (macrophages, T-cells, neutrophils) is altered in response to MSC therapy – especially since MSCs are known for their immunomodulatory effects.

We used CD68 to stain M1 macrophage and CD3 to stain T-cells, and observed the MSCs immunomodulatory effects. Compared with non-treat group, the expression of CD68 and CD3 was decreased in the treatment groups in both regenerated muscles and skin tissues. The result verified that MSCs could moderate or suppress the inflammatory activity. (Figures R55 and

R56.)

Figure R55. Immunofluorescence for CD68 and CD3 to determine the population of macrophages and T-cells and their quantification by fluorescence area in muscle tissues. The data are represented as mean \pm SD. Significant difference is determined by one-way ANOVA, followed by Tukey's test. * $p < 0.05$; **** $p < 0.0001$.

Figure R56. Immunofluorescence for CD68 and CD3 to determine the population of macrophages and T-cells and their quantification by fluorescence area in skin tissues. The data are represented as mean \pm SD. Significant difference is determined by one-way ANOVA, followed by Tukey's test. * $p < 0.05$; **** $p < 0.0001$.

20. Authors use desmin as a marker for regenerated fibers. This is not the standard in the field and is not exclusively used as a marker for new fibers. Authors should perform embryonic myosin heavy chain staining, or quantify centrally nucleated myofibers. Both of these are established markers of muscle fiber regeneration.

There are several articles applied desmin as a sign of skeletal muscle formation^{20, 24-27}. Therefore, we still keep the data of desmin immunostaining.

By following the comment advice, we used MYH7 to do embryonic myosin heavy chain staining. The MYH7 signal was significantly higher in the HDB group compared with other groups.

Figure R57. Immunofluorescence for MYH7 (b) to determine the myofiber maturation and their quantification by fluorescence area in muscle tissues. The data are represented as mean \pm SD. Significant difference is determined by one-way ANOVA, followed by Tukey's test. **** $p < 0.0001$.

21. Although HLA is shown as a marker for cell presence, no negative controls are provided and the signal intensity is very low in relation to the number of cells transplanted. Authors should perform *in vivo* imaging with labeled cells at least in the HDB group to show the duration of cell presence at the site of injury.

Thanks for this suggestion. We now provided the negative control of HLA signal.

Figure R58. Representative images of negative control stained using anti-rabbit secondary antibody (Alexa Fluor 594) without HLA primary in Matrigel HDB-MSCs after 4-week transplantation. Scale bar, 50 μm .

We have tried to transfect MSCs using VSVG-LENTAI-sEF1a-ffLuciferase-T2A-copGFP-WPRE-pA (LTR). 1×10^8 UT/ml lentivirus was used to transfect the MSCs. Before passage, the transfection rate of MSCs could achieve nearly 80%. However, transfection rate of MSCs decreased dramatically after passage, leading to the failure of transfection. We had to give up using these cells for *in vivo* imaging.

Figure R59. Representative images of transfected MSCs using luciferase gene before and after passage. Scale bar, 100 μm .

Then, we labeled MSCs using the Dil cell membrane tracker for implantation, and examined the cells at day 1 and day 7. MSCs positive signal was calculated per section. At 1 week after implantation, significant higher proportion of MSC positive signal appeared in the Matrigel-HDB group compared with Matrigel-HDG and Matrigel-LDB groups, suggesting that MSCs in HDB implantation gained higher proliferation rates *in vivo*. (**Figure R60**)

Figure R60. Representative images of MSCs in the Matrigel HDG, LDB and HDB groups after transplantation for 24 h (g) and 1 week (h) in muscle tissues. MSCs were labeled using the red Dil cell membrane tracker (Meilunbio). Acellular beads and acellular bulk gel were loaded with monodispersed green fluorescent polystyrene microspheres (Aladdin, 1 μm , 1:20 diluted in Matrigel solution). Quantification of MSCs (j) and total cells density (k) per section after transplantation for 24 h and 1 week. Scale bar, 100 μm .

22. Authors make many conjectures related to the mode of action of MSCs in their models. For example, they claim that “muscle regeneration may be attributed to MSCs proliferation and EVs production in the initial stage and MSCs differentiation in the later stage”. Also “we tend to believe that both MSCs themselves and EVs promoted skeletal muscle regeneration and, likewise,

skin regeneration”. There is no evidence provided to support these claims. MSCs’ ability to secrete EVs and also differentiate into multiple lineages is well-established in vitro but that does not necessarily mean that it’s how they stimulate regeneration. At a minimum, more experiments should be performed to characterize what these cells are producing in vivo (e.g., RNA seq of explanted cells). Other conjecture includes low MSC proliferation and migration due to intercellular communication and low concentration of secreted molecules. This is not backed up by any evidence.

We have analyzed genes related MSCs secretome to identify what MSCs might producing in vivo (Figures R61 and R62). In muscle tissue, the gene expression across different groups shows no big difference. However, in regenerated skin tissues, the gene expression of Vcam1, Ccl5, Il12b, Cxcl10 and Cxcl9, etc. are higher in Matrigel HDB-MSCs group, compared with the other groups. This may indicate MSCs could stimulate regeneration through MSC secretome function.

Figure R61. Heatmap of genes expression related to MSCs secretome in regenerated muscle tissue regenerated by Matrigel HDG, Matrigel LDB, Matrigel HDB, non-treat and normal groups.

Figure R62. Heatmap of genes expression related to MSCs secretome in regenerated skin tissue regenerated by Matrigel HDB, acellular beads (ACB) and normal groups.

Within the PI3K-AKT signaling pathway, many secretory factors, including HGF, VEGFA, FGF7 and PDGFB promoting cell survival and proliferation, and receptors, including EPOR, OSMR and TEK, were significantly enriched in the Matrigel HDB group. Therefore, based on the results of augmented cell proliferation, growth and migration, this may be attributed to multiple factors including the reduced intercellular communication in a sparse population and lower concentrations of cell secreted molecules. We also further explored secreted genes expression using qPCR., including HGF, TBK, VRGFA, and EPOR. The expression of HGF, TBK, VRGFA, and EPOR is higher in HDB group compared with the other two groups.

Figure R63. qPCR gene relative expression in the Matrigel HDG-MSCs, LDB-MSCs, HDB-MSCs groups, including HGF, TBK, VRGFA, and EPOR. $n = 3$. The data are represented as mean \pm SD. Significant difference is determined by one-way ANOVA, followed by Tukey's test. The significant differences of parallel groups are compared with the HDB group. * $p < 0.05$; **** $p < 0.0001$.

The secreted molecules production, e.g., extracellular vesicles (EVs) was also investigated using nanoparticle tracking analysis (NTA) among HDG-EVs, LDB-EVs and HDB-EVs. NTA results also revealed that Matrigel HDB-MSCs had the highest production of EVs, reaching $2.6 \times 10^{10}/\text{mL}$, whereas the HDG-EVs and the LDB-EVs only reached $1.1 \times 10^{10}/\text{mL}$ and $1.0 \times 10^{10}/\text{mL}$, respectively (**Figure R64 e**). The initial cell numbers across different conditions at EVs collection were identical (9.0×10^5 cells/group).

Figure R64. Matrigel HDB group augments MSCs production of extracellular vesicles (EVs). **a-c**, Transmission electron microscopy (TEM) images and nanoparticle tracking analysis (NTA) of HDG-EVs (**a**), LDB-EVs (**b**) and HDB-EVs (**c**). Scale bars, 200 nm. **d-e**, The size (mean diameter) (**d**) and particle concentration (**e**) of HDG-EVs, LDB-EVs and HDB-EVs. $n = 3$.

23. The authors should revisit their explanation for why beads in HDB group are breaking down more rapidly than those in LDB group. The manuscript states that this implies a higher activity of

cells in the HDB group. However, it is likely that this is due to a obvious significant difference in the number of cells that secrete enzymes such as MMPs. The authors could also perform cell metabolic activity experiments to test their hypothesis.

Thanks for this suggestion. Alamar Blue assay was employed to test cell metabolic activity. After being cultured for 7 days, the cell metabolic activity in high cell density groups were significantly higher than in the low cell density group. Therefore, the faster break down of HDB beads than LDB beads could be attributed to the elevated metabolic activity (**Figure R65 c**).

However, we also agree with the review comment that the amount of MMPs produced by HDB-MSCs was also higher than the LDB-MSCs, further resulting in accelerated break-down.

Figure R65. a, The schematic illustration of MSCs growth in the Matrigel HDG, LDB and HDB groups. MSCs in HDG and microbeads of different cell densities, including 150, 300, 1500, 3000, 6000 cells per microbead, were characterized at D0, D3, D7 and D14. Live and dead cells were stained with calcein-AM (green) and propidium iodide (red), respectively. The initial cell numbers were kept consistent among all groups. **b**, OD values of MSCs growth at 490 nm in Matrigel HDG, and in Matrigel beads comprising different cell densities at day 1 (D1), D3, D7 and D14, determined using the LDH-Cy Quant assay. **c**, Metabolic activity of MSCs at D1, D3, D7 and D14, determined using the Alamar Blue assay. *p < 0.05; **p < 0.01; ***p < 0.001; ****p < 0.0001.

24. The authors describe a loss in stemness of MSCs in LDB group by quantifying the expression of surface markers. Performing multi-lineage differentiation experiments would help contextualize this loss of stemness.

Thanks for this advice. To explore multi-lineage (osteogenic and adipogenic) differentiation on Matrigel HDG, LDB and HDB-MSCs, we set up different types of cell condition.

- (1) Differentiation of microbeads post-fabrication: Osteogenic and adipogenic differentiation culture medium was used from day 0. The differentiation tracking lasted for 3 weeks. (**Figure R66 a**)
- (2) Differentiation of microbeads after being cultured using normal condition for 5 days:

Microbeads were cultured using normal cell culture medium for 5 days. After cells spread out from microbeads, cell culture medium was replaced by osteogenic and adipogenic differentiation culture medium. **(Figure R66 b)**

(3) Differentiation of MSCs retrieved from microbeads: Microbeads were cultured using normal cell culture medium for 7 days. Then cells were dissociated from Matrigel using collagenase I. Cells were cultured in cell culture dish using normal cell culture medium and reached to ideal confluence after being cultured for 2 days. After that, the cell medium was replaced using osteogenic and adipogenic differentiation culture medium. **(Figure R66 c)**

Figure R66. Schematic illustration of osteogenic differentiation of MSCs-Matrigel complexes cultured at different conditions. **a**, Direct differentiation of microbeads from D0. **b**, Differentiation of microbeads from D5. The microbeads were cultured using normal cell culture medium (DMEM / F12 supplemented with 10% FBS and 1% penicillin/streptomycin) for 5 days before transferring to differentiation medium. **c**, Differentiation of MSCs retrieved from microbeads after being cultured in normal cell culture medium for 7 days. All differentiation tests were tracked for 3 weeks.

The figures below (**Figures R67 and R68**) showed the significantly higher rates of differentiation toward either osteogenic lineage or adipogenic lineage upon induction under the high density conditions (the HDB and HDG groups) than the low density condition, i.e. the LDB group.

Figure R67. Osteogenic differentiation of Matrigel hydrogels cultured at different conditions. **a**, Osteogenic differentiation of microbeads from D0. **b**, Osteogenic differentiation of microbeads from D5. The microbeads were cultured using normal cell culture medium for 5 days and then culture medium was replaced using osteogenic differentiation culture medium. **c**, Osteogenic differentiation of MSCs retrieved from microbeads. All differentiation lasts for 3 week and stained using Alizarin Red S staining. Scale bar, 100 μm.

For osteogenic differentiation, MSCs in the LDB group has lower expression than MSCs in the HDG and HDB groups.

Figure R68. Adipogenic differentiation of MSCs-Matrigel complexes cultured at different conditions. a, Adipogenic differentiation of microbeads from D0. **b,** Adipogenic differentiation of microbeads from D5. The microspheres were cultured using normal cell culture medium for 5 days and then the culture medium was replaced using adipogenic differentiation medium. **c,** Adipogenic differentiation of MSCs retrieved from microbeads after being cultured in normal cell culture medium for 7 days. All differentiation lasts for 3 weeks and stained using Oil Red O staining. Scale bars, 100 μ m.

For adipogenic differentiation, MSCs in LDB group has lower expression than MSCs in HDG and HDB group. Therefore, multi-lineage differentiation indicates low cell density may lose their stemness owing to low cell activity.

25. The authors perform RNA-sequencing and observe many differentially expressed genes between groups. However, they then pick some of these genes and attribute differences in cell behavior to this expression pattern. For example, they implicate the PI3K-AKT pathway – which is a ubiquitous signaling pathway involved in many biological processes. They do not discuss other highly differentially expressed genes e.g. those related to cancer pathways (which might be due to Matrigel being derived from the ECM of a cancerous cell line). Further, their analysis of the western blots of the AKT pathways needs revision as they do not normalize or quantify pAKT to AKT expression which, when performed, will likely remove any differences between groups.

Thanks for the constructive comments. We analyzed the highly differentiated gene expression

profiles related to cancer pathways. More genes were enriched in the HDB-MSCs group, which might be attributed to the augmented cell growth and regeneration. **(Figure R69)**

Figure R69. Heatmap of genes expression related to cancer pathways in Matrigel HDG, LDB and HDB-MSCs.

Quantification of pAKT to AKT has done in the previous submission. Both the western blot and the immunostaining results showed no significant difference. However, individual expression of pAKT and AKT was improved, suggesting that the total activity had increased **(Figure R70)**. This means the effect is specific to the change in expression and not to upstream kinases or phosphatases. We also redone the western blot of pAKT to AKT **(Figure R71)**. The pAKT to AKT still shows no significance.

Figure R70. Quantification of activated proteins among the Matrigel HDG, LDB and HDB groups. a, Quantification of Western blot protein levels of MSCs. n = 3. b, Quantification of immunofluorescence for pAKT, AKT and β -CATENIN of MSCs. n = 3. Data are represented as mean \pm SD. Significant difference is determined by one-way ANOVA followed by Tukey's test. Significant differences of all parallel groups are compared with the HDB group. *p < 0.05; **p < 0.01; ***p < 0.001; ****p < 0.0001.

Figure R71. Western blotting of PI3K-Akt signaling pathway proteins, pAKT and AKT.

26. Histological images are very small making it very challenging to see any structural details. Some of these seem to be from smaller regions of interest within larger wound sites. The authors should provide higher resolution images and indicate which part of the tissue the regions of interest are extracted from.

Now we have supplied higher resolution images (**Figure R72**). The regions of interest are highlighted.

Figure R72. The full H&E and magnification images of regenerated muscle tissues at 1 week, 2 weeks, 3 weeks and 4 weeks, in mice VML model with no treatment (self-healing), casted with MSCs-laden Matrigel HDG, and printed with MSCs-laden LDB and HDB beads, respectively. The HDB-MSCs were printed in sparse pattern, interspaced with acellular Matrigel beads. Scale bars, 500 μ m.

Writing and References:

27. Authors do not provide a rationale for using Matrigel and fail to discuss its well known drawbacks, as discussed in many recent papers including Nguyen et al. Nat. Biomed. Eng. 2017, and Aisenbrey and Murphy Nat. Rev. Mater. 2020). Authors should provide an objective and well-balanced discussion on the use of Matrigel and future implications on their technique and outcomes.

The rationale reason for using Matrigel in this study has been addressed in the response to comment.

Briefly, the composition of Matrigel is similar to ECM for its basement membrane-like nature, thus it was selected as the biomaterial in this research. Matthias Lutolf et al. noted that Matrigel has advantages over artificial biomaterials in directing stem-cell rate³⁵. The growth support of Matrigel on dermal papilla and sweat glands has also been verified in previous studies³⁶. The interaction of skin-derived stem cells with Matrigel acts in concert to give rise to a series of spatially and temporally coordinated events that regulate the stem-cells fate, specifically, directional differentiation into epidermis and dermis with necessary appendages.

Herein, in the main text, we added a paragraph to brief the rationale:

Matrigel, similar to extracellular matrix (ECM) for its basement membrane-like nature, was selected as the biomaterial in this research. Matrigel has advantages over commercial synthetic biomaterials in directing stem-cell rate²⁸. The interaction of skin-derived stem cells with Matrigel gives rise to a series of spatially and temporally coordinated events that regulate the stem-cells fate²⁹.

The shortcoming of Matrigel has also been discussed in the manuscript and listed as following:
However, though the cytocompatibility of Matrigel is favorable, the clinical promise is limited due to its tumor-derived, ill-defined, variable composition and low availability for lab-used condition⁵⁴⁻⁵⁵. The potential for antigenicity is one of the inherent limitations.

...Therefore, for future clinical translation of this strategy, synthetic biomaterials with high biocompatibility, cell-ligation ability, biodegradability and appropriate viscoelastic and apathogenic properties should be innovated and commercialized to provide appropriate alternatives to Matrigel.

The cited articles are as following:

28. Lutolf, M. P., Gilbert, P. M., & Blau, H. M. (2009). Designing materials to direct stem-cell fate. *Nature*, 462(7272), 433-441.

29. Lee, J., Rabbani, C. C., Gao, H., Steinhart, M. R., Woodruff, B. M., Pflum, Z. E., ... & Koehler, K. R. (2020). Hair-bearing human skin generated entirely from pluripotent stem cells. *Nature*, 582(7812), 399-404.

54. Nguyen EH, et al. Versatile synthetic alternatives to Matrigel for vascular toxicity screening and stem cell expansion. *Nat. Biomed. Eng.* 1, 1-14 (2017).

55. Isenberg EA, Murphy WL. Synthetic alternatives to Matrigel. *Nat. Rev. Mater* 5, 539-551 (2020).

In addition, the purpose of this research is to propose the concept of microbead in situ bioprinting for functional skin/muscle repair and provide inspiration for the research of intraoperative 3D bioprinting. Thus, the primary demand for biomaterials should be the most favorable biocompatibility, which led us to choose Matrigel. Synthetic biomaterials with high biocompatibility and stable mechanical and apathogenic properties would be developed as replacement to Matrigel, and the "bead-jet printing" strategy would be transferred to the new gel.

28. Relevant research articles related to 3D intra-tissue printing have not been cited. Authors should compare their printing technique with recently reported approaches including intravital 3D bioprinting (Urciuolo et al. *Nat. Biomed. Eng.* 2020) and direct in situ printing into muscle tissues (Russell et al. *ACS Appl. Biomater.* 2020).

Thanks for this suggestion. Now we have discussed the difference between bead-jet printing system and other 3D intra-tissue printing. The manuscript has been corrected accordingly.

Compared with conventional 3D intra-tissue printing techniques⁵¹⁻⁵², either microextrusion-based or photocrosslinking-based, the bead-jet printing provides an automated solution that combines the advantages of granular gel infiltration, stem cell transplantation, and free of additional scaffolding support.

The articles have been cited as following:

51. Russell CS, et al. In situ printing of adhesive hydrogel scaffolds for the treatment of skeletal muscle injuries. *ACS Applied. Bio. Materials.* 3, 1568-1579 (2020).

52. Urciuolo A, et al. *Intravital three-dimensional bioprinting. Nat. Biomedical. Eng. 4, 901-915 (2020).*

29. Authors do not discuss their proposed therapy in relation to others involving the use of hydrogels e.g. Han et al. *Sci. Adv.* 2018. They also do not discuss the merits of mesenchymal stromal cell therapy compared to transplantation of muscle stem cells (i.e. satellite cells) see Judson et al. *NPG Regen. Med.* 2020.

Your comments are highly appreciated and we added corresponding discussion in the manuscript:

To overcome the limitations of Matrigel, synthetic biomaterials with high biocompatibility and stable mechanical and apathogenic properties are good candidates. The design of hydrogel has been discussed and the manuscript has been corrected accordingly.

Recent studies about synthetic hydrogels may offer a potential alternative to natural biomaterials, as biochemical and biophysical properties can be modularly altered, and engineered materials do not suffer the risk of antigenicity or other inherent biological properties. For example, Han, et al engineered a synthetic hydrogel-based matrix and applied the matrix for regeneration of mice skeletal muscle⁵³. Therefore, for future clinical translation of this strategy, synthetic biomaterials with high biocompatibility, cell-ligation ability, biodegradability and appropriate viscoelastic and apathogenic properties should be innovated and commercialized to provide appropriate alternatives to Matrigel.

The reference article is as below:

53. Han WM, et al. *Synthetic matrix enhances transplanted satellite cell engraftment in dystrophic and aged skeletal muscle with comorbid trauma. Sci. Adv. 4, eaar4008 (2018).*

The merits of mesenchymal stromal cell therapy compared to transplantation of muscle stem cells has also been discussed. The regenerative potential of skeletal muscle is maintained by the heterogeneous population of muscle-resident stem/progenitor cells including satellite cells (SCs) capable of regenerating muscle fibers and maintaining a functional satellite pool³¹. However, each cell cycle shortens the telomeres of SCs, leading to cell senescence and a rapid decrease of the SC pool³². SCs after activation need strong paracrine support from their niche, because without it they will not be able to survive³³. In comparison, activated MSCs are not only able to differentiate into a specific cell lineage but may also establish a regenerative microenvironment by immunomodulatory activity regulating the local immune response, which is the most interesting features of the biology of MSCs. Thus, MSCs therapy has unique advantages in muscle repair and other tissue regeneration, including but not limited to self-renewal ability, multipotentiality and immunomodulation effects. The manuscript has been revised accordingly.

It is also reported that regenerative potential of skeletal muscle is maintained by the heterogeneous population of muscle-resident stem cells (MuSC)⁵⁶. Compared with MSCs, MuSCs suffer from quick cell senescence and a rapid decrease resulting from the shortening of telomeres in each cell cycle⁵⁷. Also, MuSCs after activation need strong paracrine support from their niche,

without which they will not be able to survive⁵⁸. Thus, MSCs therapy has unique advantages in muscle repair and other tissue regeneration, including but not limited to self-renewal ability, multipotentiality and immunomodulation effects.

The reference articles are as below:

56. Judson RN, Rossi F. Towards stem cell therapies for skeletal muscle repair. *NPJ Regen. Med.* 5, 1-6 (2020).

57. Decary S, et al. Shorter telomeres in dystrophic muscle consistent with extensive regeneration in young children. *Neuromuscul. Disord.* 10, 113-120 (2000).

58. Klimczak A, et al. Muscle stem/progenitor cells and mesenchymal stem cells of bone marrow origin for skeletal muscle regeneration in muscular dystrophies. *Arch. Immunol. Ther. Ex.* 66, 341-354 (2018).

References

1. Ayan B, et al. Aspiration-assisted bioprinting for precise positioning of biologics. *Sci. Adv.* 6, eaaw5111 (2020).
2. Qazi TH, Muir VG, Burdick JA. Methods to Characterize Granular Hydrogel Rheological Properties, Porosity, and Cell Invasion. *ACS Biomater. Sci. Eng.* 8, 1427-1442 (2022).
3. Muir VG, Qazi TH, Shan J, Groll Jr, Burdick JA. Influence of microgel fabrication technique on granular hydrogel properties. *ACS Biomater. Sci. Eng.* 7, 4269-4281 (2021).
4. Buzas EI. The roles of extracellular vesicles in the immune system. *Nat. Rev. Immunol.* 1-15 (2022).
5. Wiklander OP, Brennan MÁ, Lötval J, Breakefield XO, El Andaloussi S. Advances in therapeutic applications of extracellular vesicles. *Sci. Transl. Med.* 11, eaav8521 (2019).
6. Rani S, Ryan AE, Griffin MD, Ritter T. Mesenchymal stem cell-derived extracellular vesicles: toward cell-free therapeutic applications. *Mol. Ther.* 23, 812-823 (2015).
7. Kim H, et al. Comprehensive molecular profiles of functionally effective MSC-derived extracellular vesicles in immunomodulation. *Mol. Ther.* 28, 1628-1644 (2020).

8. Harrell CR, Jovicic N, Djonov V, Arsenijevic N, Volarevic V. Mesenchymal stem cell-derived exosomes and other extracellular vesicles as new remedies in the therapy of inflammatory diseases. *Cells* **8**, 1605 (2019).
9. Saldin LT, Cramer MC, Velankar SS, White LJ, Badylak SF. Extracellular matrix hydrogels from decellularized tissues: Structure and function. *Acta Biomater.* **49**, 1-15 (2017).
10. Qazi TH, *et al.* Programming hydrogels to probe spatiotemporal cell biology. *Cell Stem Cell*, (2022).
11. Soofi SS, Last JA, Liliensiek SJ, Nealey PF, Murphy CJ. The elastic modulus of Matrigel™ as determined by atomic force microscopy. *J. Struct. Biol.* **167**, 216-219 (2009).
12. Kane KI, *et al.* Determination of the rheological properties of Matrigel for optimum seeding conditions in microfluidic cell cultures. *AIP Adv.* **8**, 125332 (2018).
13. Rabata A, Fedr R, Soucek K, Hampl A, Koledova Z. 3D cell culture models demonstrate a role for FGF and WNT signaling in regulation of lung epithelial cell fate and morphogenesis. *Front. Cell Dev. Biol.* **574** (2020).
14. Aisenbrey EA, Murphy WL. Synthetic alternatives to Matrigel. *Nat. Rev. Mater.* **5**, 539-551 (2020).
15. Wagner O, *et al.* Biocompatible fluorinated polyglycerols for droplet microfluidics as an alternative to PEG-based copolymer surfactants. *Lab Chip.* **16**, 65-69 (2016).
16. Wu Y, Chen L, Scott PG, Tredget EE. Mesenchymal stem cells enhance wound healing through differentiation and angiogenesis. *Stem cells* **25**, 2648-2659 (2007).
17. Conconi MT, *et al.* CD105 (+) cells from Wharton's jelly show in vitro and in vivo myogenic differentiative potential. *Int. J. Mol. Med.* **18**, 1089-1096 (2006).
18. Harada N, *et al.* Bone regeneration in a massive rat femur defect through endochondral ossification achieved with chondrogenically differentiated MSCs in a degradable scaffold. *Biomaterials* **35**, 7800-7810 (2014).
19. Scotti C, *et al.* Recapitulation of endochondral bone formation using human adult mesenchymal stem cells as a paradigm for developmental engineering. *PNAS.* **107**, 7251-7256 (2010).
20. Sicari BM, *et al.* An acellular biologic scaffold promotes skeletal muscle formation in mice and humans with volumetric muscle loss. *Sci. Transl. Med.* **6**, 234ra258-234ra258 (2014).
21. Molina T, Fabre P, Dumont NA. Fibro-adipogenic progenitors in skeletal muscle homeostasis,

- regeneration and diseases. *Open Biol.* **11**, 210110 (2021).
22. Liu L, Cheung TH, Charville GW, Rando TA. Isolation of skeletal muscle stem cells by fluorescence-activated cell sorting. *Nat. Protoc.* **10**, 1612-1624 (2015).
 23. McKellar DW, *et al.* Large-scale integration of single-cell transcriptomic data captures transitional progenitor states in mouse skeletal muscle regeneration. *Commun. Biol.* **4**, 1-12 (2021).
 24. Park S-Y, *et al.* Stabilin-2 modulates the efficiency of myoblast fusion during myogenic differentiation and muscle regeneration. *Nat. Commun.* **7**, 1-15 (2016).
 25. Bersini S, *et al.* Engineering an environment for the study of fibrosis: a 3D human muscle model with endothelium specificity and endomysium. *Cell Rep.* **25**, 3858-3868. e3854 (2018).
 26. Seo BR, *et al.* Skeletal muscle regeneration with robotic actuation-mediated clearance of neutrophils. *Sci. Transl. Med.* **13**, eabe8868 (2021).
 27. Wagers AJ, Conboy IM. Cellular and molecular signatures of muscle regeneration: current concepts and controversies in adult myogenesis. *Cell* **122**, 659-667 (2005).
 28. Russell CS, *et al.* In situ printing of adhesive hydrogel scaffolds for the treatment of skeletal muscle injuries. *ACS Appl. Bio Mater.* **3**, 1568-1579 (2020).
 29. Urciuolo A, *et al.* Intravital three-dimensional bioprinting. *Nat. Biomed. Eng.* **4**, 901-915 (2020).
 30. Han WM, *et al.* Synthetic matrix enhances transplanted satellite cell engraftment in dystrophic and aged skeletal muscle with comorbid trauma. *Sci. Adv.* **4**, eaar4008 (2018).
 31. Judson RN, Rossi F. Towards stem cell therapies for skeletal muscle repair. *NPJ Regen.* **5**, 1-6 (2020).
 32. Decary S, Hamida CB, Mouly V, Barbet J, Hentati F, Butler-Browne G. Shorter telomeres in dystrophic muscle consistent with extensive regeneration in young children. *Neuromuscul. Disord.* **10**, 113-120 (2000).
 33. Klimczak A, Kozłowska U, Kurpisz M. Muscle stem/progenitor cells and mesenchymal stem cells of bone marrow origin for skeletal muscle regeneration in muscular dystrophies. *Arch. Immunol. Ther. Exp.* **66**, 341-354 (2018).
 34. Chaudhuri, O, *et al.* Effects of extracellular matrix viscoelasticity on cellular behaviour. *Nature* **584**, 535-546 (2020).

35. Lutolf MP, *et al.* Designing materials to direct stem-cell fate. *Nature*. **462**, 433-441 (2009).
36. Lee, J, *et al.* Hair-bearing human skin generated entirely from pluripotent stem cells. *Nature* **582**, 399-404 (2020).

REVIEWERS' COMMENTS

Reviewer #1 (Remarks to the Author):

Changes are sufficient. New results were included to address the queries raised.

Reviewer #2 (Remarks to the Author):

The authors have addressed this reviewers comments satisfactory and would be supportive of its publications.

Reviewer #4 (Remarks to the Author):

I have re-read the manuscript and the responses including the new data that has been added. My concerns about the impact of this paper remain the same. Consider:

In response to queries 1 and 2 about the broad applicability of the technique and the obvious issues of using Matrigel, the authors first state that "the purpose of this research is to propose the concept of microbead in situ bioprinting for functional skin/muscle repair and provide inspiration for the research of intraoperative 3D bioprinting" – and "cells could not grow, expand and morphogenesis in HAMA gel or beads" – and "Based on the molecular properties of alginate and PEG, we would expect a similar result in the growth and behaviors of encapsulated cells." – and "revealed the advantaged [sic] of Matrigel as the cell encapsulation scaffold, though it remained with the challenge of clinical translation."

So, in essence, not only is Matrigel not translatable, but this technique or the advantages in terms of cell behavior and tissue repair may not be well suited for other polymers like alginate (clinically translated), hyaluronic acid (clinically translated), or PEG (clinically translated). The reader is left to wonder what then is the impact of this study for the field?

The new data added by the authors (rheology, printing patterns, in vivo groups) have addressed many of the technical concerns and questions. The authors should also realize that they are not the first to propose or test intraoperative bioprinting, and relevant previous work that have not been mentioned in the manuscript should be cited.

REVIEWER COMMENTS

I have re-read the manuscript and the responses including the new data that has been added. My concerns about the impact of this paper remain the same. Consider:

In response to queries 1 and 2 about the broad applicability of the technique and the obvious issues of using Matrigel, the authors first state that “the purpose of this research is to propose the concept of microbead in situ bioprinting for functional skin/muscle repair and provide inspiration for the research of intraoperative 3D bioprinting” – and “cells could not grow, expand and morphogenesis in HAMA gel or beads” – and “Based on the molecular properties of alginate and PEG, we would expect a similar result in the growth and behaviors of encapsulated cells.” – and “revealed the advantaged [sic] of Matrigel as the cell encapsulation scaffold, though it remained with the challenge of clinical translation.”

So, in essence, not only is Matrigel not translatable, but this technique or the advantages in terms of cell behavior and tissue repair may not be well suited for other polymers like alginate (clinically translated), hyaluronic acid (clinically translated), or PEG (clinically translated). The reader is left to wonder what then is the impact of this study for the field?

The new data added by the authors (rheology, printing patterns, in vivo groups) have addressed many of the technical concerns and questions. The authors should also realize that they are not the first to propose or test intraoperative bioprinting, and relevant previous work that have not been mentioned in the manuscript should be cited.

Thanks for raising these comments.

The author agrees that the inherent properties of Matrigel hampers the clinical translation as the scaffolding materials for tissue transplantation.

However, though gelatin, hyaluronic acid¹, PEG² and alginate³ have been commercialized for clinical applications, the biocompatibility of these materials has not been verified in stem cells before and not well suited in MSCs activity compared with Matrigel that's verified in this study (**Figures 3, S17 and S18**). The drawbacks of these materials may be the potential risks brought from the crosslinking reagents including methacryloyl groups⁴ or calcium chloride⁵, as well as the absence of adhesion sites and enzymatic degradation targets for cells, which go against the differentiation and functional expression of stem cells.

Matrigel is favorable of supporting for cell proliferation and migration and could resemble the microenvironment of extracellular matrix for its basement membrane-like nature, thus we select the material as the scaffolding materials in this study. Matthias et al. also noted that Matrigel has advantages over artificial biomaterials in directing stem-cell rate⁶, that's able to regulate the functional role of MSCs during the regeneration and facilitate the differentiation of MSCs into myofiber or hair follicles (**Figures 6 and 8**), that's verified in this study.

Multiple biomaterials with various mechanical properties are compatible with the bead-jet printing system, including gelatin, GelMA, HAMA, and Matrigel (**Figures 2, 3, S17 and S18**). This indicates the printer prototype could deal with various ultra-soft and rigid materials. Therefore, we believe our printing system could also be applied in other synthetic alternatives into Matrigel that maintain comparable biocompatibility as Matrigel. It is expected to catalyze the translation of stem cell therapy. While designing the synthetic alternatives, the properties of these materials should be highly biocompatible, biodegradable, appropriately viscoelastic and apathogenic, and easy to ligate

for cells. Recently, synthetic alternatives into Matrigel have been increasingly innovated and developed in this field^{7,8}. We believe new synthetic materials will be developed in the near future and our printer is ready for the transformation. However, such new materials reported with equivalent biocompatibility and viscoelasticity have yet not been commercialized.

To summarize, we agree and am aware aware that Matrigel cannot be used for transplantation even in the future. For one aspect, this is a conceptual study and reports a technology advance and a new principle to augment stem cell therapy. For another aspect, we have been devoting to new materials development as replacement of Matrigel, which have received the support from Merck group, as the sole-project supported by MERCK RESEARCH GRANT 2021 in biomedicine. It aims to solve the global challenge of the short of Matrigel-replacement.

The drawbacks of Matrigel have been discussed in the manuscript as follows.

Page 28, line 15-Page 29, line 3: However, though the cytocompatibility of Matrigel is favorable, the clinical promise is limited due to its tumor-derived, ill-defined, variable composition and low availability for lab-used condition⁵⁵⁻⁵⁶. The potential for antigenicity is one of the inherent limitations. Recent studies about synthetic hydrogels may offer a potential alternative to natural biomaterials, as biochemical and biophysical properties can be modularly altered, and engineered materials do not suffer the risk of antigenicity or other inherent biological properties. For example, Han, et al engineered a synthetic hydrogel-based matrix and applied the matrix for regeneration of mice skeletal muscle⁵⁷. Therefore, for future clinical translation of this strategy, synthetic biomaterials with high biocompatibility, cell-ligation ability, biodegradability and appropriate viscoelastic and apathogenic properties should be innovated and commercialized to provide appropriate alternatives to Matrigel⁵⁸.

We also agree that we are not the first to propose or test intraoperative bioprinting. Multiple types of intraoperative bioprinting technique have been developed. Urciuolo, et al⁹ developed an intravital 3D bioprinting using 3D cell-laden photosensitive polymer hydrogels for organ repair or reconstruction. Russell, et al¹⁰ directly printed acellular gelatin-based hydrogels into the defect site for mice muscle regeneration. Moncal, et al¹¹ used droplet-based bioprinting comprising the soft tissue ink to accelerate the reconstruction of full-thickness skin defects and facilitate up to 60% wound closure in 6 day. Recently, Xie, et al¹² designed a bioconcrete bioink with cell-laden microgels and verified the superiority of this bioink and its potential in clinical settings.

The bead-jet printer developed in this study is specialized for stem cell therapy, and is the first report of gel microbeads-based printing for live cells with high-throughput capacity. Multiple issues related to stem cell therapy during clinical translation have been investigated, including the loading efficiency of stem cells and the retention rate of stem cells. Sparsely patterning of high density MSCs-laden beads among acellular beads is able to enlarge the transplantation coverage, increase the cell loading number and cell retention time, thus enabling therapy of large-scale traumatic injury with a limited number of MSCs. Therefore, we believe the bead-jet printing system might

revolutionize the stem cell therapy and broaden the applicability of intraoperative bioprinting.

Relevant previous work has been cited and discussed in the manuscript.

Page 28, line 3-10: Bead-jet printing also broadens the applicability of intraoperative bioprinting. Compared with conventional 3D in-situ printing techniques, including intravital 3D bioprinting⁵¹, directly printing⁵², droplet-based bioprinting⁵³ and in situ 3D bioprinting with bioconcrete bioink⁵⁴, the bead-jet printing is specialized for stem cell therapy with improved loading efficiency and retention rate of stem cells in vivo. Sparsely patterning of high density MSCs-laden beads among acellular beads is able to enlarge the transplantation coverage, thus enabling therapy of large-scale traumatic injury with a limited number of MSCs.

1 Price, R. D., Berry, M. G., & Navsaria, H. A. (2007). Hyaluronic acid: the scientific and clinical evidence. *Journal of Plastic, Reconstructive & Aesthetic Surgery*, 60(10), 1110-1119.

2 Pasut, G., & Veronese, F. M. (2009). PEG conjugates in clinical development or use as anticancer agents: an overview. *Advanced drug delivery reviews*, 61(13), 1177-1188.

3 Zimmermann, U., Thürmer, F., Jork, A., Weber, M., Mimietz, S., Hillgärtner, M., ... & Hendrich, C. (2001). A novel class of amitogenic alginate microcapsules for long-term immunisolated transplantation. *Annals of the New York Academy of Sciences*, 944(1), 199-215.

4 Su, Jimmy Jiun-Ming, et al. "Biofabrication of cell-laden gelatin methacryloyl hydrogels with incorporation of silanized hydroxyapatite by visible light projection." *Polymers* 13.14 (2021): 2354.

5 Lee, Gyun Min, et al. "Effect of calcium chloride treatment on hybridoma cell viability and growth." *Biotechnology letters* 14.10 (1992): 891-896.

6 Lutolf, Matthias P., Penney M. Gilbert, and Helen M. Blau. "Designing materials to direct stem-cell fate." *Nature* 462.7272 (2009): 433-441.

7. Nguyen EH, et al. Versatile synthetic alternatives to Matrigel for vascular toxicity screening and stem cell expansion. *Nat. Biomed. Eng.* 1, 1-14 (2017).

8. Isenberg EA, Murphy WL. Synthetic alternatives to Matrigel. *Nat. Rev. Mater* 5, 539-551 (2020).

9 Urciuolo, Anna, et al. "Intravital three-dimensional bioprinting." *Nature biomedical engineering* 4.9 (2020): 901-915.

10 Russell, Carina S., et al. "In situ printing of adhesive hydrogel scaffolds for the treatment of skeletal muscle injuries." *ACS Applied Bio Materials* 3.3 (2020): 1568-1579.

11 Moncal, Kazim K., et al. "Intraoperative Bioprinting of Hard, Soft, and Hard/Soft Composite Tissues for Craniomaxillofacial Reconstruction." *Advanced Functional Materials* 31.29 (2021): 2010858.

12 Xie, Mingjun, et al. "In situ 3D bioprinting with bioconcrete bioink." *Nature Communications* 13.1 (2022): 1-12.